# Revisiting Follow-the-Perturbed-Leader with Unbounded Perturbations in Bandit Problems

**Jongyeong Lee**
Korea Institute of Science and Technology[*]
jongyeong@kist.re.kr

**Junya Honda**
Kyoto University, RIKEN AIP
honda@i.kyoto-u.ac.jp

**Shinji Ito**
The University of Tokyo, RIKEN AIP
shinji@mist.i.u-tokyo.ac.jp

**Min-hwan Oh**
Seoul National University
minoh@snu.ac.kr

## Abstract

Follow-the-Regularized-Leader (FTRL) policies have achieved Best-of-Both-Worlds (BOBW) results in various settings through hybrid regularizers, whereas analogous results for Follow-the-Perturbed-Leader (FTPL) remain limited due to inherent analytical challenges. To advance the analytical foundations of FTPL, we revisit classical FTRL-FTPL duality for unbounded perturbations and establish BOBW results for FTPL under a broad family of asymmetric unbounded Fréchet-type perturbations, including hybrid perturbations combining Gumbel-type and Fréchet-type tails. These results not only extend the BOBW results of FTPL but also offer new insights into designing alternative FTPL policies competitive with hybrid regularization approaches. Motivated by earlier observations in two-armed bandits, we further investigate the connection between the $1/2$-Tsallis entropy and a Fréchet-type perturbation. Our numerical observations suggest that it corresponds to a symmetric Fréchet-type perturbation, and based on this, we establish the first BOBW guarantee for symmetric unbounded perturbations in the two-armed setting. In contrast, in general multi-armed bandits, we find an instance in which symmetric Fréchet-type perturbations violate the key condition for standard BOBW analysis, which is a problem not observed with asymmetric or nonnegative Fréchet-type perturbations. Although this example does not rule out alternative analyses achieving BOBW results, it suggests the limitations of directly applying the relationship observed in two-armed cases to the general case and thus emphasizes the need for further investigation to fully understand the behavior of FTPL in broader settings.

## 1 Introduction

In multi-armed bandit (MAB) problems, an agent plays an arm $I_t$ from a set of $K$ arms at each round $t \in [T] := \{1, \ldots, T\}$ over a time horizon $T$. After playing an arm, the agent observes only the loss $\ell_{t,I_t}$ of the played arm, where the loss vectors $\ell_t = (\ell_{t,1}, \ldots, \ell_{t,K})^\top \in [0,1]^K$ are determined by the environment. Given the constraints of partial feedback, the agent must handle the tradeoff between gathering information about the arms and playing arms strategically to minimize total loss.

Although numerous policies have been developed for MAB, many of them can be largely classified into two primary frameworks, namely Follow-the-Perturbed-Leader (FTPL) [32] and Follow-the-Regularized-Leader (FTRL) [6]. While FTPL was inspired by a game theoretic approach [22, 25] and FTRL emerged from the context of online optimization [26], both frameworks have analogs

---

[*]He was affiliated with Seoul National University at the time of submission.

39th Conference on Neural Information Processing Systems (NeurIPS 2025).

in discrete choice theory, a field in economics that models decision-making through probabilistic frameworks to maximize utility over finite alternatives [51]. In particular, FTPL is known as the additive random utility model [5, 50], while FTRL corresponds to the representative agent model [4].

Beyond these conceptual analogies, a line of work has formalized the relationship between FTPL and FTRL in discrete choice theory [21, 27]. In particular, when the joint perturbation distribution has a strictly positive density on $\mathbb{R}^K$, the existence of a corresponding regularizer, along with detailed results on its properties, has been established [19, 44]. A classical example is the multinomial logit model [40], known to be equivalent to both FTPL with Gumbel perturbations (also known as Exp3 [7, 48]) and FTRL with a Shannon entropy regularizer [4]. In the context of online learning, Abernethy et al. [3] discussed this relationship for general perturbations, where the formal theorem was later established for the independent and identically distributed (i.i.d.) perturbations absolutely continuous with respect to Lebesgue measure by Suggala and Netrapalli [49, Proposition 3.1].

While the above results mainly discuss the transformation of FTPL into FTRL, Abernethy et al. [1, 3] demonstrated that (nearly) every instance of FTRL can be viewed as a special case of FTPL in one-dimensional online optimization and more general equivalences are discussed by Feng et al. [19]. Nevertheless, when $K \geq 4$, no FTPL counterpart exists for FTRL with log-barrier regularizer [27, Proposition 2.2] and Tsallis entropy regularizer [33, Theorem 8]. These findings indicate that FTRL strictly subsumes FTPL as a special case.

Despite this narrower coverage, FTPL has gained significant attention due to its computational efficiency and simplicity, making it suitable for a variety of problems in online learning, including combinatorial semi-bandits [43], online learning with non-linear losses [17], and MDP bandits [15]. Still, while FTRL policies have achieved optimal results in several problems such as graph bandits [16] and partial monitoring [52], comparable progress for FTPL has been relatively underexplored.

This gap is primarily due to the complexity of expressing arm-selection probabilities of FTPL, which poses significant analytical challenges despite its computational efficiency in practice. In the standard analysis of FTRL and FTPL, a key factor in achieving the optimal adversarial regret is evaluating the stability of the arm-selection probability against the changes in the estimated cumulative loss. Abernethy et al. [2] tackled this challenge by leveraging the hazard function of perturbations, but it only resulted in near-optimal regret of $\mathcal{O}(\sqrt{KT \log K})$. Later, motivated by the observation in two-armed bandits that FTRL with $\beta$-Tsallis entropy roughly correspond to Fréchet-type perturbations with tail index $(1-\beta)^{-1}$, Kim and Tewari [33] conjectured that FTPL with Fréchet-type distributions could achieve optimal $\mathcal{O}(\sqrt{KT})$ regret. Recently, this conjecture has been partially validated, as it was shown that FTPL with nonngetative Fréchet-type distributions, under certain conditions, indeed obtain the optimal $\mathcal{O}(\sqrt{KT})$ regret and even Best-of-Both-Worlds (BOBW) result [12, 28, 38]. Here, unfamiliar readers can think of a Fréchet-type as any distributions whose right tail decays polynomially.

**Contribution** Since hybrid regularizers are typically constructed by summing distinct regularizers [29, 56], a natural analogue for FTPL is to construct hybrid perturbations, either by summing perturbations drawn from different distributions or by specifying different behaviors for the left and right parts of the density. This motivates the study of asymmetric or unbounded perturbations, where "unbounded" distributions in this paper refers to distributions that are unbounded on both the positive and negative sides and "semi-infinite" refers to those unbounded on only one side. However, the requirement for existing results is easily violated when the perturbation has strictly positive density on $\mathbb{R}$ since existing results require the ratio of density to cumulative distribution function, $f/F$, to be monotonically decreasing over the entire support.

In this paper, we first relax these previous conditions for the BOBW guarantee by providing milder conditions under which FTPL with unbounded perturbations achieves optimal $\mathcal{O}(\sqrt{KT})$ regret and even logarithmic stochastic regret, thereby achieving the BOBW result. As shown below, the key conditions are related to the difference in tail indices of right and left tails, where (right) tail index $\alpha$ roughly corresponds to the order of polynomial decay, $\Pr_{X \sim \mathcal{D}}[X \geq x] \approx x^{-\alpha}$.

**Proposition 1.1** (asymmetric, informal). *Let $\mathcal{D}_\alpha$ denote the unimodal Fréchet-type distribution fully supported on $\mathbb{R}$ under mild conditions, with right and left tails characterized by tail indices $2$ and $\alpha > 0$, respectively. If $\alpha \in \mathbb{R}_{\geq 4} \cup \{\infty\}$, FTPL with $\mathcal{D}_\alpha$ achieves BOBW guarantee for $K \geq 2$.*

Here, Fréchet-type perturbation with infinite index denotes the Gumbel-type family, which is of exponential tail such as exponential distribution and Gamma distribution. Consequently, $\mathcal{D}_\alpha$ with $\alpha =$

$\infty$ can be seen as a hybrid perturbation, combining the Gumbel-type and Fréchet-type distributions, analogous to the hybrid regularizers widely used in FTRL [11, 31]. While it remains unclear which regularizers are associated with these hybrid perturbations, we expect that our findings can provide insights into developing an FTPL policy that serves as an alternative to a hybrid regularizer for FTRL. For example, we expect it can approximately reproduce the combination of Tsallis entropy with Shannon entropy, $-x \log x$, or Shannon entropy for the complement, $-(1-x) \log(1-x)$, where the latter has been used in FTRL policies [52, 56].

While asymmetry appears natural in the context of hybrid perturbations, one may wonder whether it is essential or just a technical artifact. To explore this, we revisit the original motivation for using Fréchet-type perturbation, specifically its equivalence to FTRL with Tsallis entropy in the two-armed bandit setting [33], where the latter is a well-known BOBW policy [55]. Although the exact distribution is not available in closed form, our numerical analysis suggests that the perturbation associated with $1/2$-Tsallis entropy for $K = 2$ can be realized by a symmetric Fréchet-type distribution. Therefore, it is natural to expect that the BOBW guarantee can be extended to symmetric Fréchet-type at least for $K = 2$, and perhaps even $K \geq 3$. However, our analysis reveals a limitation in the latter case.

**Proposition 1.2** (symmetric, informal). *For FTPL with Fréchet-type distributions with both left and right tail indices equal to 2, the BOBW result holds when $K = 2$. However, for $K \geq 3$, the key conditions required by standard BOBW analyses are violated unlike the asymmetric tail indices.*

Details on this key condition are provided in later sections. Here, it is worth noting that no perturbation exactly reproduces Tsallis entropy for $K \geq 4$ [33]. Yet, the regularizer induced by the symmetric Pareto distribution, the simplest example of symmetric Fréchet-type with density $f(x) = 1/(|x|+1)^3$, behaves like the Tsallis entropy even in the counterexample for $K \geq 3$. Although alternative, non-standard techniques might be able to establish general BOBW guarantees for symmetric Fréchet perturbations, our findings, together with the impossibility result in Kim and Tewari [33], highlights the difficulty of extending the BOBW guarantees beyond $K = 2$. Further investigation is therefore needed to develop a deeper and more comprehensive understanding of FTPL in broader settings.

## 2 Preliminaries

In bandit problems, the loss vectors $\ell_t$ are determined typically in either a stochastic or adversarial manner. In the stochastic setting, the loss vector $\ell_t$ is i.i.d. from an unknown but fixed distribution over $[0, 1]^K$. Hence, one can define the expected losses of arms $\mu_i := \mathbb{E}[\ell_{t,i}]$ and the optimal arm $i^* \in \arg\min_{i \in [K]} \mu_i$. The suboptimality gap of each arm is denoted by $\Delta_i = \mu_i - \mu_{i^*}$ and the optimal problem-dependent regret bound is known to be $\sum_{i:\Delta_i > 0} \mathcal{O}(\log T / \Delta_i)$ [36]. On the other hand, in the adversarial setting, an (adaptive) adversary determines the loss vector based on the history of the decisions, and thus no specific assumptions are made about the loss distribution. In this environment, the optimal regret bound is $\mathcal{O}(\sqrt{KT})$ [8]. A policy is called a BOBW policy when the policy achieves (near) optimal guarantee for both stochastic and adversarial setting [10].

### 2.1 Follow-the-Perturbed-Leader and Follow-the-Regularized-Leader policies

Since the agent only observes the loss of the played arm, $\ell_{t,I_t}$, one uses the loss estimator $\hat{\ell}_t$. Let $\hat{L}_t = \sum_{s=1}^{t-1} \hat{\ell}_s$ denote the estimated cumulative loss up to round $t$, with $L_t = \sum_{s=1}^{t-1} \ell_s$ as the true cumulative loss. Then FTPL is a policy that plays an arm

$$I_t \in \arg\min_{i \in [K]} \left\{ \hat{L}_{t,i} - \frac{r_{t,i}}{\eta_t} \right\}, \quad \text{where} \quad r_t \sim \mathcal{D}^K, \tag{FTPL}$$

where $\eta_t$ is the learning rate specified later and $r_t = (r_{t,1}, \ldots, r_{t,K})$ denotes the random perturbation vector drawn from the distribution $\mathcal{D}^K$ over $\mathbb{R}^K$. In the case of i.i.d. perturbations, which are common in the bandit literature, we denote the common distribution by $\mathcal{D}$. The probability of playing an arm $i \in [K]$ by FTPL, given $\hat{L}_t$, is denoted by $w_{t,i} = \phi_i(\eta_t \hat{L}_t; \mathcal{D}^K)$, where for $\lambda \in \mathbb{R}^K$

$$\phi_i(\lambda; \mathcal{D}^K) := \Pr_{r \sim \mathcal{D}^K} \left[ i = \arg\min_{j \in [K]} \{\lambda_j - r_j\} \right]. \tag{1}$$

When the perturbations are i.i.d., we denote $\phi_i(\lambda; \mathcal{D}^K)$ by $\phi_i(\lambda; \mathcal{D})$. We denote the distribution function and density function of $\mathcal{D}$ by $F$ and $f$, respectively. Meanwhile, FTRL plays an arm

according to the probability vector $p_t$, i.e., $I_t \sim p_t$, defined as

$$p_t = p(\eta_t \hat{L}_t; V) := \underset{p \in \mathcal{P}_{K-1}}{\arg\min} \left\{ \left\langle \hat{L}_t, p \right\rangle + \frac{V(p)}{\eta_t} \right\}, \tag{FTRL}$$

where $\mathcal{P}_K$ denotes $K$ dimensional probability simplex and $V : \mathcal{P}_{K-1} \to \mathbb{R}_{\geq 0}$ denotes a regularizer function. Therefore, FTPL and FTRL are equivalent if $\phi_i(\lambda; \mathcal{D}) = p_i(\lambda; V)$ holds for any $\lambda \in \mathbb{R}^K$. For example, it is known that $p(\lambda; V_S) = \phi(\lambda; \text{Gumbel})$, where $V_S(p) = \sum_{i \in [K]} p_i \log p_i$ denotes negative Shannon entropy [4].

For the unbiased loss estimator, both FTPL and FTRL policies often use an importance-weighted (IW) estimator $\hat{\ell}_t = \ell_t e_{I_t} / \Pr[I_t = i]$ of the loss vector $\ell_t$, where $e_i$ is the $i$-th standard basis vector. In general, $\phi_i$ does not have a closed form, making computations of $\phi_i$ and the IW estimator $\hat{\ell}_t$ difficult. To address this, FTPL policies usually construct $\hat{\ell}_t$ with a geometric resampling estimator of $1/w_{t,i}$, instead of explicitly computing $w_{t,i}$ [28, 43]. Unlike FTPL, FTRL can directly use $p_{t,i}$ to construct $\hat{\ell}_t$ since it plays an arm according to $p_t$ obtained by solving an optimization problem.

## 2.2 Standard analysis techniques

The standard FTRL regret analysis proceeds by decomposing the regret (largely) into two pieces, (i) a stability term, which is usually related to the $p_t$ itself, and (ii) a penalty term that is related to the value of regularizer function $V(p_t)$ [30, 37, 45]. By choosing the regularizer and learning rate so that these two terms are of the same order, several FTRL policies obtain the optimal performance [16, 29, 31, 55]. In this context, it is known that a uniform bound on $-p_i'(\eta_t \hat{L}_t)/p_{t,i}^{3/2}$ is a sufficient condition to guarantee $\mathcal{O}(\sqrt{KT})$ regret, where $p_i'(\lambda) = \partial p_i(\lambda)/\partial \lambda_i$ [9], as this ensures the stability and penalty terms remain of the same order. For FTRL with $\beta$-Tsallis entropy, defined as $-\frac{1}{1-\beta}\sum p_i^\beta$, refined analysis showed that a uniform bound on $p_i'/p_i^{2-\beta}$ is sufficient to guarantee $\mathcal{O}(\sqrt{KT})$ regret for $\beta \in (0,1)$ [2]. Zimmert and Seldin [55] further provided a unified analysis that covers $\beta \in [0,1]$, achieving $\mathcal{O}(\sqrt{KT \log K})$ regret for $\beta = 1$, $\mathcal{O}(\sqrt{KT \log T})$ regret for $\beta = 0$ and even BOBW guarantee for $\beta = 1/2$.

While FTRL with Tsallis entropy has achieved the optimal $\mathcal{O}(\sqrt{KT})$ regret in MAB problems, Kim and Tewari [33] showed that no corresponding FTPL policy exists for FTRL with the Tsallis entropy regularizer when $K \geq 4$. This makes it unclear how to apply the standard FTRL techniques to FTPL to achieve the optimal results. As a result, only a near optimal $\mathcal{O}(\sqrt{KT \log K})$ regret was obtained by considering a uniform bound on $-\phi_i'/\phi_i$ [2], rather than $-\phi_i'/\phi_i^{2-\beta}$. To address this challenge, Honda et al. [28] and Lee et al. [38] derived refined bounds on $\phi_i'(\eta_t \hat{L}_t)/w_{t,i}$ that depends on $1/\hat{L}_{t,i}$, which is indeed related to $w_{t,i}^{1/\alpha}$, rather than relying on a uniform constant. Here, $\alpha$ denotes the index of Fréchet-type perturbations. The dependency of $1/\alpha$ arises naturally from observations in two-armed bandits, where FTRL with $\beta$-Tsallis entropy roughly corresponds to FTPL with Fréchet-type perturbations with index $\alpha = (1-\beta)^{-1}$, which will be elaborated in Section 5.

This refined analysis enables the derivation of BOBW guarantees for FTPL policies, illustrating optimal performance (up to multiplicative constant) in both stochastic and adversarial settings. In particular, they derived that for adversarial regret, $-\phi_i'(\eta_t \hat{L}_t)/w_{t,i} = \mathcal{O}(\sigma_i^{-1/\alpha})$ when $\hat{L}_{t,i}$ is the $\sigma_i$-th smallest among the components of $\hat{L}_t$, and for stochastic regret, $-\phi_i'(\eta_t \hat{L}_t)/w_{t,i} = \mathcal{O}(1/\hat{L}_{t,i})$. These bounds provide a more refined characterization of the sensitivity of arm-selection probabilities to changes in estimated losses, leading to improved regret analyses for FTPL.

## 2.3 Previous assumptions on perturbations in FTPL literature

In econometrics, FTPL with i.i.d. Gumbel distribution has been extensively studied since its arm-selection probability model can be written in the closed-form, known as the multinomial logit model [4, 40]. On the other hand, in the online learning literature, Kalai and Vempala [32] introduced FTPL with the exponential distribution. Following this work, FTPL policies for bandits have mainly adopted exponential distributions [34, 35, 43, 46], where near-optimal results are established.

Beyond distribution-specific analysis, based on the relationship between FTRL and FTPL, Abernethy et al. [2] provided unified analysis for perturbations with bounded hazard function $f/(1-F)$. However, using this result with best tuning of parameters always leads to near-optimal $\mathcal{O}(\sqrt{KT \log K})$

regret [33], which provokes to explore different approaches to achieve the minimax optimality. While there is no corresponding FTPL for FTRL with Tsallis entropy in general, Honda et al. [28] showed that FTPL with Fréchet perturbation with shape 2 can achieve optimal $\mathcal{O}(\sqrt{KT})$ regret and even BOBW result, which is generalized to Fréchet-type perturbations satisfying certain conditions [38]. To avoid unnecessary technicalities, we state here a minimal, sufficient definition and the key relevant conditions, where the full and precise statements appear in Appendix A for completeness.

**Definition 2.1** (Informal). A distribution is said to be Fréchet-type with index $\alpha$ if the tail function satisfies $1 - F(x) = \tilde{\mathcal{O}}(x^{-\alpha})$ for $x > 0$, where $\tilde{\mathcal{O}}$ hides polylogarithmic factors.

**Assumption 2.2.** $\text{supp}\mathcal{D} \subseteq [0, \infty)$ and the hazard function $f/(1 - F)$ is bounded.

**Assumption 2.3.** $f/F$ is monotonically decreasing in the whole support.

Let $\mathfrak{D}_\alpha$ denote the set of distributions satisfying all the conditions in Lee et al. [38] given in Appendix A.1 including Assumptions 2.2 and 2.3, where index $\alpha > 0$ represents the tail index. It was shown that FTPL with $\mathcal{D} \in \mathfrak{D}_\alpha$ can achieve $\mathcal{O}(\sqrt{KT})$ adversarial regret for any $\alpha > 1$ and further $\mathcal{O}(\sum_{i \neq i^*} \log T / \Delta_i)$ stochastic regret for $\alpha = 2$. Although the nonnegativity assumption was introduced for simplicity, it is worth noting that Assumption 2.3 can be easily violated when $\text{supp}\mathcal{D} = \mathbb{R}$, which implicitly requires the use of semi-infinite perturbations.

## 3  Relationship between FTPL and FTRL

In this section, we revisit classical results in discrete choice theory under the assumption that the density is strictly positive on $\mathbb{R}^K$. While the existence of a corresponding regularizer for FTPL has been extended to general distributions [19, 49], unbounded perturbations remain the most thoroughly analyzed and thus offer a solid foundation for understanding FTPL. We also note that, unlike the bandit settings where FTPL typically uses i.i.d. perturbations, discrete choice models allow for correlated perturbations.

### 3.1  Classical results in discrete choice theory

For perturbations fully supported on $\mathbb{R}^K$, the existence of a corresponding regularizer has been established by classical results in discrete choice theory [27, 41]. Following the conventions in discrete choice theory, where the objective is to maximize rewards (which is analogous to minimizing losses), in this section, we redefine the arm-selection probability for arm $i$ in terms of cumulative rewards $\nu$, which corresponds to the negative loss, instead of cumulative loss $\lambda$ as

$$\varphi_i(\nu) = \Pr \left[ i = \arg\max_{j \in [K]} \{\nu_j + r_j\} \right], \tag{2}$$

where $r_j$s are random perturbations. This definition relates to the loss-based definition $\phi_i(\lambda) = \varphi_i(\nu)$ as the gap of cumulative losses $\lambda_i$ in (1) can be written in terms of the gap of cumulative rewards by $\lambda_i - \min_j \lambda_j = \max_j \nu_j - \nu_i$. It is known that for any given FTPL policy, a corresponding FTRL always exists, although the associated regularizer may lack a closed-form.

**Lemma 3.1** (Theorem 2.1 in Hofbauer and Sandholm [27]). *For $\varphi$ defined in (2), let the joint distribution of random vector $r$ be of finite mean and absolutely continuous, and fully supported on $\mathbb{R}^K$. Then, there exists a regularization function $V : \text{Int}(\mathcal{P}_{K-1}) \to \mathbb{R}$ such that*

$$\varphi(\nu) = \arg\max_{p \in \text{Int}(\mathcal{P}_{K-1})} \{\langle p, \nu \rangle - V(p)\},$$

*where* Int *denotes the interior. Moreover, V can be obtained as the Legendre transformation of the potential function $\Phi$ of $\varphi$, i.e.,*

$$V(p) = \Phi^*(p) := \sup_{\nu \in \mathbb{R}^K} \langle p, \nu \rangle - \Phi(\nu), \quad \text{where} \quad \nabla\Phi(\nu) = \varphi(\nu). \tag{3}$$

Here, potential function of $\varphi(\nu)$, denoted by $\Phi(\nu)$, is known as surplus function of the discrete choice model [42], which is defined as $\Phi(\nu) = \mathbb{E}_r\left[\max_{i \in [K]} \nu_i + r_i\right]$ so that $\partial\Phi(\nu)/\partial\nu_i = \mathbb{E}_r[\arg\max_i \nu_i + r_i] = \varphi_i(\nu)$. By definition, the location normalization on reward $\nu$ does not change the arm-selection probability. Therefore, we can assume $\nu_K = 0$ without loss of generality.

## 3.2 Perturbations on the real line: connection to generalized entropy

Lemma 3.1 requires $\mathcal{D}^K$ to have strictly positive density on $\mathbb{R}^K$, which might appear somewhat restrictive. For this reason, Lemma 3.1 was extended to general distributions with absolutely continuous density in Feng et al. [19] and Suggala and Netrapalli [49]. Still, it is known that several useful properties hold when we assume perturbations are fully supported on $\mathbb{R}^K$. For example, Norets and Takahashi [44] showed that $\varphi$ is a bijection between the space of reward vectors $\mathbb{R}^{K-1} \times \{0\}$ and that of arm-selection probabilities $\mathrm{Int}\,(\mathcal{P}_{K-1})$, that is, for any $p \in \mathrm{Int}\,(\mathcal{P}_{K-1})$, there exists a unique $\nu \in \mathbb{R}^{K-1} \times \{0\}$, and vice versa. Furthermore, Fosgerau et al. [21] showed that the associated regularizer can be expressed as a generalized entropy:

$$V(p) = p \cdot \log S(p), \quad \text{where } S : \mathbb{R}^K_{\geq 0} \to \mathbb{R}^K_{\geq 0} \text{ is the inverse function of } \nabla_\nu(e^{\Phi(\nu)}).$$

They also showed that the arm-selection probability $\varphi$ can be expressed by using expected Bregman divergence under some distribution associated with the negative generalized entropy. Investigating unbounded perturbations is therefore natural for two reasons: (i) it reconnects FTPL theory with the perturbations already studied thoroughly, and (ii) it enables the design of analogues to hybrid regularizers in FTRL. The following sections formalize this intuition and shows that a broad class of asymmetric Fréchet-type perturbations is sufficient to achieve BOBW guarantees.

## 4 BOBW guarantee with asymmetric perturbations

We now consider a class of hybrid perturbations whose left and right tails follow different types of distributions, a family of unimodal asymmetric distributions $\mathcal{U}_{\alpha,\beta}$ supported on $\mathbb{R}$, defined by

$$F(x; \mathcal{U}_{\alpha,\beta}) = \begin{cases} \frac{1}{2} + \frac{F(x; \mathcal{D}_\alpha)}{2}, & \text{if } x \geq 0, \\ \frac{1}{2} - \frac{F(-x; \mathcal{D}_\beta)}{2}, & \text{if } x < 0, \end{cases}$$

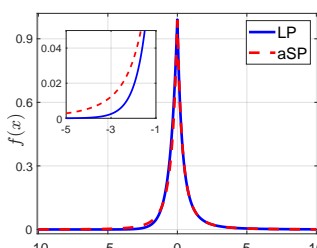

Figure 1: Density examples

where $\mathcal{D}_\alpha \in \mathfrak{D}_\alpha$ and $\mathcal{D}_\beta \in \mathfrak{D}_\beta$ are Fréchet-type distributions over $[0, \infty)$ with tail indices $\alpha > 0$ and $\beta > 0$, respectively. For example, one can consider Laplace-Pareto distribution with density and distribution defined by

$$f_{\mathrm{LP}}(x) = \begin{cases} e^{2x}, & x < 0, \\ \frac{1}{(x+1)^3}, & x \geq 0, \end{cases} \quad \text{and} \quad F_{\mathrm{LP}}(x) = \begin{cases} \frac{e^{2x}}{2}, & x < 0, \\ 1 - \frac{1}{2(x+1)^2}, & x \geq 0. \end{cases} \tag{4}$$

Although, strictly speaking, the Laplace distribution is Gumbel-type distribution, it can be seen as a Fréchet-type distribution with $\alpha = \infty$, that is, $\mathcal{D}_\infty$. Another example is the asymmetric Pareto distribution, which is equivalent to a generalized Pareto distribution with shape 3 on the negative side and a Lomax distribution with shape 2 on the positive side, such that

$$f_{\mathrm{aSP}_{2,3}} = \begin{cases} \frac{(3/2)^4}{(3/2-x)^4}, & x < 0, \\ \frac{1}{(x+1)^3}, & x \geq 0, \end{cases} \quad \text{and} \quad F_{\mathrm{aSP}_{2,3}}(x) = \begin{cases} \frac{27}{16} \frac{1}{(3/2-x)^3}, & x < 0, \\ 1 - \frac{1}{2(x+1)^2}, & x \geq 0. \end{cases} \tag{5}$$

For illustration, we provide the plot of density functions of $f_{\mathrm{LP}}$ and $f_{\mathrm{aSP}_{2,3}}$ in Figure 1.

In the previous BOBW analysis of FTPL, Assumption 2.3, which requires the monotonic decrease of $f/F$, played a crucial role in bounding $-\phi_i'/\phi_i$. However, for $\mathcal{U}_{\alpha,\beta}$, the interval where $f/F$ increases can be unbounded, e.g., asymmetric Pareto distribution in (5), and so we cannot directly apply the previous analysis. To address this issue, we derive a tighter evaluation of $-\phi_i'$ by explicitly accounting for the negative term induced by $-f'(x)$ for $x < 0$, which leads to the following result.

**Theorem 4.1.** *Let $\sigma_i$ be the rank of $\lambda_i$ in the nondecreasing order of $\lambda_1, \ldots, \lambda_K$ (ties are broken arbitrarily). When $\alpha > 1$ and $\beta \geq \alpha + 2$, then FTPL with the unimodal distribution $\mathcal{U}_{\alpha,\beta}$ satisfies for any $\lambda \in \mathbb{R}^K_{\geq 0}$ that*

$$\frac{-\phi_i'(\lambda)}{\phi_i(\lambda)} \leq \mathcal{O}\Big(\sigma_i^{-1/\alpha}\Big) \wedge \mathcal{O}\Big(\frac{1}{\lambda_i - \lambda_{\sigma_1}}\Big).$$

Theorem 4.1 shows that when the left tail index $\beta$ of the distribution (associated with negative perturbations) is lighter than the right tail index $\alpha$ by two, the standard analysis techniques used in Honda et al. [28] and Lee et al. [38] can be largely applied, leading to the following results.

**Corollary 4.2.** *If $\alpha > 1$ and $\beta \geq \alpha + 2$, then FTPL with $\mathcal{U}_{\alpha,\beta}$ and a learning rate of order $K^{\frac{1}{\alpha} - \frac{1}{2}}/\sqrt{t}$ achieves $\mathcal{O}(\sqrt{KT})$ adversarial regret for all $\alpha > 1$. Moreover, if $\alpha = 2$, FTPL achieves a stochastic regret of $\sum_i \mathcal{O}(\log T/\Delta_i)$ provided there is a unique optimal arm.*

Similarly to Theorem 4.1, establishing Corollary 4.2 requires a more refined evaluation to address the unbounded interval where $f'(x) > 0$ in the regret decomposition. We can also derive explicit regret bounds for specific distribution in $\mathcal{U}_{2,\infty}$ by explicitly using the expression of the distributions, whose example is given as follows.

**Theorem 4.3.** *FTPL with learning rate $\eta_t = \frac{m}{\sqrt{t}}$ and the Laplace-Pareto distributions in (4) satisfies*

$$\mathrm{Reg}(T) \leq \left(60m\sqrt{\pi} + \frac{5.7}{m}\right)\sqrt{KT} + \left(\frac{2K}{27} + e^2\right)\log(T+1) + \frac{\sqrt{K\pi}}{2m},$$

*whose dominant term is optimized as $\mathrm{Reg}(T) \leq 49.25\sqrt{KT} + \mathcal{O}(\sqrt{K} + K\log T)$ when $m = 0.23$.*

Using the techniques from Theorem 4.1, which address additional terms due to negative perturbations, we can also obtain logarithmic regret in the stochastic setting with unique optimal arm.

**Theorem 4.4.** *Assume that $i^* = \arg\min_{i \in [K]} \mu_i$ is unique and let $\Delta = \min_{i \neq i^*} \Delta_i$. Then, FTPL with learning rate $\eta_t = m/\sqrt{t}$ for $m > 0$ and Laplace-Pareto perturbation in (4) satisfies*

$$\mathrm{Reg}(T) \leq \sum_{i \neq i^*} \frac{(60m + m^{-1})^2 \log T}{0.035\Delta_i} + \left(\frac{2K}{27} + e^2\right)\log(T+1) + \Theta\left(\frac{(107m + 3/m)^2 K}{\Delta}\right).$$

Although the dominant term in Theorem 4.4 is larger compared to that of FTRL policies, it can be further optimized by using the same arguments as in Honda et al. [28, Remark 12]. Theorems 4.3 and 4.4 highlight the potential of using hybrid perturbations, as Laplace-Pareto perturbations can be roughly viewed as hybrid regularizers combining Shannon and Tsallis entropies. This is because the Laplace distribution belongs to the Gumbel-type, which are associated with the Shannon entropy.

*Remark* 4.5. While we provided concrete results for asymmetric perturbations with different tails, it is also natural to consider hybrid perturbations formed by summing perturbations generated from different distributions. In many such cases, the density may not be available in simple form. Nevertheless, our conditions are sufficiently general to be verified even in such settings.

## 5 FTPL with symmetric perturbations

While asymmetric unbounded perturbations would suffice for constructing hybrid perturbations, it is important to examine whether the asymmetry is truly essential or merely technical artifact for better understanding of FTPL. To this end, we revisit the original motivation for using Fréchet-type, namely, its equivalence to FTRL with Tsallis entropy in the two-armed bandits [33]. Since FTRL with Tsallis entropy is a well-established BOBW policy, this connection would provide a natural starting point for investigating this question, even though such equivalence is established only for $K = 2$.

When $K = 2$, $\mathcal{P}_1$ becomes a line segment, where we can write $p = (x, 1-x)$ for some $x \in (0,1)$. In this case, there exists a unique reward vector $\nu(p) = (c(x), 0)$ for $(c(x), 0) = \varphi^{-1}(p)$ [44]. Accordingly, we can define a regularizer $\bar{V} : \mathbb{R} \to \mathbb{R}$ that coincides with $V(p)$ in terms of $x$ as

$$\bar{V}(x) := V((x, 1-x)) = xc(x) - \Phi((c(x), 0)), \quad \forall x \in (0,1),$$

which satisfies $\bar{V}'(x) = c(x)$. On the other hand, we have $\Pr[c(x) + r_1 \geq r_2] = x$ in two-armed bandits by definition, which implies that $\bar{V}'(x)$ is equivalent to the quantile function of $r_2 - r_1$. When $V_\beta(p) = -\frac{1}{1-\beta}\sum_i p_i^\beta$ denotes $\beta$-Tsallis entropy regularizer for $\beta \in (0,1)$, let $\bar{\mathcal{D}}_\beta^{\mathrm{Ts}}$ denote the distribution of $r_2 - r_1$ and $F(\cdot; \mathcal{D})$ be a CDF of $\mathcal{D}$. Then, we have $F(c(x); \bar{\mathcal{D}}_\beta^{\mathrm{Ts}}) = x$, that is

$$F\left(-\frac{\beta}{1-\beta}\big(x^{\beta-1} - (1-x)^{\beta-1}\big); \bar{\mathcal{D}}_\beta^{\mathrm{Ts}}\right) = x, \tag{6}$$

which shows that $\bar{\mathcal{D}}_\beta^{\mathrm{Ts}}$ is symmetric. By letting $z = c(x)$, we obtain for $\beta \in (0,1)$

$$\lim_{z \to \infty} \frac{zf(z; \bar{\mathcal{D}}_\beta^{\mathrm{Ts}})}{1 - F(z; \bar{\mathcal{D}}_\beta^{\mathrm{Ts}})} = \lim_{x \to 1} \frac{-\frac{\beta}{1-\beta}\big(x^{\beta-1} - (1-x)^{\beta-1}\big) \cdot \frac{1}{\beta(x^{\beta-2} + (1-x)^{\beta-2})}}{1 - x} = \frac{1}{1-\beta}. \tag{7}$$

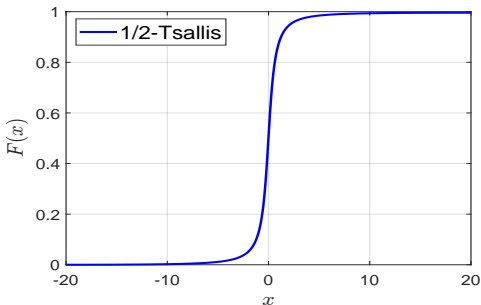
Figure 2: Cumulative sum of IFT.

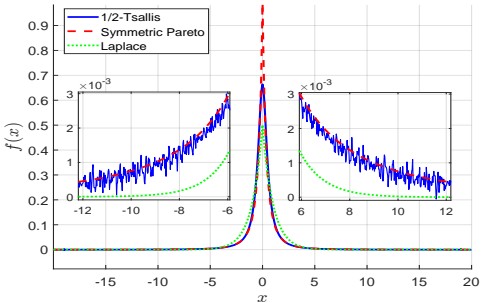
Figure 3: Output of IFT and densities.

Here, (7) is known as von Mises condition for Fréchet-type, which is a sufficient condition for a distribution to be Fréchet-type [24]. This implies that $\bar{\mathcal{D}}_\beta^{\mathrm{Ts}}$, i.e., the distribution of a difference between two random variables, is Fréchet-type with index $(1-\beta)^{-1}$.

While a distribution $(r_1, r_2) \sim \mathcal{D}^2$ satisfying (6) can be easily constructed by dependent perturbations in Feng et al. [19] as explained in Appendix A.3, it remains unclear whether $\bar{\mathcal{D}}_\beta^{\mathrm{Ts}}$ is realizable by two i.i.d. perturbations $r_1, r_2 \sim \mathcal{D}_\beta$. Kim and Tewari [33] simply conjectured that it can be realized by i.i.d. Fréchet-type $\mathcal{D}$. Though it is still difficult to show the existence of i.i.d. perturbation, we can formally show that $\mathcal{D}$, if it exists, is indeed Fréchet-type, as stated below.

**Lemma 5.1.** *Let $-\mathcal{D}$ denote the distribution of $-X$ for $X \sim \mathcal{D}$. Let $(r_1, r_2)$ be i.i.d. from $\mathcal{D}$ and $\bar{\mathcal{D}}$ be distribution of $r_1 - r_2$. Then, if $\bar{\mathcal{D}}$ is Fréchet-type, then either $\mathcal{D}$ or $-\mathcal{D}$ is Fréchet-type.*

The proof of this lemma is given in Appendix G.1, where we analyze the tail behavior of distributions using results from extreme value theory and tail convolutions. Recall that all these observations are made under the assumptions of Lemma 3.1, which implies that $\mathcal{D}_\beta^{\mathrm{Ts}}$ is also fully supported on $\mathbb{R}$ and absolutely continuous if it exists.

## 5.1 Perturbation associated with Tsallis entropy: numerical observation

In two-armed bandits, we can view FTRL with Tsallis entropy as FTPL with some Fréchet-type perturbations. While the existence of i.i.d. perturbation reproducing Tsallis entropy is not formally shown, numerical approximations suggest the following conjecture.

**Conjecture 5.2.** *There exists a symmetric distribution such that the corresponding FTPL reproduces FTRL with $1/2$-Tsallis entropy regularizer.*

To prove Conjecture 5.2, it suffices to consider the characteristic function of distributions. Let $\bar{g}(t)$ be the characteristic function of $\bar{\mathcal{D}}_\beta^{\mathrm{Ts}}$, i.e., $\mathbb{E}[e^{it(r_1-r_2)}]$ and $g$ be that of $\mathcal{D}_\beta^{\mathrm{Ts}}$. By definition of $\bar{\mathcal{D}}$, we have $\bar{g}(t) = g(t)g(-t)$. Since $\bar{\mathcal{D}}_\beta^{\mathrm{Ts}}$ is symmetric by (6), $\bar{g}(t)$ is real and positive function. Therefore, proving Conjecture 5.2 is equivalent to showing that $\sqrt{\bar{g}}$ is a characteristic function of some probability distributions. We expect that this conjecture can be proved by showing that $\sqrt{\bar{g}}$ is positive definite, which is a necessary and sufficient condition, according to Bochner's theorem [18].

**Numerical validation** By definition, $\bar{g}$ can be expressed in terms of the quantile function $c(\cdot)$ as

$$\bar{g}(t) = \int_{-\infty}^{\infty} e^{itx} f(x; \bar{\mathcal{D}}_{1/2}^{\mathrm{Ts}}) \mathrm{d}x = \int_0^1 e^{itc(p)} \mathrm{d}p = \int_0^1 \exp\left(-it(p^{-1/2} - (1-p)^{-1/2})\right) \mathrm{d}p,$$

which can be computed numerically for any $t \in \mathbb{R}$. If $\sqrt{\bar{g}}$ is a valid characteristic function, then applying the inverse Fourier transform (IFT) to $\sqrt{\bar{g}}$ should yield the appropriate density function. Note that $\sqrt{\bar{g}}$ is real-valued regardless of symmetry of $\mathcal{D}_{1/2}^{\mathrm{Ts}}$ as $\bar{g}(t)$ itself is real and positive.

Figure 2 shows that the cumulative sum of the IFT numerically converges to 1, suggesting that $\sqrt{\bar{g}}$ would be a valid characteristic function. In Figure 3, the blue solid line represents the IFT of $\sqrt{\bar{g}}$, while the red dashed and green dotted lines represent the density functions of the symmetric Pareto with shape 2 (Fréchet-type) and standard Laplace (Gumbel-type) distributions, respectively. As shown there, the IFT of $\sqrt{\bar{g}}$ closely resembles the density of a symmetric Pareto distribution with shape parameter 2, though minor fluctuations appear due to numerical approximations. This behavior supports the conjecture that $\mathcal{D}_{1/2}^{\mathrm{Ts}}$ is symmetric and Fréchet-type with index 2. Details on the numerical evaluation and further results including the imaginary part are provided in Appendix F.

## 5.2 Two-armed bandits

The numerical observations indicate that FTRL with $1/2$-Tsallis entropy, a BOBW policy, corresponds to FTPL with a certain unimodal symmetric Fréchet-type distribution with index 2. While Theorem 4.1 and the previous analysis do not extend to the symmetric Fréchet-type perturbations, it is reasonable to expect that FTPL with certain unimodal symmetric Fréchet distributions can achieve BOBW, at least in two-armed bandits. One of the simplest examples of such distributions is the symmetric Pareto distribution, illustrated in Figure 3, where we obtain the following positive result as expected.

**Proposition 5.3.** *Let the perturbations be i.i.d. from the symmetric Pareto distribution with shape* 2 *whose density function and distribution function are respectively defined as*

$$f(x) = \frac{1}{(|x|+1)^3} \quad and \quad F(x) = \begin{cases} \frac{1}{2(1-x)^2}, & x < 0, \\ 1 - \frac{1}{2(x+1)^2}, & x \geq 0. \end{cases}$$

*Then,* $-\phi_i'(\lambda)/\phi_i^{3/2}(\lambda)$ *is uniformly bounded for any* $\lambda \in \mathbb{R}_+^2$ *and* $i \in \{1, 2\}$.

**Corollary 5.4.** *FTPL with symmetric Pareto perturbations with shape* 2 *in Proposition 5.3 achieves* $\mathcal{O}(\sqrt{KT})$ *adversarial regret and* $\mathcal{O}(\log T/\Delta_{i:i\neq i^*})$ *stochastic regret when* $K = 2$.

Note that $f(x)/F(x) = \frac{1}{2(1-x)}$ for $x < 0$, which is increasing on $\mathbb{R}_-$. This observation demonstrates that Assumption 2.3 and asymmetry of the perturbation are not necessary conditions, as far as $K = 2$.

## 5.3 Generalization to multi-armed bandits

While standard Fréchet and Pareto perturbations whose support is $\mathbb{R}_+$ do not correspond to FTRL with Tsallis entropy, they can indeed attain optimal results for general $K$ [28, 38]. Although there is no perturbation associated with Tsallis entropy for $K \geq 4$ [33], one may expect the optimality of FTPL with symmetric Fréchet-type perturbations as achieved by asymmetric (Corollary 4.2) and nonnegative perturbations [38]. One would guess the main difficulty for $K \geq 3$ arises when $\phi_i$ or $\lambda_i$ are not identical to each other, a complexity that does not occur in $K = 2$. Surprisingly, however, we find that the problem happens even when the losses of the currently suboptimal arms are identical, i.e., when $\phi$ lies on the line segment as in $K = 2$.

**Proposition 5.5.** *Let the perturbations be i.i.d. from a unimodal Fréchet-type* $\mathcal{U}_{\alpha_1,\alpha_2}$ *with* $\alpha_j = 2$, *satisfying* $xf(x; \mathcal{D}_{\alpha_j})/(1 - F(x; \mathcal{D}_{\alpha_j})) \leq 2$ *for any* $x > 0$ *and* $j \in \{1, 2\}$. *If* $\lambda \in \mathbb{R}_+^K$ *satisfies* $\lambda = (0, c, \ldots, c)$, *then for* $i \neq 1$ *and* $c \geq 2\sqrt{K}$,

$$\frac{-\phi_i'(\lambda)}{\phi_i^{3/2}(\lambda)} \geq \tilde{\Omega}\left(\frac{c+2}{K\sqrt{K}}\right) \quad and \quad \frac{-\phi_i'(\lambda)}{\phi_i(\lambda)} \geq \tilde{\Omega}\left(\frac{1}{K\sqrt{K}}\right). \tag{8}$$

Although the condition in Proposition 5.5 may seem restrictive, it is known to hold for well-known Fréchet-type distributions such as Fréchet, Pareto and Student-$t$ [38]. The LHS of (8) means that $-\phi'/\phi^{3/2} = \mathcal{O}(1)$ does not hold, which is the key tool to obtain $\mathcal{O}(\sqrt{KT})$ regret with simple learning rate [9, 55]. In addition, while the RHS of (8) does not contradict another key tool $-\phi'/\phi = \mathcal{O}(1)$ for $\mathcal{O}(\sqrt{KT \log K})$ regret [2], it violates the key property, $-\phi'/\phi = \mathcal{O}(1/\lambda_i)$, used for achieving logarithmic regret in the stochastic setting [28, 38]. The proof of Proposition 5.5 and additional results for $K = 3$ with any $\alpha > 1$ and with symmetric Pareto distributions are given in Appendix G. These results suggest that perturbations with equally heavy tails lose some essential properties preserved in FTPL with asymmetric Fréchet-type perturbations and FTRL with Tsallis entropy.

Building on this result, it might be reasonable to expect that the corresponding regularizer for symmetric Pareto distribution behaves differently from $1/2$-Tsallis entropy when $K \geq 3$. However, we obtain similar results even in cases in Proposition 5.5 where the symmetric Pareto fails.

**Proposition 5.6.** *When* $K = 3$ *and* $p = \left(x, \frac{1-x}{2}, \frac{1-x}{2}\right)$ *for* $x \in [1/3, 1)$, *let* $\bar{V}_{\mathrm{sP}}(x)$ *be a corresponding regularizer of the symmetric Pareto distribution with shape* 2 *considered in Proposition 5.3. Then, it holds that*

$$\frac{1}{2\sqrt{1-x}} - 1 \leq \bar{V}_{\mathrm{sP}}'(x) \leq \frac{2\sqrt{2}}{\sqrt{1-x}} - 1,$$

*which implies* $\lim_{x\uparrow 1} \bar{V}_{\mathrm{sP}}'(x)$ *is of the same order up to multiplicative constant with the limit of derivative of* $1/2$-*Tsallis entropy regularizer since* $\bar{V}_{1/2}'(x) = \sqrt{2}/\sqrt{1-x} - 1/\sqrt{x}$.

Proposition 5.6 shows that similar results to the $K = 2$ case can be obtained even when the probability vector lies on the line segment in $\mathcal{P}_2$. Notably, the case considered in Proposition 5.6 is the same as that in Proposition 5.5, which serves as a counterexample where the standard analysis cannot be applied. This suggests that there are limitations to arguments based solely on analogies from the equivalence in two-armed bandits.

*Remark* 5.7. These results do not rule out the possibility of FTPL with symmetric Fréchet-type distributions achieving BOBW guarantees or optimal adversarial regret. Still, the ratio between $\phi'$ and $\phi$ have been believed to be the key for the regret bound. For example, it is conjectured by Abernethy et al. [2, Conjecture 4.5] that FTPL with Gaussian perturbation suffers linear regret because of the unbounded $-\phi'/\phi$. Although this conjecture was later shown to be false, the known regret bound for FTPL with Gaussian perturbation still suffers the regret with additional $\log T$ factor, i.e., $\mathcal{O}(\sqrt{KT \log K} \log T)$ [39]. Our conjecture is in parallel to these observations when we try to achieve the optimal $\mathcal{O}(\sqrt{KT})$ regret, instead of $\mathcal{O}(\sqrt{KT \log K})$ regret. We conjecture that the behavior of $-\phi'/\phi$ of symmetric perturbations may introduce additional factors depending on $K$ or $T$, thereby limiting the achievable regret to near-optimal rates.

# 6    Conclusion

In this paper, we first extended the BOBW guarantee of FTPL to asymmetric perturbations under mild conditions, including a hybrid of Gumbel-type and Fréchet-type perturbations. Given the development of several FTPL policies with hybrid regularizers, a promising direction for future work is designing computationally efficient FTPL counterparts to FTRL policies by incorporating hybrid perturbations in complicated settings. While FTRL with $1/2$-Tsallis entropy regularizer (numerically) corresponds to FTPL with symmetric Fréchet-type perturbations, and FTPL with its tail-equivalent perturbations achieves BOBW guarantee in two-armed bandits, we found that this result does not straightforwardly extend to general case. This finding highlights the limitations of directly extending the relationship observed in two-armed bandits to the general case, emphasizing the need for further investigation to better understand the behavior of FTPL in broader settings.

## Acknowledgments and Disclosure of Funding

JL was supported by the National Research Foundation of Korea (NRF) grant funded by the Korea government (MSIT) (No. RS-2024-00395303), by AI-Bio Research Grant through Seoul National University, and this work was also supported by the grant Nos. 2023-00262155; 2024-00460980; and 2025-02304717 (IITP) funded by the Korea government (the Ministry of Science and ICT). JH was supported by JST/CREST Innovative Measurement and Analysis (Grant Number JPMJCR2333). SI was supported by JSPS KAKENHI Grant Number JP25K03184. MO was supported by the National Research Foundation of Korea (NRF) grant funded by the Korea government (MSIT) (No. RS-2022-NR071853 and RS-2023-00222663).

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

# Appendix Contents

# A  Additional details omitted in main paper

Here, we provide the details omitted in the main paper due to space constraints for completeness.

**Notation**  Throughout the appendix, we define the gap of each vector element from its minimum using underlines, i.e., $\underline{\lambda} = \lambda - \mathbf{1} \min_{i \in [K]} \lambda_i$. Therefore, the arm-selection probability based on the gap of the cumulative loss can be interchanged with that based on the cumulative rewards since $\underline{\lambda}_i = \max_j \nu_j - \nu_i$ holds. When $\mathcal{D}$ is absolutely continuous, $\phi(\cdot; \mathcal{D})$ can be written as

$$\phi_i(\lambda; \mathcal{D}) = \Pr_{r_1, \dots, r_K \sim \mathcal{D}} \left[ i = \arg\min_{j \in [K]} \{\lambda_j - r_j\} \right] = \int_{\mathbb{R}} \prod_{j \neq i} F(z + \lambda_j) \, \mathrm{d}F(z + \lambda_i).$$

Also, we denote the geometric resampling estimator of $w_{t,i}^{-1}$ by $\widehat{w_{t,i}^{-1}}$.

**Definition of Fréchet-type perturbation**  Here, we readily follow the notation used in Lee et al. [38]. To define the Fréchet-type perturbation, we first require the notion of regular variation defined as below.

**Definition A.1** (Regular variation [24]).  An eventually positive function $g$, which becomes positive after a certain point, is called regularly varying at infinity with index $\alpha$, $g \in \mathrm{RV}_\alpha$ if

$$\lim_{x \to \infty} \frac{g(tx)}{g(x)} = t^\alpha, \; \forall t > 0.$$

If $g(x)$ is regularly varying with index 0, then $g$ is called slowly varying.

Then, a necessary and sufficient condition for a distribution to be Fréchet-type is expressed in terms of regular variation as below.

**Proposition A.2.**  *A distribution $\mathcal{D}$ is Fréchet-type with index $\alpha > 0$ iff its right endpoint is infinite and the tail function, $1 - F$, is regularly varying at infinity with index $-\alpha$, i.e., $1 - F \in \mathrm{RV}_{-\alpha}$. In this case,*

$$F^n(a_n x) \to \begin{cases} \exp(-x^{-\alpha}), & x \geq 0, \\ 0, & x < 0, \end{cases} \quad n \to \infty,$$

*where $a_n = \inf \{x : F(x) \geq 1 - 1/n\}$.*

Moreover, if $\mathcal{D}$ is Fréchet-type, we can express the tail distribution with a slowly varying function $S_F \in \mathrm{RV}_0$ as

$$1 - F(x) = x^{-\alpha} S_F(x), \quad \forall x > 0.$$

Note that by definition, a slowly varying function $S_F(x)$ grows at most polylogarithmically. For further details on Fréchet-type distributions, see Haan and Ferreira [24] and Resnick [47].

## A.1  Previous assumptions for BOBW guarantee

In the previous analysis, Lee et al. [38] showed the optimality of FTPL with perturbations satisfying following assumptions.

**Assumption 2.2**  $\mathrm{supp}\mathcal{D}_\alpha \subseteq [x_{\min}, \infty)$ *for some $x_{\min} \geq 0$ and the hazard function $\frac{f(x)}{1 - F(x)}$ is bounded.*

**Assumption 2.3**  $\frac{f(x)}{F(x)}$ *is monotonically decreasing in $x \geq x_{\min}$.*

**Assumption A.3.**  $F(x)$ *has a density function $f(x)$ that is decreasing in $x \geq x_0$ for some $x_0 > x_{\min}$.*

**Assumption A.4.**  There exist positive constants $M_u = M_u(\mathcal{D}_\alpha)$ and $M_l = M_l(\mathcal{D}_\alpha)$ satisfying

$$\mathbb{E}_{X_1, \dots, X_k \sim \mathcal{D}_\alpha} \left[ \max_{i \in [k]} X_i / a_k \right] \leq M_u$$

$$\mathbb{E}_{X_1, \dots, X_k \sim \mathcal{D}_\alpha} \left[ \frac{1}{\max_{i \in [k]} X_i / a_k} \right] \leq M_l$$

for $a_k = \inf \{x : F(x) \geq 1 - 1/k\}$ and it satisfies $A_l k^{\frac{1}{\alpha}} \leq a_k \leq A_u k^{\frac{1}{\alpha}}$ for some positive constants $A_l, A_u$.

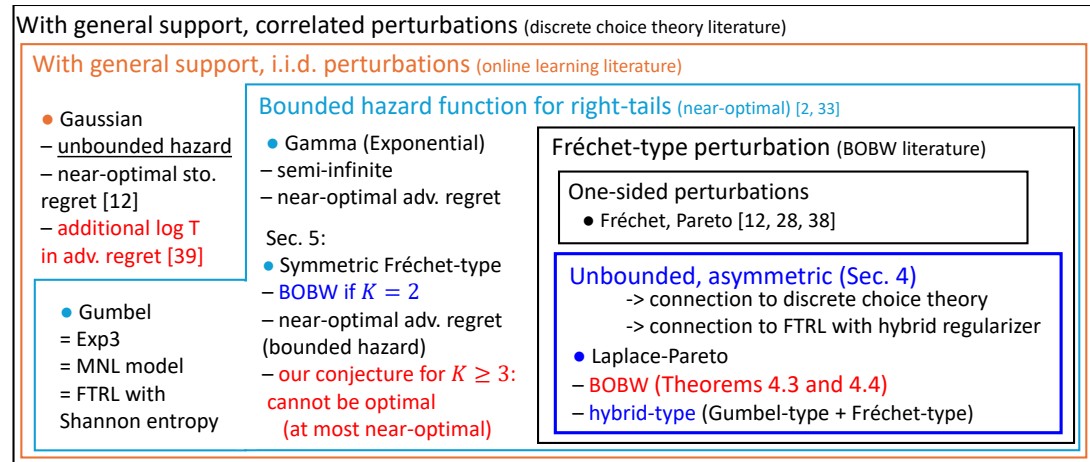

Figure 4: Perturbation landscape summarizing distributions studied in the literature.

**Assumption A.5.** $\lim_{x\to\infty} \frac{-xf'(x)}{f(x)} = \alpha + 1$ and $\frac{-f'(x)}{f(x)}$ is bounded almost everywhere on $[x_{\min}, \infty)$.

To simplify the analysis and avoid technical complications related to the boundedness of the hazard function, this paper focuses on a tail-equivalent distribution satisfying the assumption by considering the truncated version $F^*$ of $F$ when we consider the general distributions. Specifically, we define the truncated distribution function as

$$F^*(x) = \Pr[X \geq 1 + x | X > 1] = \frac{F(x+1) - F(1)}{1 - F(1)}, \quad x > 0,$$

which is supported on $\mathbb{R}_+$. This truncation technique was also used to provide $\mathcal{O}(\sqrt{KT \log K})$ regret [2]. Then, for any $\mathcal{D}_\alpha \in \mathfrak{D}_\alpha$, its distribution function can be written as

$$1 - F^*(x; \mathcal{D}_\alpha) = \tilde{\Theta}((x+1)^{-\alpha}), \quad x > 0,$$

For simplicity, we denote $F^*$ by $F$ since we only consider truncated one in this paper.

Note that by Corollary 4.2, Assumption 2.3 can be relaxed to the existence of $x_0 > x_{\min}$ such that $f/F$ is decreasing in $x \geq x_0$ unless the left tail is lighter than the right tail.

For a quick overview of the position of this paper, see Figure 4.

## A.2 Derivation of potential function

For any i.i.d. perturbations under assumptions in Lemma 3.1, we can express the potential function as

$$\Phi(\nu) = \sum_{i \in [K]} \int_{-\infty}^{\infty} z f(z - \nu_i) \prod_{j \neq i} F(z - \nu_j) \mathrm{d}z, \tag{9}$$

By definition of potential function $\Phi$ and its convex conjugate $V$, it holds $\Phi(\nu) + V(p) = \sum_{i \in [K]} p_i \nu_i$ iff $p \in \partial\Phi(\nu)$. Chiong et al. [13] showed that

$$V(p) = -\sum_i p_i \mathbb{E}_{r \sim \mathcal{D}_\alpha} \left[ r_i \middle| i = \arg\max_{j \in [K]} (\nu_j + r_j) \right].$$

Let us assume $\nu$ is of decreasing order, i.e., $\nu_1 \geq \nu_2 \geq \cdots \geq \nu_K$ holds without loss of generality, where $\underline{\lambda}_i = \nu_1 - \nu_i$ holds. By definition, when we consider the perturbation distributions supported

on $\mathbb{R}$, we obtain

$$
\begin{aligned}
\Phi(\nu) &= \int_{-\infty}^{\infty} \mathbb{E}\Big[\max_i (r_i + \nu_i)\Big| r_1 = x\Big] f(x)\mathrm{d}x \\
&= \int_{-\infty}^{\infty} (x + \nu_1) f(x) \prod_{j \neq 1} F(x + \nu_1 - \nu_j)\mathrm{d}x \\
&\quad + \int_{-\infty}^{\infty} f(x) \sum_{i \neq 1} \int_{x + \nu_1 - \nu_i}^{\infty} (y + \nu_i) f(y) \prod_{j \neq 1, i} F(y + \nu_i - \nu_j)\mathrm{d}y\mathrm{d}x \\
&= \int_{-\infty}^{\infty} x f(x - \nu_1) \prod_{j \neq 1} F(x - \nu_j)\mathrm{d}x \\
&\quad + \int_{-\infty}^{\infty} f(x) \sum_{i \neq 1} \int_{x + \nu_1}^{\infty} y f(y - \nu_i) \prod_{j \neq 1, i} F(y - \nu_j)\mathrm{d}y\mathrm{d}x \\
&= \int_{-\infty}^{\infty} x f(x - \nu_1) \prod_{j \neq 1} F(x - \nu_j)\mathrm{d}x \\
&\quad + \int_{-\infty}^{\infty} f(x - \nu_1) \sum_{i \neq 1} \int_{x}^{\infty} y f(y - \nu_i) \prod_{j \neq 1, i} F(y - \nu_j)\mathrm{d}y\mathrm{d}x.
\end{aligned}
$$

By changing the order of integral, we obtain that

$$
\begin{aligned}
\int_{-\infty}^{\infty} f(x - \nu_1) &\sum_{i \neq 1} \int_{x}^{\infty} y f(y - \nu_i) \prod_{j \neq 1, i} F(y - \nu_j)\mathrm{d}y\mathrm{d}x \\
&= \int_{-\infty}^{\infty} \sum_{i \neq 1} y f(y - \nu_i) \prod_{j \neq 1, i} F(y - \nu_j) \int_{-\infty}^{y} f(x - \nu_1)\mathrm{d}x\mathrm{d}y \\
&= \sum_{i \neq 1} \int_{-\infty}^{\infty} y f(y - \nu_i) \prod_{j \neq i} F(y - \nu_j)\mathrm{d}y.
\end{aligned}
$$

Therefore,

$$
\Phi(\nu) = \sum_{i \in [K]} \int_{-\infty}^{\infty} z f(z - \nu_i) \prod_{j \neq i} F(z - \nu_j)\mathrm{d}z.
$$

Since $p$ and $\nu$ are bijective, as discussed in Section 3.2, we have

$$
V(p) = \sum_{i \in [K]} p_i \nu_i(p) - \Phi(\nu(p)), \quad \forall p \in \mathrm{Int}\,(\mathcal{P}_{K-1}). \tag{10}
$$

Therefore, by (10) and letting $w = \phi(\underline{\lambda}) = \varphi(\nu)$, it holds that

$$
\begin{aligned}
-V(w) &= \sum_i w_i \left( (\nu_1 - \nu_i) + \sum_{j \in [K]} \int_{-\infty}^{\infty} z f(z + \underline{\lambda}_j) \prod_{l \neq j} F(z + \underline{\lambda}_j) \right) \mathrm{d}z \\
&= \sum_i w_i \left( \underline{\lambda}_i + \sum_{j \in [K]} \int_{-\infty}^{\infty} z f(z + \underline{\lambda}_j) \prod_{l \neq j} F(z + \underline{\lambda}_j)\mathrm{d}z \right).
\end{aligned}
$$

Note that the above potential function and corresponding regularization function can be obtained for any perturbations with density $f$ and distribution $F$ supported on $\mathbb{R}$.

Here, one can check that

$$\frac{\partial \Phi(\nu)}{\partial \nu_i} = \int_{-\infty}^{\infty} -zf'(z-\nu_i) \prod_{j\neq i} F(z-\nu_j) \mathrm{d}z$$

$$+ \int_{-\infty}^{\infty} - \sum_{j\neq i} zf(z-\nu_j)f(z-\nu_i) \prod_{l\neq i,j} F(z-\nu_j) \mathrm{d}z$$

$$= \int_{-\infty}^{\infty} f(z-\nu_i) \prod_{j\neq i} F(z-\nu_j) \mathrm{d}z = \varphi_i(\nu), \tag{11}$$

where (11) holds by partial integral. To be precise, we have

$$\int_{-\infty}^{\infty} -zf'(z-\nu_i) \prod_{j\neq i} F(z-\nu_j) \mathrm{d}z$$

$$= -zf(z-\nu_i) \prod_{j\neq i} F(z-\nu_j) \Big|_{z=-\infty}^{z=\infty}$$

$$+ \int_{-\infty}^{\infty} f(z-\nu_i) \prod_{j\neq i} F(z-\nu_j) + \sum_{j\neq i} zf(z-\nu_i) \prod_{l\neq i,j} F(z-\nu_l) \mathrm{d}z$$

$$= \int_{-\infty}^{\infty} f(z-\nu_i) \prod_{j\neq i} F(z-\nu_j) + \sum_{j\neq i} zf(z-\nu_i) \prod_{l\neq i,j} F(z-\nu_l) \mathrm{d}z.$$

### A.3 Equivalence results in two-armed bandits: the existence of correlated perturbations

In discrete choice theory, as the assumptions in Lemma 3.1 implies, the additive random utility model (ARUM) admits correlated perturbations, whereas FTPL in bandit problems typically assumes i.i.d. perturbations. The equivalence between ARUM and FTRL in two-dimensional case was established as follows.

**Lemma A.6** (Theorem 4 in Feng et al. [19]). *For any differentiable choice welfare function $\mathcal{C}(\nu_1, \nu_2)$, there exists a distribution $\mathcal{D}$ of $\{r_1, r_2\}$ such that*

$$\mathcal{C}(\nu_1, \nu_2) = \mathbb{E}_{(r_1,r_2)\sim\mathcal{D}}[\max\{\nu_1 + r_1, \nu_2 + r_2\}].$$

While the notation of choice welfare function in Feng et al. [19] is a general notion to define discrete choice model, it suffices to interpret it as a type of potential function $\Phi$ in this section, such that the choice model (arm-selection probability) is given by $\varphi(\nu) = \nabla\Phi(\nu)$.

This lemma shows the equivalence between differentiable choice functions and ARUM. Furthermore, Theorem 2 in Feng et al. [19] shows the equivalence between differentiable choice function and FTRL with (essentially) strictly convex regularizer in general case, implying that for any FTRL with strictly convex regularizer $V$, there exists a differentiable choice function such that $p(\nu; V) = \nabla\mathcal{C}(\nu)$, and vice versa. Since $\beta$-Tsallis entropy regularizer is strictly convex function, this implies that there exists a corresponding ARUM. It is important to note that this result does not necessarily imply the independence of $r_1$ and $r_2$, as suggested by Lemma A.6. Specifically, one can define a distribution $\mathcal{D}$ of $(r_1, r_2)$ as

$$(r_1, r_2) = (\mathcal{C}(0,0) - \max\{\xi, 0\}, \mathcal{C}(0,0) - \max\{-\xi, 0\}),$$

where $\xi$ is a random variable with distribution function $F_\xi(x) = c(x)$. In this case, whenever $r_1$ is observed, the value of $r_2$ is determined by definition of $\mathcal{D}$, which shows dependency. While this example illustrates how strictly convex regularizers can be associated with correlated perturbations, we expect that the $\beta$-Tsallis entropy regularizer corresponds to i.i.d. perturbations, as such constructed examples appear artificial and were introduced to address a broad class of (essentially) strictly convex regularizers.

## B  Proof of Theorem 4.1: asymmetric perturbations

Here, we assume $\lambda_1 \leq \ldots \leq \lambda_K$ without loss of generality, where $\sigma_i = i$ holds.

By definition, for any $i \in [K]$,

$$
\frac{-\phi_i'(\underline{\lambda})}{\phi_i(\underline{\lambda})} = \frac{\int_{-\infty}^{\infty} -f'(z + \underline{\lambda}_i) \prod_{j \neq i} F(z + \underline{\lambda}_j) \mathrm{d}z}{\int_{-\infty}^{\infty} f(z + \underline{\lambda}_i) \prod_{j \neq i} F(z + \underline{\lambda}_j) \mathrm{d}z}
$$

$$
\leq \frac{\int_{-\underline{\lambda}_i}^{\infty} -f'(z + \underline{\lambda}_i) \prod_{j \neq i} F(z + \underline{\lambda}_j) \mathrm{d}z}{\int_{-\infty}^{\infty} f(z + \underline{\lambda}_i) \prod_{j \neq i} F(z + \underline{\lambda}_j) \mathrm{d}z}. \qquad \text{(unimodality of } f)
$$

Then, we first show the existence of constant $\zeta > 0$ satisfying

$$
\frac{\int_{-\underline{\lambda}_i}^{0} -f'(z + \underline{\lambda}_i) \prod_{j \neq i} F(z + \underline{\lambda}_j) \mathrm{d}z}{\int_{0}^{\infty} -f'(z + \underline{\lambda}_i) \prod_{j \neq i} F(z + \underline{\lambda}_j) \mathrm{d}z} \leq \zeta \qquad (12)
$$

for any $\underline{\lambda}$. When $i = 1$, it holds trivially. Let us consider $i \neq 1$. Define $G(z) = \prod_{j \neq 1, i} F(z + \underline{\lambda}_j)$, which is increasing with respect to $z$. Then,

$$
\frac{\int_{-\underline{\lambda}_i}^{0} -f'(z + \underline{\lambda}_i) \prod_{j \neq i} F(z + \underline{\lambda}_j) \mathrm{d}z}{\int_{0}^{\infty} -f'(z + \underline{\lambda}_i) \prod_{j \neq i} F(z + \underline{\lambda}_j) \mathrm{d}z} = \frac{\int_{-\underline{\lambda}_i}^{0} -f'(z + \underline{\lambda}_i) F(z) G(z) \mathrm{d}z}{\int_{0}^{\infty} -f'(z + \underline{\lambda}_i) F(z) G(z) \mathrm{d}z}
$$

$$
\leq \frac{\int_{-\underline{\lambda}_i}^{0} -f'(z + \underline{\lambda}_i) F(z) G(0) \mathrm{d}z}{\int_{0}^{\infty} -f'(z + \underline{\lambda}_i) F(z) G(0) \mathrm{d}z}
$$

$$
= \frac{\int_{-\underline{\lambda}_i}^{0} -f'(z + \underline{\lambda}_i) F(z) \mathrm{d}z}{\int_{0}^{\infty} -f'(z + \underline{\lambda}_i) F(z) \mathrm{d}z}.
$$

Since $F(z) \geq 1/2$ and $-f'(z + \underline{\lambda}_i) \geq 0$ hold on $z \geq 0$, we have

$$
\int_{0}^{\infty} -f'(z + \underline{\lambda}_i) F(z) \mathrm{d}z \geq \frac{1}{2} \int_{0}^{\infty} -f'(z + \underline{\lambda}_i) \mathrm{d}z
$$

$$
= \frac{f(\underline{\lambda}_i)}{2}.
$$

Hence,

$$
\frac{\int_{-\underline{\lambda}_i}^{0} -f'(z + \underline{\lambda}_i) F(z) \mathrm{d}z}{\int_{0}^{\infty} -f'(z + \underline{\lambda}_i) F(z) \mathrm{d}z} \leq \frac{2}{f(\underline{\lambda}_i)} \int_{-\underline{\lambda}_i}^{0} -f'(z + \underline{\lambda}_i) F(z) \mathrm{d}z.
$$

By Assumption A.5, it holds that for $z \geq 0$

$$
-f'(z) = \Theta\left( \frac{1}{(z + 1)^{\alpha + 2}} \right).
$$

Also, by definition of $F$ of $\mathcal{U}_{\alpha, \beta}$, we have

$$
F(z; \mathcal{U}_{\alpha, \beta}) = \frac{1 - F(-z; \mathcal{D}_\beta)}{2} = \Theta\left( \frac{1}{(1 - z)^\beta} \right), \forall z < 0.
$$

Here,

$$
\max_{z \in [-\underline{\lambda}_i, 0]} \frac{1}{(z + \underline{\lambda}_i + 1)^{\alpha + 2}} \frac{1}{(1 - z)^\beta} = \frac{1}{(\underline{\lambda}_i + 1)^{\alpha + 2}} \vee \frac{1}{(\underline{\lambda}_i + 1)^\beta}.
$$

Therefore, whenever $\beta \geq \alpha + 2$,

$$
\frac{2}{f(\underline{\lambda}_i)} \int_{-\underline{\lambda}_i}^{0} -f'(z + \underline{\lambda}_i) F(z) \mathrm{d}z = \frac{2}{f(\underline{\lambda}_i)} \Theta\left( \int_{-\underline{\lambda}_i}^{0} \frac{1}{(z + \underline{\lambda}_i + 1)^{\alpha + 2} (1 - z)^\beta} \mathrm{d}z \right)
$$

$$
\leq \frac{2}{f(\underline{\lambda}_i)} \Theta\left( \frac{\underline{\lambda}_i}{(1 + \underline{\lambda}_i)^{\alpha + 2}} \right)
$$

$$
= \Theta\left( \frac{\underline{\lambda}_i (\underline{\lambda}_i + 1)^{\alpha + 1}}{(1 + \underline{\lambda}_i)^{\alpha + 2}} \right) = \Theta(1).
$$

Therefore, by applying (12), we obtain

$$
\begin{aligned}
\frac{-\phi_i'(\lambda)}{\phi_i(\lambda)} &\le (\zeta+1)\frac{\int_0^\infty -f'(z+\underline{\lambda}_i)\prod_{j\ne i}F(z+\underline{\lambda}_j)\mathrm{d}z}{\int_{-\infty}^\infty f(z+\underline{\lambda}_i)\prod_{j\ne i}F(z+\underline{\lambda}_j)\mathrm{d}z}\\
&\le (\zeta+1)\frac{\int_0^\infty -f'(z+\underline{\lambda}_i)\prod_{j\ne i}F(z+\underline{\lambda}_j)\mathrm{d}z}{\int_0^\infty f(z+\underline{\lambda}_i)\prod_{j\ne i}F(z+\underline{\lambda}_j)\mathrm{d}z}.
\end{aligned}
$$

Since $\mathcal{U}_{\alpha,\beta}$ is equivalent to $\mathcal{D}_\alpha \in \mathfrak{D}_\alpha$ on $z\ge 0$, which satisfies all assumptions in Appendix A.1 (with $x_{\min}=0$), Lemmas 9 and 10 in Lee et al. [38] concludes the proof.

## C  Proof of Theorem 4.3: Laplace-Pareto perturbation

The proof given in this section readily follows those given by Honda et al. [28] and Lee et al. [38], with the main distinction being terms related to the negative perturbations. For clarity, we omit notation related to LP, as we focus solely on the Laplace-Pareto distribution defined in (4).

We begin by decomposing the regret. Using Lemma 7 from Lee et al. [38], the regret can be decomposed as follows.

$$
\begin{aligned}
\mathrm{Reg}(T) &\le \sum_{t=1}^T \mathbb{E}\Big[\big\langle \hat{\ell}_t, w_t-w_{t+1}\big\rangle\Big] + \sum_{t=1}^T\left(\frac{1}{\eta_{t+1}}-\frac{1}{\eta_t}\right)\mathbb{E}[r_{t+1,I_{t+1}}-r_{t+1,i^*}] + \frac{\mathbb{E}_{r_1\sim \mathrm{LP}}[\max_{i\in[K]}r_{1,i}]}{\eta_1}\\
&\le \sum_{t=1}^T \mathbb{E}\Big[\big\langle \hat{\ell}_t, w_t-w_{t+1}\big\rangle\Big] + \sum_{t=1}^T\left(\frac{1}{\eta_{t+1}}-\frac{1}{\eta_t}\right)\mathbb{E}[r_{t+1,I_{t+1}}-r_{t+1,i^*}] + \frac{\mathbb{E}_{r_1\sim \mathrm{P}_2}[\max_{i\in[K]}r_{1,i}]}{\eta_1}\\
&\le \underbrace{\sum_{t=1}^T \mathbb{E}\Big[\big\langle \hat{\ell}_t, w_t-w_{t+1}\big\rangle\Big]}_{\text{stability}} + \underbrace{\sum_{t=1}^T\left(\frac{1}{\eta_{t+1}}-\frac{1}{\eta_t}\right)\mathbb{E}[r_{t+1,I_{t+1}}-r_{t+1,i^*}]}_{\text{penalty}} + \frac{\sqrt{K\pi}}{2\eta_1}, \qquad (13)
\end{aligned}
$$

where the second inequality follows that the block maxima of LP can be upper bounded by that of the Pareto distribution with shape 2, $\mathrm{P}_2$.

Then, it remains to bound two terms, stability term and penalty term. For the stability term, we can further decompose it into two terms.

**Lemma C.1.** *It holds that*

$$
\sum_{t=1}^T\mathbb{E}\Big[\big\langle \hat{\ell}_t, w_t-w_{t+1}\big\rangle\Big] \le \sum_{t=1}^T\mathbb{E}\Big[\big\langle \hat{\ell}_t, \phi(\eta_t\hat{L}_t)-\phi(\eta_t(\hat{L}_t+\hat{\ell}_t))\big\rangle\Big] + \left(\frac{4K}{27}+2e^2\right)\log\left(\frac{\eta_1}{\eta_{T+1}}\right).
$$

To prove this lemma, we must address certain terms that were ignored in previous approaches, which considered only positive perturbations. This is necessary because, due to symmetry, there exists a semi-infinite interval where $f'>0$, resulting in a loose upper bound.

Then, the first term can be bounded as follows.

**Lemma C.2.** *For any $i\in[K]$, if $\hat{L}_{t,i}$ is the $\sigma_i$-th smallest among $\big\{\hat{L}_{t,j}\big\}$, then*

$$
\mathbb{E}\Big[\hat{\ell}_{t,i}(\phi_i(\eta_t\hat{L}_t)-\phi_i(\eta_t(\hat{L}_t+\hat{\ell}_t)))\Big|\hat{L}_t\Big] \le \frac{30\sqrt{\pi}}{\sqrt{\sigma_i}}\eta_t \wedge \frac{10e\sqrt{2}}{\underline{\hat{L}}_{t,i}}.
$$

In the proof of Lemma C.2, we apply the same techniques as in the proof of Theorem 4.1, which ensures that the terms $-\frac{\phi_i'}{\phi_i}$ are of the desired order.

Finally, the penalty term can be bounded as follows.

**Lemma C.3.** *It holds that*

$$
\mathbb{E}\Big[r_{t,I_t}-r_{t,i^*}\Big|\hat{L}_t\Big] \le 5.7\sqrt{K}\wedge\sum_{i\ne i^*}\frac{1}{\eta_t\underline{\hat{L}}_{t,i}}.
$$

By applying Lemmas C.1–C.3 into (13) with $\eta_t = \frac{m}{\sqrt{t}}$, we obtain

$$\text{Reg}(T)$$

$$\leq \sum_{t=1}^{T}\sum_{i=1}^{K} \eta_t \frac{30\sqrt{\pi}}{\sqrt{i}} + \sum_{t=1}^{T}\left(\frac{1}{\eta_{t+1}} - \frac{1}{\eta_t}\right)5.7\sqrt{K} + \left(\frac{4K}{27} + 2e^2\right)\log\left(\frac{\eta_1}{\eta_{T+1}}\right) + \frac{\sqrt{K}\pi}{2\eta_1}$$

$$\leq \sum_{t=1}^{T} 30\eta_t\sqrt{\pi}(1 + 2(\sqrt{K} - 1) + \sum_{t=1}^{T}\left(\frac{1}{\eta_{t+1}} - \frac{1}{\eta_t}\right) + 5.7\sqrt{K}$$

$$+ \left(\frac{4K}{27} + 2e^2\right)\log\left(\frac{\eta_1}{\eta_{T+1}}\right) + \frac{\sqrt{K}\pi}{2\eta_1}$$

$$= \sum_{t=1}^{T} \frac{60m\sqrt{K}\pi}{\sqrt{t}} + \frac{5.7\sqrt{K}}{m}\sum_{t=1}^{T}\left(\sqrt{t+1} - \sqrt{t}\right) + \left(\frac{2K}{27} + e^2\right)\log(T+1) + \frac{\sqrt{K}\pi}{2m}$$

$$\leq \left(120m\sqrt{K}\pi T + \frac{5.7(\sqrt{T+1} - 1)}{m}\right)\sqrt{K} + \left(\frac{2K}{27} + e^2\right)\log(T+1) + \frac{\sqrt{K}\pi}{2m}$$

$$\leq \left(120m\sqrt{\pi} + \frac{5.7}{m}\right)\sqrt{KT} + \left(\frac{2K}{27} + e^2\right)\log(T+1) + \frac{\sqrt{K}\pi}{2m}.$$

## C.1 Proof of Lemma C.1

For generic $L \in \mathbb{R}^K$, define $\underline{L} = L - \mathbf{1}\min_i L_i$. Then, by definition of $\phi$, we have

$$\frac{\partial}{\partial\eta}\phi_i(\eta L) = \int_{-\infty}^{\infty} \underline{L}_i f'(z + \eta\underline{L}_i)\prod_{j\neq i} F(z + \eta\underline{L}_j)\mathrm{d}z$$

$$+ \int_{-\infty}^{\infty} f(z + \eta\underline{L}_i)\sum_{j\neq i}\underline{L}_j f(z + \eta\underline{L}_j)\left(\prod_{l\neq i,j} F(z + \eta\underline{L}_l)\right)\mathrm{d}z. \quad (14)$$

By definition of $f$, one can see that $f'(x) > 0$ for $x < 0$ and $f'(x) < 0$ for $x > 0$. Therefore, we have

$$\int_{-\infty}^{\infty} \underline{L}_i f'(z + \eta\underline{L}_i)\prod_{j\neq i} F(z + \eta\underline{L}_j)\mathrm{d}z$$

$$\leq \int_{-\infty}^{0} \underline{L}_i f'(z + \eta\underline{L}_i)\prod_{j\neq i} F(z + \eta\underline{L}_j)\mathrm{d}z$$

$$= \underline{L}_i f(z + \eta\underline{L}_i)\prod_{j\neq i} F(z + \eta\underline{L}_j)\bigg|_{z=-\infty}^{z=0}$$

$$- \int_{-\infty}^{0} \underline{L}_i f(z + \eta\underline{L}_i)\sum_{j\neq i} f(z + \eta\underline{L}_j)\left(\prod_{l\neq i,j} F(z + \eta\underline{L}_l)\right)\mathrm{d}z$$

$$= \underline{L}_i f(\eta\underline{L}_i)\prod_{j\neq i} F(\eta\underline{L}_j) - \int_{-\infty}^{0} \underline{L}_i f(z + \eta\underline{L}_i)\sum_{j\neq i} f(z + \eta\underline{L}_j)\left(\prod_{l\neq i,j} F(z + \eta\underline{L}_l)\right)\mathrm{d}z$$

$$\leq \frac{\underline{L}_i}{(\eta\underline{L}_i + 1)^3} - \int_{-\infty}^{0} \underline{L}_i f(z + \eta\underline{L}_i)\sum_{j\neq i} f(z + \eta\underline{L}_j)\left(\prod_{l\neq i,j} F(z + \eta\underline{L}_l)\right)\mathrm{d}z$$

$$\leq \frac{4}{27\eta} - \int_{-\infty}^{0} \underline{L}_i f(z + \eta\underline{L}_i)\sum_{j\neq i} f(z + \eta\underline{L}_j)\left(\prod_{l\neq i,j} F(z + \eta\underline{L}_l)\right)\mathrm{d}z. \quad (15)$$

By injecting the result of (15) into (14), we obtain for any $i \in [K]$

$$\frac{\partial}{\partial \eta} \phi_i(\eta L) \leq \frac{4}{27\eta} + \underbrace{\int_0^\infty f(z + \eta \underline{L}_i) \sum_{j \neq i} \underline{L}_j f(z + \eta \underline{L}_j) \left( \prod_{l \neq i,j} F(z + \eta \underline{L}_l) \right) dz}_{(\dagger_i)}$$

$$+ \underbrace{\int_{-\infty}^0 f(z + \eta \underline{L}_i) \sum_{j \neq i} (\underline{L}_j - \underline{L}_i) f(z + \eta \underline{L}_j) \left( \prod_{l \neq i,j} F(z + \eta \underline{L}_l) \right) dz}_{(\ddagger_i)}. \quad (16)$$

Similarly to the proof of Lemma 8 in Lee et al. [38], we obtain for $z \geq 0$ that

$$\prod_{l \neq i,j} F(z + \eta \underline{L}_l) = \prod_{l \neq i,j} (1 - (1 - F(z + \eta \underline{L}_l)))$$

$$\leq \exp\left( - \sum_{l \neq i,j} (1 - F(z + \eta \underline{L}_l)) \right)$$

$$\leq e^2 \exp\left( - \sum_{l \in [K]} (1 - F(z + \eta \underline{L}_l)) \right) = e^2 \exp\left( - \sum_{l \in [K]} \frac{1}{2(z + \eta \underline{L}_l)^2} \right),$$

where the last inequality follows from $F(x) \in [0, 1]$ for all $x \in \mathbb{R}$, i.e., $e^{1-F(x)} \leq e$ and the last equality follows from the definition of $F(x)$ for $x \geq 0$. Therefore, we have

$$(\dagger_i) \leq e^2 \int_0^\infty f(z + \eta \underline{L}_i) \left( \sum_{j \neq i} \underline{L}_j f(z + \eta \underline{L}_j) \right) \cdot \exp\left( - \sum_{l \in [K]} \frac{1}{2(z + \eta \underline{L}_l)^2} \right) dz$$

$$\leq e^2 \int_0^\infty f(z + \eta \underline{L}_i) \left( \sum_{j \in [K]} \underline{L}_j f(z + \eta \underline{L}_j) \right) \cdot \exp\left( - \sum_{l \in [K]} \frac{1}{2(z + \eta \underline{L}_l)^2} \right) dz$$

$$= e^2 \int_0^\infty \frac{1}{(z + \eta \underline{L}_i)^3} \left( \sum_{j \in [K]} \frac{\underline{L}_j}{(z + \eta \underline{L}_j)^3} \right) \cdot \exp\left( - \sum_{l \in [K]} \frac{1}{2(z + \eta \underline{L}_l)^2} \right) dz$$

$$\leq e^2 \int_0^\infty \frac{1}{(z + \eta \underline{L}_i)^3} \left( \sum_{j \in [K]} \frac{1}{2(z + \eta \underline{L}_j)^2} \frac{2}{\eta} \right) \cdot \exp\left( - \sum_{l \in [K]} \frac{1}{2(z + \eta \underline{L}_l)^2} \right) dz, \quad (17)$$

which implies

$$\sum_{i \in [K]} (\dagger_i) \leq \frac{2e^2}{\eta} \int_0^\infty \sum_{i \in [K]} \frac{1}{(z + \eta \underline{L}_i)^3} \left( \sum_{j \in [K]} \frac{1}{2(z + \eta \underline{L}_j)^2} \right) \cdot \exp\left( - \sum_{l \in [K]} \frac{1}{2(z + \eta \underline{L}_l)^2} \right) dz$$

$$\leq \frac{2e^2}{\eta} \int_0^K w e^{-w} dw \qquad \text{(by } w = \sum_{l \in [K]} \frac{1}{2(z + \eta \underline{L}_l)^2} = \sum_{l \in [K]} F(z + \eta \underline{L}_l))$$

$$\leq \frac{2e^2}{\eta}.$$

On the other hand, we have

$$\sum_{i \in [K]} (\ddagger_i) = \int_{-\infty}^0 \sum_{i \in [K]} f(z + \eta \underline{L}_i) \sum_{j \neq i} (\underline{L}_j - \underline{L}_i) f(z + \eta \underline{L}_j) \left( \prod_{l \neq i,j} F(z + \eta \underline{L}_l) \right) dz = 0. \quad (18)$$

This is because the value of $f(z + \eta \underline{L}_i) f(z + \eta \underline{L}_j) \left( \prod_{l \neq i,j} F(z + \eta \underline{L}_l) \right)$ remains unchanged when $i$ and $j$ are swapped, which makes the integrand zero.

Let $L = \hat{L}_t + \hat{\ell}_t$. Then, we obtain

$$\mathbb{E}\left[\left\langle \hat{\ell}_t, \phi(\eta_t(\hat{L}_t + \hat{\ell}_t)) - \phi(\eta_{t+1}(\hat{L}_t + \hat{\ell}_t))\right\rangle\right] = \sum_{i \in [K]} \mathbb{E}\left[I[I_t = i]\ell_{t,i}\widehat{w_{t,i}^{-1}}(\phi_i(\eta_t L) - \phi_i(\eta_{t+1}L))\right]$$

$$= \sum_{i \in [K]} \mathbb{E}\left[I[I_t = i]\ell_{t,i}\widehat{w_{t,i}^{-1}} \int_{\eta_{t+1}}^{\eta_t} \frac{\partial}{\partial \eta}\phi_i(\eta L)\mathrm{d}\eta\right]$$

$$\leq \sum_{i \in [K]} \mathbb{E}\left[\ell_{t,i} \int_{\eta_{t+1}}^{\eta_t} \frac{\partial}{\partial \eta}\phi_i(\eta L)\mathrm{d}\eta\right]$$

$$\leq \sum_{i \in [K]} \mathbb{E}\left[\int_{\eta_{t+1}}^{\eta_t} \frac{\partial}{\partial \eta}\phi_i(\eta L)\mathrm{d}\eta\right].$$

By combining the results in (17) and (18) with (16), we obtain

$$\mathbb{E}\left[\left\langle \hat{\ell}_t, \phi(\eta_t(\hat{L}_t + \hat{\ell}_t)) - \phi(\eta_{t+1}(\hat{L}_t + \hat{\ell}_t))\right\rangle\right] \leq \sum_{i \in [K]} \mathbb{E}\left[\int_{\eta_{t+1}}^{\eta_t} \frac{4}{27\eta} + (\dagger_i) + (\ddagger_i)\mathrm{d}\eta\right]$$

$$\leq \mathbb{E}\left[\int_{\eta_{t+1}}^{\eta_t} \frac{4K}{27\eta} + \frac{2e^2}{\eta}\mathrm{d}\eta\right]$$

$$= \left(\frac{4K}{27} + 2e^2\right)\log\left(\frac{\eta_t}{\eta_{t+1}}\right).$$

Therefore,

$$\sum_{t=1}^{T}\mathbb{E}\left[\left\langle \hat{\ell}_t, \phi(\eta_t(\hat{L}_t + \hat{\ell}_t)) - \phi(\eta_{t+1}(\hat{L}_t + \hat{\ell}_t))\right\rangle\right] \leq \left(\frac{4K}{27} + 2e^2\right)\log\left(\frac{\eta_1}{\eta_{T+1}}\right).$$

### C.2 Proof of Lemma C.2

By injecting $f$ and $F$, it holds that

$$\phi_i(\underline{\lambda}) = \int_{-\infty}^{-\underline{\lambda}_i} e^{2(z+\underline{\lambda}_i)}\prod_{j \neq i}F(z + \underline{\lambda}_j)\mathrm{d}z + \int_{-\underline{\lambda}_i}^{\infty} \frac{1}{(z + \underline{\lambda}_i + 1)^3}\prod_{j \neq i}F(z + \underline{\lambda}_j)\mathrm{d}z,$$

and

$$\phi_i'(\underline{\lambda}) := \frac{\partial}{\partial \underline{\lambda}_i}\phi_i(\underline{\lambda}) = \int_{-\infty}^{-\underline{\lambda}_i} 2e^{2(z+\underline{\lambda}_i)}\prod_{j \neq i}F(z + \underline{\lambda}_j)\mathrm{d}z + \int_{-\underline{\lambda}_i}^{\infty} \frac{-3}{(z + \underline{\lambda}_i + 1)^4}\prod_{j \neq i}F(z + \underline{\lambda}_j)\mathrm{d}z.$$

Therefore, we obtain for any $\lambda \in [0, \infty)^K$

$$-\phi_i'(\underline{\lambda}) = 3\int_{-\underline{\lambda}_i}^{\infty} \frac{1}{(z + \underline{\lambda}_i + 1)^4}\prod_{j \neq i}F(z + \underline{\lambda}_j)\mathrm{d}z - \int_{-\infty}^{-\underline{\lambda}_i} 2e^{2(z+\underline{\lambda}_i)}\prod_{j \neq i}F(z + \underline{\lambda}_j)\mathrm{d}z$$

$$\leq 3\int_{-\underline{\lambda}_i}^{\infty} \frac{1}{(z + \underline{\lambda}_i + 1)^4}\prod_{j \neq i}F(z + \underline{\lambda}_j)\mathrm{d}z.$$

Moreover, we have for any $x \geq 0$,

$$-\phi_i'(\underline{\lambda} + xe_i) \leq 3\int_{-\underline{\lambda}_i-x}^{\infty} \frac{1}{(z + \underline{\lambda}_i + x + 1)^4}\prod_{j \neq i}F(z + \underline{\lambda}_j)\mathrm{d}z$$

$$\leq 3\int_{-\underline{\lambda}_i}^{\infty} \frac{1}{(z + \underline{\lambda}_i + 1)^4}\prod_{j \neq i}F(z + \underline{\lambda}_j - x)\mathrm{d}z$$

$$\leq 3\int_{-\underline{\lambda}_i}^{\infty} \frac{1}{(z + \underline{\lambda}_i + 1)^4}\prod_{j \neq i}F(z + \underline{\lambda}_j)\mathrm{d}z. \tag{19}$$

Based on these results, we can derive the upper bounds on $\phi(\underline{\lambda}) - \phi(\underline{\lambda} + \eta_t \hat{\ell}_t))$ for any $\underline{\lambda} \in [0, \infty)^K$ as

$$
\begin{aligned}
\phi(\underline{\lambda}) - \phi(\underline{\lambda} + \eta_t \hat{\ell}_t) &= \int_0^{\eta_t \ell_{t,i} \widehat{w_{t,i}^{-1}}} (-\phi_i'(\eta_t \hat{L}_t + x e_i)) \mathrm{d}x \\
&\leq \int_0^{\eta_t \ell_{t,i} \widehat{w_{t,i}^{-1}}} 3 \int_{-\underline{\lambda}_i}^{\infty} \frac{1}{(z + \underline{\lambda}_i + 1)^4} \prod_{j \neq i} F(z + \underline{\lambda}_j) \mathrm{d}z \mathrm{d}x \qquad \text{(by (19))} \\
&\leq 3 \eta_t \ell_{t,i} \widehat{w_{t,i}^{-1}} \int_{-\underline{\lambda}_i}^{\infty} \frac{1}{(z + \underline{\lambda}_i + 1)^4} \prod_{j \neq i} F(z + \underline{\lambda}_j) \mathrm{d}z \\
&= 3 \eta_t \ell_{t,i} \widehat{w_{t,i}^{-1}} \int_{-\underline{\lambda}_i}^{\infty} \frac{1}{(z + \underline{\lambda}_i + 1)^4} \prod_{j \neq i} F(z + \underline{\lambda}_j) \mathrm{d}z.
\end{aligned}
$$

Here, let us decompose the integral above into two terms.

$$
\begin{aligned}
&\int_{-\underline{\lambda}_i}^{\infty} \frac{1}{(z + \underline{\lambda}_i + 1)^4} \prod_{j \neq i} F(z + \underline{\lambda}_j) \mathrm{d}z \\
&= \underbrace{\int_{-\underline{\lambda}_i}^{0} \frac{1}{(z + \underline{\lambda}_i + 1)^4} \prod_{j \neq i} F(z + \underline{\lambda}_j) \mathrm{d}z}_{I_{1,i}} + \underbrace{\int_0^{\infty} \frac{1}{(z + \underline{\lambda}_i + 1)^4} \prod_{j \neq i} F(z + \underline{\lambda}_j) \mathrm{d}z}_{I_{2,i}}.
\end{aligned}
$$

By explicitly considering the formulation of distribution in Proposition 4.1, we can obtain for all $i \in [K]$ that

$$
\frac{I_{1,i}}{I_{2,i}} \leq 4, \ \forall \lambda \in [0, \infty)^K, \tag{20}
$$

whose detailed computation is given in Section C.5 for completeness.

By (20), we obtain

$$
\phi(\lambda) - \phi(\lambda + \eta_t \hat{\ell}_t) \leq 15 \eta_t \ell_{t,i} \widehat{w_{t,i}^{-1}} \int_0^{\infty} \frac{1}{(z + \underline{\lambda}_i + 1)^4} \prod_{j \neq i} F(z + \underline{\lambda}_j) \mathrm{d}z.
$$

For notational simplicity, we define

$$
\psi_i(\lambda) = \int_0^{\infty} \frac{1}{(z + \lambda_i + 1)^4} \prod_{j \neq i} F(z + \lambda_j) \mathrm{d}z.
$$

Since $\widehat{w_{t,I_t}^{-1}}$ follows the geometric distributions with expectation $w_{t,I_t}^{-1}$ given $\hat{L}_t$ and $I_t$, it holds that

$$
\mathbb{E}\left[\widehat{w_{t,I_t}^{-1}}^2 \,\middle|\, \hat{L}_t, I_t\right] = \frac{2}{w_{t,I_t}^2} - \frac{1}{w_{t,I_t}} \leq \frac{2}{w_{t,I_t}^2}. \tag{21}
$$

Therefore, when $I_t = i$ and $\{\lambda_j\}$ are sorted, we obtain

$$
\mathbb{E}\left[\hat{\ell}_{t,i}(\phi_i(\eta_t \hat{L}_t) - \phi_i(\eta_t(\hat{L}_t + \hat{\ell}_t))) \,\middle|\, \hat{L}_t\right]
$$

$$
\leq \mathbb{E}\left[\mathbb{1}[I_t = i]\ell_{t,i}\widehat{w_{t,I_t}^{-1}} \cdot 15\eta_t \ell_{t,i}\widehat{w_{t,i}^{-1}} \int_0^\infty \frac{1}{(z + \eta_t \hat{\underline{L}}_{t,i} + 1)^4} \prod_{j \neq i} F(z + \eta_t \hat{\underline{L}}_{t,j})\mathrm{d}z \,\middle|\, \hat{L}_t\right]
$$

$$
\leq 30\eta_t \mathbb{E}\left[w_{t,i}\frac{\ell_{t,i}^2 \psi_i(\eta_t \hat{L}_t)}{w_{t,i}^2} \,\middle|\, \hat{L}_t\right]
$$

$$
\leq 30\eta_t \mathbb{E}\left[\frac{\psi_i(\eta_t \hat{L}_t)}{w_{t,i}} \,\middle|\, \hat{L}_t\right] \qquad\qquad\qquad (\text{by } \ell_{t,i} \in [0,1])
$$

$$
= 30\eta_t \mathbb{E}\left[\frac{\psi_i(\eta_t \hat{L}_t)}{\int_{-\infty}^\infty f(z + \eta_t \hat{\underline{L}}_{t,i}) \prod_{j \neq i} F(z + \eta_t \hat{\underline{L}}_{t,j})\mathrm{d}z} \,\middle|\, \hat{L}_t\right]
$$

$$
\leq 30\eta_t \mathbb{E}\left[\frac{\psi_i(\eta_t \hat{L}_t)}{\int_0^\infty f(z + \eta_t \hat{\underline{L}}_{t,i}) \prod_{j \neq i} F(z + \eta_t \hat{\underline{L}}_{t,j})\mathrm{d}z} \,\middle|\, \hat{L}_t\right] = 30\eta_t \mathbb{E}\left[\frac{\psi_i(\eta_t \hat{L}_t)}{\bar{\phi}_i(\eta_t \hat{L}_t)} \,\middle|\, \hat{L}_t\right],
$$

where $\bar{\phi}_i(\lambda) = \int_0^\infty f(z + \lambda_i) \prod_{j \neq i} F(z + \lambda_j)\mathrm{d}z$. Then, the following lemma concludes the proof.

**Lemma C.4.** *If $\lambda_i$ is the $\sigma_i$-th smallest among $\{\lambda_j\}$ (ties are broken arbitrarily), then*

$$
\frac{\psi_i(\underline{\lambda})}{\bar{\phi}_i(\underline{\lambda})} \leq \frac{\sqrt{\pi}}{\sqrt{\sigma_i}} \wedge \frac{\sqrt{2e}}{3}\frac{1}{\underline{\lambda}_i}.
$$

### C.3  Proof of Lemma C.3

Since $\mathbb{E}_{X \sim \mathrm{LP}}[X] = \frac{1}{4}$, it holds that

$$
\mathbb{E}\left[r_{t,I_t} - r_{t,i^*} \,\middle|\, \hat{L}_t\right] \leq \sum_{i \neq i^*} \mathbb{E}\left[\mathbb{1}[I_t = i]r_{t,I_t} \,\middle|\, \hat{L}_t\right]
$$

$$
\leq \sum_{i \neq i^*} \mathbb{E}\left[\mathbb{1}[I_t = i]\mathbb{1}[r_{t,I_t} \geq 0]r_{t,I_t} \,\middle|\, \hat{L}_t\right]
$$

$$
= \int_0^\infty \sum_{i \neq i^*}\left(\frac{1}{(z + \eta_t \hat{\underline{L}}_{t,i} + 1)^2} \prod_{j \neq i}\left(1 - \frac{1}{2(z + \eta_t \hat{\underline{L}}_{t,j} + 1)^2}\right)\right)\mathrm{d}z \quad (22)
$$

$$
\leq \int_0^\infty \sum_{i \neq i^*} \frac{1}{(z + \eta_t \hat{\underline{L}}_{t,i} + 1)^2}\mathrm{d}z
$$

$$
\leq \int_0^\infty \sum_{i \neq i^*} \frac{1}{(z + \eta_t \hat{\underline{L}}_{t,i})^2}\mathrm{d}z = \sum_{i \neq i^*} \frac{1}{\eta_t \hat{\underline{L}}_{t,i}}.
$$

Let $f(z) = \sum_i \frac{1}{2(z+\eta_t \hat{\underline{L}}_{t,i}+1)^2} \in \left(0, \frac{K}{2(z+1)^2}\right]$. Then, we can also bound (22) by

$$\int_0^\infty \sum_{i \neq i^*} \left( \frac{1}{(z+\eta_t \hat{\underline{L}}_{t,i}+1)^2} \prod_{j \neq i} \left( 1 - \frac{1}{2(z+\eta_t \hat{\underline{L}}_{t,j}+1)^2} \right) \right) dz$$

$$\leq e \int_0^\infty \sum_{i \neq i^*} \left( \frac{1}{(z+\eta_t \hat{\underline{L}}_{t,i}+1)^2} e^{-f(z)} \right) dz$$

$$\leq 2e \int_0^\infty f(z) e^{-f(z)} dz$$

$$\leq 2e \int_0^{\sqrt{2(K-1)}} e^{-1} dz + 2e \int_{\sqrt{2(K-1)}}^\infty \frac{K}{2(z+1)^2} \exp\left( -\frac{K}{2(z+1)^2} \right) dz$$

$$= 2\sqrt{2}\sqrt{K-1} + \frac{e}{\sqrt{2}}\sqrt{K} \int_0^1 w^{-1/2} e^{-w} dw$$

$$\leq 5.7\sqrt{K}.$$

## C.4 Proof of Lemma C.4

We first show that $\frac{\psi_i(\lambda)}{\bar{\phi}_i(\lambda)}$ is monotonically increasing in $\lambda_j$ for any $j \neq i$ and $\lambda \in [0,\infty)^K$.

By definition, it holds that

$$\frac{\psi_i(\lambda)}{\bar{\phi}_i(\lambda)} = \frac{\int_0^\infty \frac{1}{(z+\lambda_i+1)^4} \prod_{j \neq i} F(z+\lambda_j) dz}{\int_0^\infty f(z+\lambda_i) \prod_{j \neq i} F(z+\lambda_j) dz}$$

$$= \frac{\int_0^\infty \frac{1}{(z+\lambda_i+1)^4} \prod_{j \neq i} \left( 1 - \frac{1}{2(z+\lambda_j+1)^2} \right) dz}{\int_0^\infty \frac{1}{(z+\lambda_i+1)^3} \prod_{j \neq i} \left( 1 - \frac{1}{2(z+\lambda_j+1)^2} \right) dz}$$

$$= \frac{\int_1^\infty \frac{1}{(z+\lambda_i)^4} \prod_{j \neq i} \left( 1 - \frac{1}{2(z+\lambda_j)^2} \right) dz}{\int_1^\infty \frac{1}{(z+\lambda_i)^3} \prod_{j \neq i} \left( 1 - \frac{1}{2(z+\lambda_j)^2} \right) dz}.$$

Since $\frac{1}{z^3} \frac{1}{1 - \frac{1}{2z^2}}$ is monotonically decreasing for $z \geq 1$, applying Lemma 9 in Lee et al. [38] implies that $\frac{\psi_i(\lambda)}{\bar{\phi}_i(\lambda)}$ is monotonically decreasing in $\lambda_j$.

By the monotonicity of $\psi_i(\lambda)/\bar{\phi}_i(\lambda)$, we have

$$\frac{\psi_i(\underline{\lambda})}{\bar{\phi}_i(\underline{\lambda})} \leq \frac{\psi_i(\lambda^*)}{\bar{\phi}_i(\lambda^*)}, \quad \text{where} \quad \lambda_j^* = \begin{cases} \underline{\lambda}_i, & j \leq i, \\ \infty, & j > i. \end{cases}$$

By definition, we have

$$\psi_i(\lambda^*) = \int_0^\infty \frac{1}{(z+\underline{\lambda}_i+1)^4} \left( 1 - \frac{1}{2(z+\underline{\lambda}_i+1)^2} \right)^{i-1} dz$$

$$= \int_0^{\frac{1}{2(\underline{\lambda}_i+1)^2}} \sqrt{2} w^{\frac{1}{2}} (1-w)^{i-1} dw \qquad \text{(by } w = \frac{1}{2(z+\underline{\lambda}_i+1)^2} \text{)}$$

$$= \sqrt{2} B\left( \frac{1}{2(\underline{\lambda}_i+1)^2}; \frac{3}{2}, i \right),$$

where $B(x; a, b) = \int_0^x t^{a-1}(1-t)^{b-1}\mathrm{d}t$ denotes the incomplete Beta function. Similarly, we obtain

$$\bar{\phi}_i(\lambda^*) = \int_0^\infty \frac{1}{(z + \underline{\lambda}_i + 1)^3}\left(1 - \frac{1}{2(z + \underline{\lambda}_i + 1)^2}\right)^{i-1}\mathrm{d}z$$

$$= \int_0^{\frac{1}{2(\underline{\lambda}_i+1)^2}} \sqrt{2}w^0(1-w)^{i-1}\mathrm{d}w$$

$$= \sqrt{2}B\left(\frac{1}{2(\underline{\lambda}_i + 1)^2}; 1, i\right).$$

Therefore, we can directly apply the result in Lee et al. [38, Appendix C.2.2.] with $\alpha = 2$ and $\frac{1}{2(\underline{\lambda}_i+1)^2}$ instead of $\frac{1}{(\underline{\lambda}_i+1)^2}$, which gives

$$\frac{\psi_i(\lambda^*)}{\bar{\phi}_i(\lambda^*)} \leq \frac{\sqrt{\pi}}{\sqrt{\sigma_i}} \wedge \frac{\sqrt{2e}}{3}\frac{1}{\underline{\lambda}_i}.$$

## C.5 Proof of (20)

When $i = 1$, it holds trivially by $I_{1,i} = 0$. Assume $i > 1$. Let $G_i(z) = \prod_{j \neq 1, i} F(z + \underline{\lambda}_j)$. Since $\underline{\lambda}_1 = 0$ and $G_i(z)$ is increasing, we have

$$I_{1,i} = \int_{-\underline{\lambda}_i}^0 \frac{1}{(z + \underline{\lambda}_i + 1)^4}\frac{e^{2z}}{2}G_i(z)\mathrm{d}z$$

$$\leq \int_{-\underline{\lambda}_i}^0 \frac{1}{(z + \underline{\lambda}_i + 1)^4}\frac{e^{2z}}{2}G_i(0)\mathrm{d}z,$$

$$I_{2,i} = \int_0^\infty \frac{1}{(z + \underline{\lambda}_i + 1)^4}\left(1 - \frac{1}{2(x+1)^2}\right)G_i(z)\mathrm{d}z$$

$$\geq \int_0^\infty \frac{1}{(z + \underline{\lambda}_i + 1)^4}\left(1 - \frac{1}{2(x+1)^2}\right)G_i(0)\mathrm{d}z.$$

Therefore,

$$\frac{I_{1,i}}{I_{2,i}} \leq \frac{\int_{-\underline{\lambda}_i}^0 \frac{1}{(z+\underline{\lambda}_i+1)^4}\frac{e^{2z}}{2}\mathrm{d}z}{\int_0^\infty \frac{1}{(z+\underline{\lambda}_i+1)^4}\left(1 - \frac{1}{2(x+1)^2}\right)\mathrm{d}z}$$

$$\leq 2\frac{\int_{-\underline{\lambda}_i}^0 \frac{1}{(z+\underline{\lambda}_i+1)^4}\frac{e^{2z}}{2}\mathrm{d}z}{\int_0^\infty \frac{1}{(z+\underline{\lambda}_i+1)^4}\mathrm{d}z}$$

$$= 3(\underline{\lambda}_i + 1)^3\int_{-\underline{\lambda}_i}^0 \frac{1}{(z + \underline{\lambda}_i + 1)^4}e^{2z}\mathrm{d}z.$$

Here, $\frac{\mathrm{d}}{\mathrm{d}z}\frac{1}{(z+\underline{\lambda}_i+1)^4}e^{2z} = \frac{2e^{2z}(z+\underline{\lambda}_i-1)}{(z+\underline{\lambda}_i+1)^5}$ holds, whose maximum in the interval of integral occurs at either $z = -\underline{\lambda}_i$ or $z = 0$. Therefore, we obtain that

$$3(\underline{\lambda}_i + 1)^3\int_{-\underline{\lambda}_i}^0 \frac{1}{(z + \underline{\lambda}_i + 1)^4}e^{2z}\mathrm{d}z \leq 3(\underline{\lambda}_i + 1)^3\underline{\lambda}_i\left(e^{-2\underline{\lambda}_i} \vee \frac{1}{(\underline{\lambda}_i + 1)^4}\right),$$

where $\vee$ denotes the max operator. Since $\underline{\lambda}_i > 0$, it is easy to see

$$\frac{3(\underline{\lambda}_i + 1)^3\underline{\lambda}_i}{(\underline{\lambda}_i + 1)^4} = \frac{3\underline{\lambda}_i}{\underline{\lambda}_i + 1} \leq 3.$$

On the other hand, by simple calculus, we obtain for $x > 0$

$$3(x + 1)^3xe^{-2x} \leq \frac{9}{4}(12 + 7\sqrt{3})e^{-1-\sqrt{3}} < 4.$$

where equality holds when $x = \frac{1+\sqrt{3}}{2}$. This completes the proof.

# D Proofs of the stochastic regret for Laplace-Pareto perturbations

In this section, we provide the proof of Theorem 4.4 based on the self-bounding techniques, where we adopt idea in Honda et al. [28] and Lee et al. [38]. Firstly, define a event $D_t$ by

$$D_t := \left\{ \sum_{i \neq i^*} \frac{1}{(1 + \eta_t \hat{\underline{L}}_{t,i})^2} \leq \frac{1}{4} \right\}, \tag{23}$$

where on $D_t$

$$\hat{\underline{L}}_{t,i^*} = 0 \quad \text{and} \quad \eta_t \hat{\underline{L}}_{t,j} \geq 1, \ \forall j \neq i^*. \tag{24}$$

## D.1 Regret lower bounds

Here, we provide the regret lower bounds, which are used for the self-bound technique.

**Lemma D.1.** *Let $\Delta := \min_{i \neq i^*} \Delta_i$. Then, it holds that*

*(i) On $D_t$, $\sum_{i \neq i^*} \Delta_i w_{t,i} \geq \frac{1}{8e^{5/4}} \sum_{i \neq i^*} \frac{\Delta_i}{(\eta_t \hat{\underline{L}}_{t,i})^2}$ and $w_{t,i^*} \geq \frac{e^{-1/4}}{2}$.*

*(ii) On $D_t^c$, $\sum_{i \neq i^*} \Delta_i w_{t,i} \geq \Delta \frac{e^{-1}}{2}(1 - e^{-1/4})$.*

Although the proof is largely the same as in Lemma 22 of Lee et al. [38], we include the details here for completeness.

*Proof.* Let $\hat{\underline{L}}' = \min_{i \neq i^*} \hat{\underline{L}}_{t,i}$. Then, by definition of $w$, we have

$$\sum_{i \neq i^*} \Delta_i w_{t,i} = \int_{-\infty}^{\infty} \left( \sum_{i \neq i^*} \Delta_i f(z + \eta_t \hat{\underline{L}}_{t,i}) \prod_{j \neq i} F(z + \eta_t \hat{\underline{L}}_{t,j}) \right) dz$$

$$\geq \int_{-\infty}^{\infty} \left( \sum_{i \neq i^*} \Delta_i f(z + \eta_t \hat{\underline{L}}_{t,i}) \right) \prod_{j \in [K]} F(z + \eta_t \hat{\underline{L}}_{t,j}) dz$$

$$\geq \int_{-\infty}^{\infty} \left( \sum_{i \neq i^*} \Delta_i f(z + \eta_t \hat{\underline{L}}_{t,i}) \right) \exp\left( -\sum_{j \in [K]} \frac{1 - F(z + \eta_t \hat{\underline{L}}_{t,i})}{F(z + \eta_t \hat{\underline{L}}_{t,i})} \right) dz \tag{25}$$

$$\geq \int_{-\infty}^{\infty} \left( \sum_{i \neq i^*} \Delta_i f(z + \eta_t \hat{\underline{L}}_{t,i}) \right) \exp\left( -\sum_{j \neq i^*} \frac{1 - F(z + \eta_t \hat{\underline{L}}_{t,i})}{F(z + \eta_t \hat{\underline{L}}_{t,i})} \right) \exp\left( -\frac{1 - F(z)}{F(z)} \right) dz, \tag{26}$$

where (25) holds since $e^{-\frac{x}{1-x}} < 1 - x$ holds for $x < 1$.

(i) On $D_t$, we obtain

$$\int_{-\infty}^{\infty}\left(\sum_{i\neq i^*}\Delta_i f(z+\eta_t\hat{\underline{L}}_{t,i})\right)\exp\left(-\sum_{j\neq i^*}\frac{1-F(z+\eta_t\hat{\underline{L}}_{t,i})}{F(z+\eta_t\hat{\underline{L}}_{t,i})}\right)\exp\left(-\frac{1-F(z)}{F(z)}\right)\mathrm{d}z$$

$$\geq e^{-1}\int_0^{\infty}\left(\sum_{i\neq i^*}\Delta_i f(z+\eta_t\hat{\underline{L}}_{t,i})\right)\exp\left(-2\sum_{j\neq i^*}(1-F(z+\eta_t\hat{\underline{L}}_{t,i}))\right)\mathrm{d}z$$

$$= e^{-1}\int_0^{\infty}\left(\sum_{i\neq i^*}\Delta_i f(z+\eta_t\hat{\underline{L}}_{t,i})\right)\exp\left(-\sum_{j\neq i^*}\frac{1}{(1+\eta_t\hat{\underline{L}}_{t,j})^2}\right)\mathrm{d}z$$

$$\geq e^{-5/4}\int_0^{\infty}\left(\sum_{i\neq i^*}\Delta_i f(z+\eta_t\hat{\underline{L}}_{t,i})\right)\mathrm{d}z \qquad\qquad \text{(by definition of } D_t \text{ in (23))}$$

$$= e^{-5/4}\sum_{i\neq i^*}\Delta_i\left(1-F\left(\eta_t\hat{\underline{L}}_{t,i}\right)\right)$$

$$= e^{-5/4}\sum_{i\neq i^*}\frac{\Delta_i}{2(1+\eta_t\hat{\underline{L}}_{t,i})^2} \geq \frac{1}{8e^{5/4}}\sum_{i\neq i^*}\frac{\Delta_i}{(\eta_t\hat{\underline{L}}_{t,i})^2}. \qquad (\eta_t\hat{\underline{L}}_{t,j}\geq 1 \text{ for } j\neq i^* \text{ on } D_t)$$

On the other hand, on $D_t$, we have

$$w_{t,i^*} = \int_{-\infty}^{\infty}\frac{1}{(|z|+1)^3}\prod_{j\neq i^*}F(z+\eta_t\hat{\underline{L}}_{t,j})\mathrm{d}z$$

$$\geq \int_0^{\infty}\frac{1}{(z+1)^3}\prod_{j\neq i^*}F(z+\eta_t\hat{\underline{L}}_{t,j})\mathrm{d}z$$

$$\geq \int_0^{\infty}\frac{1}{(z+1)^3}\exp\left(-\sum_{j\neq i^*}\frac{1-F(z+\eta_t\hat{\underline{L}}_{t,j})}{F(z+\eta_t\hat{\underline{L}}_{t,j})}\right)\mathrm{d}z$$

$$\geq \int_0^{\infty}\frac{1}{(z+1)^3}\exp\left(-2\sum_{j\neq i^*}(1-F(z+\eta_t\hat{\underline{L}}_{t,j}))\right)\mathrm{d}z$$

$$= \int_0^{\infty}\frac{1}{(z+1)^3}\exp\left(-\sum_{j\neq i^*}\frac{1}{(1+\eta_t\hat{\underline{L}}_{t,j})^2}\right)\mathrm{d}z$$

$$\geq e^{-\frac{1}{4}}\int_0^{\infty}\frac{1}{(z+1)^3}\mathrm{d}z \qquad\qquad \text{(by definition of } D_t \text{ in (23))}$$

$$= \frac{e^{-1/4}}{2} \approx 0.3894,$$

which concludes the proof of the case (i).

(ii) From (26), on $D_t^c$, we have

$$\int_{-\infty}^{\infty} \left( \sum_{i \neq i^*} \Delta_i f(z + \eta_t \underline{\hat{L}}_{t,i}) \right) \exp\left( -\sum_{j \neq i^*} \frac{1 - F(z + \eta_t \underline{\hat{L}}_{t,i})}{F(z + \eta_t \underline{\hat{L}}_{t,i})} \right) \exp\left( -\frac{1 - F(z)}{F(z)} \right) dz$$

$$\geq \Delta \int_0^{\infty} \left( \sum_{i \neq i^*} f(z + \eta_t \underline{\hat{L}}_{t,i}) \right) \exp\left( -\sum_{j \neq i^*} \frac{1 - F(z + \eta_t \underline{\hat{L}}_{t,i})}{F(z + \eta_t \underline{\hat{L}}_{t,i})} \right) \exp\left( -\frac{1 - F(z)}{F(z)} \right) dz$$

$$\geq \Delta e^{-1} \int_0^{\infty} \left( \sum_{i \neq i^*} f(z + \eta_t \underline{\hat{L}}_{t,i}) \right) \exp\left( -2 \sum_{j \neq i^*} (1 - F(z + \eta_t \underline{\hat{L}}_{t,i})) \right) dz$$

$$= \Delta \frac{e^{-1}}{2} \left( 1 - \exp\left( -2 \sum_{j \neq i^*} (1 - F(\eta_t \underline{\hat{L}}_{t,j})) \right) \right)$$

$$= \Delta \frac{e^{-1}}{2} \left( 1 - \exp\left( -\sum_{j \neq i^*} \frac{1}{(1 + \eta_t \underline{\hat{L}}_{t,j})^2} \right) \right)$$

$$\geq \Delta \frac{e^{-1}}{2} (1 - e^{-1/4}), \qquad\qquad (\textstyle\sum_{j \neq i^*} \frac{1}{(1+\eta_t \underline{\hat{L}}_{t,j})^2} \geq \frac{1}{4} \text{ on } D_t^c)$$

which concludes the proof. $\qquad\square$

## D.2 Regret for the optimal arm

To apply the self-bounding technique, we need to express the regret of the optimal arm using the statistics of the other arms. Although the proof is largely the same as in Lemma 11 of Honda et al. [28], we include the details here for completeness, accounting for some additional terms due to negative perturbations.

Similarly to the proof in Lee et al. [38], we begin by introducing the following lemma.

**Lemma D.2** (Partial result of Lemma 11 in Honda et al. [28]). *For any $\hat{L}_t$ and $\zeta \in (0,1)$, it holds that*

$$\mathbb{E}\left[ \mathbb{1}\left[ \hat{\ell}_{t,i^*} > \frac{\zeta}{\eta_t} \right] \hat{\ell}_{t,i^*} \,\Big|\, \hat{L}_t \right] \leq \frac{1}{1 - e^{-1}} (1 - e^{-1})^{\frac{\zeta}{\eta_t}} \left( \frac{\zeta}{\eta_t} + e \right)$$

*and when $\eta_t = \frac{m}{\sqrt{t}}$ and $\zeta = 1 - (4e/21)^{1/3}$, it holds that*

$$\sum_{t=1}^{\infty} \frac{1}{1 - e^{-1}} (1 - e^{-1})^{\frac{\zeta}{\eta_t}} \left( \frac{\zeta}{\eta_t} + e \right) \leq 2743 m^2 + 77 m.$$

Based on this result, we obtain the following lemma.

**Lemma D.3.** *On $D_t$, for any $\zeta \in (0,1)$, it holds that*

$$\mathbb{E}\left[ \hat{\ell}_{t,i^*} \left( \phi_{i^*}(\eta_t \hat{L}_t) - \phi_{i^*}(\eta_t (\hat{L}_t + \hat{\ell}_t)) \right) \,\Big|\, \hat{L}_t \right] \leq \frac{13 e^{1/4}}{(1 - \zeta)} \sum_{j \neq i^*} \frac{1}{\underline{\hat{L}}_{t,j}} + \frac{1}{1 - e^{-1}} (1 - e^{-1})^{\frac{\zeta}{\eta_t}} \left( \frac{\zeta}{\eta_t} + e \right).$$

*Proof.* Recall that $\underline{\hat{L}}_{t,i^*} = 0$ and $\underline{\hat{L}}_{t,j} > 0$ for all $j \neq i^*$ on $D_t$. Moreover, from (24), $\underline{\hat{L}}_{t,j} \geq 1/\eta_t$ holds for all $j \neq i^*$ on $D_t$. Then, we consider two cases separately: (i) $\widehat{w_{t,i^*}^{-1}} \leq \frac{\zeta}{\eta_t}$ and (ii) $\widehat{w_{t,i^*}^{-1}} > \frac{\zeta}{\eta_t}$.

(i) When $\widehat{w_{t,i^*}^{-1}} \le \frac{\zeta}{\eta_t}$, we have $\hat{\ell}_{t,i^*} \le \frac{\zeta}{\eta_t}$ by definition. For any $x \le \frac{\zeta}{\eta_t}$ and $i \ne i^*$, we have

$$\phi_i\Big(\eta_t(\hat{L}_t + xe_{i^*})\Big) = \int_{-\infty}^{\infty} f(z + \eta_t \hat{L}_{t,i}) \prod_{j \ne i} F\Big(z + \eta_t \hat{L}_{t,j} + x\mathbb{1}[j = i^*]\Big) dz$$

$$= \int_{-\infty}^{\infty} f(z + \eta_t \underline{\hat{L}}_{t,i}) \prod_{j \ne i} F\Big(z + \eta_t \underline{\hat{L}}_{t,j} + x\mathbb{1}[j = i^*]\Big) dz$$

$$= \int_{-\infty}^{\infty} f(z + \eta_t \underline{\hat{L}}_{t,i}) F(z + \eta_t x) \prod_{j \ne i, i^*} F\Big(z + \eta_t \underline{\hat{L}}_{t,j}\Big) dz.$$

Here, note that whenever $x \le \frac{\zeta}{\eta_t}$, $\hat{L}_{t,i} - x \ge (1 - \zeta)\hat{L}_{t,i}$ holds for all $i \ne i^*$ on $D_t$. Then, by differentiating with respect to $x$, we obtain

$$\frac{d}{dx}\phi_i\Big(\eta_t(\hat{L}_t + xe_{i^*})\Big) = \eta_t \int_{-\infty}^{\infty} f\Big(z + \eta_t \underline{\hat{L}}_{t,i}\Big) f(z + \eta_t x) \prod_{j \ne i, i^*} F\Big(z + \eta_t \underline{\hat{L}}_{t,j}\Big) dz$$

$$= \eta_t \int_{-\infty}^{\infty} f\Big(z + \eta_t(\underline{\hat{L}}_{t,i} - x)\Big) f(z) \prod_{j \ne i, i^*} F\Big(z + \eta_t(\underline{\hat{L}}_{t,j} - x)\Big) dz$$

$$\le \underbrace{\eta_t \int_{-\infty}^{0} f\Big(z + \eta_t(\underline{\hat{L}}_{t,i} - x)\Big) f(z) dz}_{\dagger_i} + \underbrace{\eta_t \int_{0}^{\infty} f\Big(z + \eta_t(\underline{\hat{L}}_{t,i} - x)\Big) f(z) dz}_{\ddagger_i}.$$

$$(27)$$

For $\ddagger_i$ term, since $\eta_t(\hat{L}_{t,i} - x) \ge 0$ in case (i), we obtain for any $i \ne i^*$ that

$$\ddagger_i \le \eta_t \int_0^{\infty} \frac{1}{(z + \eta_t(\hat{L}_{t,i} - x) + 1)^3} \frac{1}{(z + 1)^3} dz$$

$$\le \eta_t \frac{1}{(1 + (1 - \zeta)\eta_t \underline{\hat{L}}_{t,i})^3} \int_0^{\infty} \frac{1}{(z + 1)^3} dz$$

$$\le \frac{\eta_t}{6(1 - \zeta)\eta_t \underline{\hat{L}}_{t,i}} = \frac{1}{6(1 - \zeta)\underline{\hat{L}}_{t,i}}.$$

On the other hand, $\dagger_i$ term can be decomposed into two terms by

$$\dagger_i = \underbrace{\eta_t \int_{-\infty}^{-\eta_t(\underline{\hat{L}}_{t,i} - x)} f\Big(z + \eta_t(\underline{\hat{L}}_{t,i} - x)\Big) f(z) dz}_{\dagger_{i,1}} + \underbrace{\eta_t \int_{-\eta_t(\underline{\hat{L}}_{t,i} - x)}^{0} f\Big(z + \eta_t(\underline{\hat{L}}_{t,i} - x)\Big) f(z) dz}_{\dagger_{i,2}}.$$

Since $f(x) = e^{2x}$ on $x < 0$, it holds that

$$\dagger_{i,1} = \eta_t \int_{-\infty}^{-\eta_t(\underline{\hat{L}}_{t,i} - x)} \exp\Big(4z + \eta_t(\underline{\hat{L}}_{t,i} - x)\Big) dz$$

$$= \frac{\eta_t e^{-3\eta_t(\underline{\hat{L}}_{t,i} - x)}}{4}$$

$$\le \frac{\eta_t e^{-3\eta_t(1 - \zeta)\hat{L}_{t,i}}}{4}$$

$$\le \frac{\eta_t}{4} \frac{1}{3(1 - \zeta)\eta_t \underline{\hat{L}}_{t,i} + 1} \qquad \text{(by } e^{-x} < \tfrac{1}{1+x} \text{ for } x > -1\text{)}$$

$$\le \frac{1}{12(1 - \zeta)\underline{\hat{L}}_{t,i}}.$$

For the second term, we have

$$\dagger_{i,2} = \eta_t \int_{-\eta_t(\underline{\hat{L}}_{t,i} - x)}^{0} \frac{e^{2z}}{(z + \eta_t(\underline{\hat{L}}_{t,i} - x) + 1)^3} dz,$$

where the maximum of $\frac{e^{2z}}{(z+c+1)^3}$ on $z \in [-c, 0]$ occurs at either $z = -c$ or $z = 0$ since its minimum is achieved at $z = -c + \frac{1}{2}$. This implies that

$$\frac{e^{2z}}{(z + \eta_t(\hat{\underline{L}}_{t,i} - x) + 1)^3} \leq \frac{1}{(\eta_t(\hat{\underline{L}}_{t,i} - x) + 1)^3} \vee \exp\left(-2\eta_t(\hat{\underline{L}}_{t,i} - x)\right) \tag{28}$$

Here, since $\frac{1}{(y+1)^3} \geq e^{-2y}$ for all $y \geq 1.15$, we obtain

$$\mathbb{1}\left[\eta_t(\hat{\underline{L}}_{t,i} - x) \geq 1.15\right]\dagger_{i,2} \leq \eta_t \frac{\eta_t(\hat{\underline{L}}_{t,i} - x)}{(\eta_t(\hat{\underline{L}}_{t,i} - x) + 1)^3}$$

$$\leq \frac{\eta_t}{(\eta_t\hat{\underline{L}}_{t,i}(1 - \zeta) + 1)^2}$$

$$\leq \frac{1}{2(1 - \zeta)\hat{\underline{L}}_{t,i}}. \tag{29}$$

On the other hand, when $\eta_t(\hat{\underline{L}}_{t,i} - x) \in (0, 1.15)$, we need more careful approach to provide a tight bound. Similarly to (20), we first evaluate $\frac{\ddagger_{i,2}}{\dagger_i}$ on $D_t \cap \left\{\eta_t(\hat{\underline{L}}_{t,i} - x) \leq 1.15\right\}$, which satisfies

$$\frac{\int_{-\eta_t(\hat{\underline{L}}_{t,i}-x)}^0 \frac{e^{2z}}{(z+\eta_t(\hat{\underline{L}}_{t,i}-x)+1)^3}\mathrm{d}z}{\int_0^\infty \frac{1}{(z+\eta_t(\hat{\underline{L}}_{t,i}-x)+1)^3}\frac{1}{(z+1)^3}\mathrm{d}z} \leq \frac{1/2}{\int_0^\infty \frac{1}{(z+\eta_t(\hat{\underline{L}}_{t,i}-x)+1)^3}\frac{1}{(z+1)^3}\mathrm{d}z}.$$

When $\eta_t(\hat{\underline{L}}_{t,i} - x) \leq 1.15$, the denominator can be evaluated as

$$\int_0^\infty \frac{1}{(z + \eta_t(\hat{\underline{L}}_{t,i} - x) + 1)^3}\frac{1}{(z+1)^3}\mathrm{d}z \geq \int_0^\infty \frac{1}{(z + 2.15)^3}\frac{1}{(z+1)^3}\mathrm{d}z$$

$$\geq 0.028.$$

Therefore, on $D_t \cap \left\{\eta_t(\hat{\underline{L}}_{t,i} - x) \leq 1.15\right\}$, we obtain

$$\dagger_{i,2} \leq \frac{0.5}{0.028}\ddagger_i \leq 18\ddagger_i \leq \frac{3}{(1 - \zeta)\hat{\underline{L}}_{t,i}}.$$

Combining this result with (29), on $D_t$, it holds that

$$\dagger_{i,2} \leq \frac{3}{(1 - \zeta)\hat{\underline{L}}_{t,i}}.$$

Therefore, for any $i \neq i^*$, we obtain

$$\frac{\mathrm{d}}{\mathrm{d}x}\phi_i\left(\eta_t(\hat{L}_t + xe_{i^*})\right) \leq \frac{3 + 1/4}{(1 - \zeta)\hat{\underline{L}}_{t,i}}.$$

Combining this with $\sum_i \phi_i(\lambda) = 1$, we have

$$\mathbb{E}\left[\mathbb{1}[\hat{\ell}_{t,i^*} \le \zeta/\eta_t]\hat{\ell}_{t,i^*}\left(\phi_{i^*}(\eta_t\hat{L}_t) - \phi_{i^*}(\eta_t(\hat{L}_t + \hat{\ell}_t))\right)\Big|\hat{L}_t\right]$$

$$= \mathbb{E}\left[\mathbb{1}[I_t = i^*, \hat{\ell}_{t,i^*} \le \zeta/\eta_t]\hat{\ell}_{t,i^*}\left(\phi_{i^*}(\eta_t\hat{L}_t) - \phi_{i^*}(\eta_t(\hat{L}_t + \hat{\ell}_t))\right)\Big|\hat{L}_t\right]$$

$$= \mathbb{E}\left[\mathbb{1}[I_t = i^*, \hat{\ell}_{t,i^*} \le \zeta/\eta_t]\hat{\ell}_{t,i^*}\left(\phi_{i^*}(\eta_t\hat{L}_t) - \phi_{i^*}(\eta_t(\hat{L}_t + \hat{\ell}_t))\right)\Big|\hat{L}_t\right]$$

$$= \mathbb{E}\left[\mathbb{1}[I_t = i^*, \hat{\ell}_{t,i^*} \le \zeta/\eta_t]\hat{\ell}_{t,i^*}\sum_{i \ne i^*}\left(\phi_i(\eta_t\hat{L}_t) - \phi_i(\eta_t(\hat{L}_t + \hat{\ell}_t))\right)\Big|\hat{L}_t\right]$$

$$\le \mathbb{E}\left[\mathbb{1}[I_t = i^*]\hat{\ell}_{t,i^*}^2 \sum_{i \ne i^*}\frac{3 + 1/4}{(1-\zeta)\underline{\hat{L}}_{t,i}}\Big|\hat{L}_t\right]$$

$$\le \mathbb{E}\left[\frac{2\ell_{t,i^*}^2}{w_{t,i^*}}\sum_{i \ne i^*}\frac{3 + 1/4}{(1-\zeta)\underline{\hat{L}}_{t,i}}\Big|\hat{L}_t\right] \qquad \text{(by (21))}$$

$$\le 4e^{1/4}(3 + 1/4)\sum_{i \ne i^*}\frac{1}{(1-\zeta)\underline{\hat{L}}_{t,i}}. \qquad \text{(by (i) of Lemma D.1)}$$

For the case where $x \ge \zeta/\eta_t$, we can directly apply Lemma D.2, which concludes the proof. $\qquad\square$

### D.3 Proof of Theorem 4.4

We begin by revisiting the regret decomposition in (13) and Lemma C.1, where we obtain

$$\text{Reg}(T) \le \sum_{t=1}^{T}\mathbb{E}\left[\left\langle\hat{\ell}_t, \phi(\eta_t\hat{L}_t) - \phi(\eta_t(\hat{L}_t + \hat{\ell}_t))\right\rangle\right] + \sum_{t=1}^{T}\left(\frac{1}{\eta_{t+1}} - \frac{1}{\eta_t}\right)\mathbb{E}[r_{t+1,I_{t+1}} - r_{t+1,i^*}] + C_1$$

$$= \sum_{t=1}^{T}\mathbb{E}\left[\mathbb{E}\left[\left\langle\hat{\ell}_t, \phi(\eta_t\hat{L}_t) - \phi(\eta_t(\hat{L}_t + \hat{\ell}_t))\right\rangle + \left(\frac{1}{\eta_{t+1}} - \frac{1}{\eta_t}\right)(r_{t+1,I_{t+1}} - r_{t+1,i^*})\Big|\hat{L}_t\right]\right] + C_1$$

$$\le \sum_{t=1}^{T}\mathbb{E}\left[\mathbb{E}\left[\left\langle\hat{\ell}_t, \phi(\eta_t\hat{L}_t) - \phi(\eta_t(\hat{L}_t + \hat{\ell}_t))\right\rangle + \frac{r_{t+1,I_{t+1}} - r_{t+1,i^*}}{2m\sqrt{t}}\Big|\hat{L}_t\right]\right] + C_1 \qquad (30)$$

for $C_1 = \frac{\sqrt{K\pi}}{2m} + \left(\frac{2K}{27} + e^2\right)\log(T+1)$ and $\eta_t = m/\sqrt{t}$.

If $\hat{L}_t$ satisfies $D_t$ defined in (23), then the inner expectation is bounded by

$$\mathbb{E}\left[\left\langle\hat{\ell}_t, \phi(\eta_t\hat{L}_t) - \phi(\eta_t(\hat{L}_t + \hat{\ell}_t))\right\rangle + \frac{r_{t+1,I_{t+1}} - r_{t+1,i^*}}{2m\sqrt{t}}\Big|\hat{L}_t\right]$$

$$\le \sum_{i \ne i^*}\left(\frac{10e\sqrt{2}}{\underline{\hat{L}}_{t,i}} + \frac{13e^{1/4}}{(1-\zeta)}\frac{1}{\underline{\hat{L}}_{t,i}} + \frac{1}{m^2\underline{\hat{L}}_{t,i}}\right) + \frac{1}{1 - e^{-1}}(1 - e^{-1})^{\frac{\zeta}{\eta_t}}\left(\frac{\zeta}{\eta_t} + e\right)$$

$$\text{(by Lemmas C.2, C.3, and D.3)}$$

$$= \sum_{i \ne i^*}\frac{60 + m^{-2}}{\underline{\hat{L}}_{t,i}} + \frac{1}{1 - e^{-1}}(1 - e^{-1})^{\frac{\zeta}{\eta_t}}\left(\frac{\zeta}{\eta_t} + e\right), \qquad (31)$$

where we set $\zeta = 1 - (4e/21)^{1/3}$ following Honda et al. [28].

On the other hand, on $D_t^c$, the inner expectation can be bounded by

$$\mathbb{E}\left[\left\langle\hat{\ell}_t, \phi(\eta_t\hat{L}_t) - \phi(\eta_t(\hat{L}_t + \hat{\ell}_t))\right\rangle + \frac{r_{t+1,I_{t+1}} - r_{t+1,i^*}}{2m\sqrt{t}}\Big|\hat{L}_t\right]$$

$$\le 60m\sqrt{\frac{K\pi}{t}} + \frac{5.7}{2m}\sqrt{\frac{K}{t}} \qquad \text{(by Lemmas C.2 and C.3)}$$

$$\le (107m + 3/m)\sqrt{\frac{K}{t}}. \qquad (32)$$

By combining (31), (32) and Lemma D.2 with (30) and $\zeta = 1 - (4e/21)^{1/3}$, we obtain

$$\mathrm{Reg}(T) \le \sum_{t=1}^{T} \mathbb{E}\left[ \mathbb{1}[D_t]\frac{60 + m^{-2}}{\hat{\underline{L}}_{t,i}} + \mathbb{1}[D_t^c](107m + 3/m)\sqrt{\frac{K}{t}} \right] + C_1 + 2743m^2 + 77m. \quad (33)$$

On the other hand, by the lower bound given in Lemma D.1, we have

$$\mathrm{Reg}(T) \ge \sum_{t=1}^{T} \mathbb{E}\left[ \mathbb{1}[D_t]\frac{1}{8e^{5/4}} \sum_{i\neq i^*} \frac{t\Delta_i}{m^2 \hat{\underline{L}}_{t,i}^2} + \mathbb{1}[D_t^c]\frac{(1 - e^{-1/4})}{2e}\Delta \right]. \quad (34)$$

From $(33) - (34)/2$, we obtain

$$\frac{\mathrm{Reg}(T)}{2}$$

$$\le \sum_{t=1}^{T} \left[ \mathbb{1}[D_t] \sum_{i\neq i^*} \left( \frac{60 + m^{-2}}{\hat{\underline{L}}_{t,i}} - \frac{tm^{-2}\Delta_i}{16e^{5/4}\hat{\underline{L}}_{t,i}^2} \right) + \mathbb{1}[D_t^c]\left( (107m + 3/m)\sqrt{\frac{K}{t}} - \frac{1 - e^{-1/4}}{2e}\Delta \right) \right]$$
$$+ C_1 + 2743m^2 + 77m$$

$$\le \sum_{t=1}^{T} \left[ \mathbb{1}[D_t] \sum_{i\neq i^*} \frac{(60m + m^{-1})^2}{t\Delta_i/(4e^{5/4})} + \mathbb{1}[D_t^c]\left( (107m + 3/m)\sqrt{\frac{K}{t}} - \frac{1 - e^{-1/4}}{2e}\Delta \right) \right]$$
$$+ C_1 + 2743m^2 + 77m \qquad\qquad\qquad (\text{by } ax - bx^2 \le a^2/4b \text{ for } b > 0)$$

$$\le \sum_{t=1}^{T}\sum_{i\neq i^*} \frac{(60m + m^{-1})^2}{0.07t\Delta_i} + \sum_{t=1}^{T} \max\left\{ (107m + 3/m)\sqrt{K/t} - 0.04\Delta, 0 \right\} + C_1 + 2743m^2 + 77m$$

$$\le \sum_{i\neq i^*} \frac{(60m + m^{-1})^2(1 + \log T)}{0.07\Delta_i} + \frac{(107m + 3/m)^2 K}{0.04\Delta} + C_1 + 2743m^2 + 77m$$

$$= \Theta\left( \sum_{i\neq i^*} \frac{\log T}{\Delta_i} + K\log T + K \right),$$

which concludes the proof.

## E  Proofs of Corollary 4.2

Here, we provide the proof of Corollary 4.2, based on the discussion given in Appendices C and D. The main idea is to recast our arguments in the form analyzed in previous works [28, 38].

Firstly, as in (13), we can decompose the regret as follows:

$$\mathrm{Reg}(T)$$

$$\le \sum_{t=1}^{T} \mathbb{E}\left[ \left\langle \hat{\ell}_t, w_t - w_{t+1} \right\rangle \right] + \sum_{t=1}^{T} \left( \frac{1}{\eta_{t+1}} - \frac{1}{\eta_t} \right)\mathbb{E}[r_{t+1,I_{t+1}} - r_{t+1,i^*}] + \frac{\mathbb{E}_{r_1\sim\mathcal{U}_{\alpha,\beta}}[\max_{i\in[K]} r_{1,i}]}{\eta_1}$$

$$\le \sum_{t=1}^{T} \mathbb{E}\left[ \left\langle \hat{\ell}_t, w_t - w_{t+1} \right\rangle \right] + \sum_{t=1}^{T} \left( \frac{1}{\eta_{t+1}} - \frac{1}{\eta_t} \right)\mathbb{E}[r_{t+1,I_{t+1}} - r_{t+1,i^*}] + \frac{\mathbb{E}_{r_1\sim\mathcal{D}_{\alpha}}[\max_{i\in[K]} r_{1,i}]}{\eta_1}$$

$$\le \sum_{t=1}^{T} \mathbb{E}\left[ \left\langle \hat{\ell}_t, w_t - w_{t+1} \right\rangle \right] + \sum_{t=1}^{T} \left( \frac{1}{\eta_{t+1}} - \frac{1}{\eta_t} \right)\mathbb{E}[r_{t+1,I_{t+1}} - r_{t+1,i^*}] + \frac{M_u A_u \sqrt{K}}{m},$$

where the last inequality directly follows from Lemma 7 in Lee et al. [38] since $\mathcal{D}_{\alpha}$ is assumed to satisfy all the conditions.

## E.1 Adversarial regret analysis

As shown in Appendix C, it suffices to generalize Lemmas C.1–C.3. Here, the generalization of Lemma C.2 can be obtained by Theorem 4.1.

The generalization of Lemma C.1 is straightforward by discussion after (16), where the main difference comes from the additional ($\ddagger_i$) term induced by the negative parts. Specifically, the first two terms in (16) can be directly bounded by using Lemma 8 in Lee et al. [38], and as shown in (18), the sum of $\ddagger_i$ terms over all $i$ is zero for any distribution.

Similarly, the generalization of Lemma C.3 follows directly from Lemma 13 in Lee et al. [38]. The introduction of negative parts only decreases the expected value of perturbations, which in turn reduces the penalty term unless the expected value is positive, as shown in Appendix C.3.

## E.2 Stochastic regret analysis

To obtain BOBW guarantee, we further need to generalize Lemmas D.1 and D.3. For Lemma D.1, we can directly apply Lemma 22 in Lee et al. [38], since the lower bound can be obtained by restricting the integral interval to the positive side.

To generalize Lemma D.3, one can see that it suffices to bound the term

$$\dagger_i = \eta_t \int_{-\infty}^{0} f\left(z + \eta_t(\underline{\hat{L}}_{t,i} - x)\right) f(z) \mathrm{d}z,$$

when $x \le \zeta/\eta_t$ since the induced part by $\ddagger_i$ term in (27) can be analyzed by Lemma 25 in Lee et al. [38]. Then, we can decompose $\dagger_i$ into two terms by

$$\dagger_i = \underbrace{\eta_t \int_{-\infty}^{-\eta_t(\underline{\hat{L}}_{t,i} - x)} f\left(z + \eta_t(\underline{\hat{L}}_{t,i} - x)\right) f(z) \mathrm{d}z}_{\dagger_{i,1}} + \underbrace{\eta_t \int_{-\eta_t(\underline{\hat{L}}_{t,i} - x)}^{0} f\left(z + \eta_t(\underline{\hat{L}}_{t,i} - x)\right) f(z) \mathrm{d}z}_{\dagger_{i,2}}.$$

Since $f(x)$ is the density of $\mathcal{D}_\beta$ on $x < 0$, we have

$$\eta_t \int_{-\infty}^{-\eta_t(\underline{\hat{L}}_{t,i} - x)} f\left(z + \eta_t(\underline{\hat{L}}_{t,i} - x)\right) f(z) \mathrm{d}z \le \eta_t f(0) \int_{-\infty}^{-\eta_t(\underline{\hat{L}}_{t,i} - x)} f(z) \mathrm{d}z$$

$$= \eta_t f(0) F(-\eta_t(\underline{\hat{L}}_{t,i} - x))$$

$$\le \eta_t f(0) F(-\eta_t(1 - \zeta)\underline{\hat{L}}_{t,i})$$

$$= \eta_t f(0) \Theta\left(\frac{1}{(1 - \zeta)^\beta (\eta_t \underline{\hat{L}}_{t,i})^\beta}\right)$$

$$= f(0) \Theta\left(\frac{1}{(1 - \zeta)^\beta \underline{\hat{L}}_{t,i}}\right)$$

as desired. For the second term, we can do the similar derivations in (28) by considering the maximum of $\frac{1}{(z + c + 1)^\beta} \frac{1}{(z + 1)^\alpha}$.

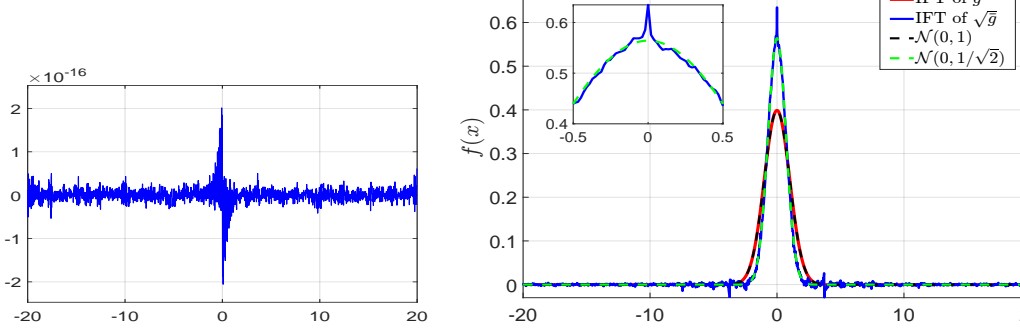

Figure 5: Imaginary part after IFT.

Figure 6: Sanity check with normal distribution.

## F  Details on numerical validation

In this section, we provide the Matlab code for the inverse Fourier transform (IFT) and additional plots to confirm whether the obtained IFT is real-valued. The code is implemented based on the code of cf2DistFFT in CharFunTool [53, 54].

```matlab
% Generate IFT given quantile function c(x).
x_min = -20.0; x_max = 20.0; % x-axis
N= 2^11; k = (0:(N-1))'; % number of samples
w = (0.5-N/2+k) * (2*pi / (x_max-x_min)); % frequency axis
quantile_func = @(p) -(p^(-1/2)-(1-p)^(-1/2)); % given c(x)
% compute characteristic function g
fun = @(w, p) exp(1i * w .* quantile_func(p));
cffun = @(w) arrayfun(@(w_val) sqrt(integral(@(p) fun(w_val, p)
    , 0.0001, 0.9999, 'ArrayValued', true)), w);
cf  = cffun(w(N/2+1:end));
cf  = [conj(cf(end:-1:1));cf]; % for the negative side
% Do inverse Fourier Transform
dx    = (x_max-x_min)/N;
C     = (-1).^((1-1/N)*(x_min/dx+k))/(x_max-x_min);
D     = (-1).^(-2*(x_min/(x_max-x_min))*k);
ifft = C.*fft(D.*cf);
pdf   = real(ifft); img_part = imag(ifft); cdf = cumsum(pdf*dx)
    ;
```

Note that in line 8, we compute the square root of the integral, i.e., the square root of the characteristic function $\bar{g}$. As illustrated in Figure 5, the imaginary part resulting from the IFT is negligible, with the y-axis scale on the order of $10^{-16}$, which can be attributed to numerical errors.

To validate the correctness of our code, we perform a sanity check by modifying the quantile function in the Matlab code to that of the standard normal distribution, given by $c(x) = \sqrt{2}\mathrm{erf}^{-1}(2x - 1)$, where $\mathrm{erf}^{-1}$ denotes the inverse error function. Since the characteristic function of $\mathcal{N}(0,\sigma)$ is $\exp(-\sigma^2 t^2/2)$, the characteristic function of $\mathcal{N}(0,1)$ is $\bar{g}(t) = \exp(-t^2/2)$. Therefore, applying the IFT to $\sqrt{\bar{g}}$ should yield the density of $\mathcal{N}(0, 1/\sqrt{2})$. As shown in Figure 6, the result confirms the correctness of our implementation.

## G  Proofs of results under symmetric perturbations

Here, we provide proofs for results in Section 5.

### G.1  Proof of Lemma 5.1

**Lemma 5.1 (restated)** *Let* $-\mathcal{D}$ *denote the distribution of* $-X$. *Let* $(r_1, r_2)$ *be i.i.d. from continuous* $\mathcal{D}$ *and* $\bar{\mathcal{D}}$ *be distribution of* $r_1 - r_2$. *Then, if* $\bar{\mathcal{D}}$ *is Fréchet-type, then either* $\mathcal{D}$ *or* $-\mathcal{D}$ *is Fréchet-type.*

*Proof.* According to the Fisher-Tippett-Gnedenko (FTG) theorem [20, 23], if the distribution of normalized maximum converges, such limit distributions must belong to one of three types of extreme value distributions: Fréchet, Gumbel, and Weibull. Distributions with polynomial tails (i.e., heavy tails) are classified as Fréchet-type distributions, while distributions with exponential tails are Gumbel-type, and bounded tails (i.e., bounded maximum) belong to either the Gumbel-type or the Weibull-type [47].

Whenever the tails of distribution of $X$ are either exponential or bounded, the distribution of $X - Y$ also has either an exponential right tail or a bounded right tail. It is straightforward to verify this when the distributions have bounded tails. In the case where both distributions have exponential tails, classical results on convolution tails indicate that their convolution also exhibits an exponential tail, where the general proof can be found in Cline [14] and references therein. This implies that if tails of $\mathcal{D}_\beta^{\mathrm{Ts}}$ are either Gumbel-type or Weibul-type, $\mathcal{D}_\beta^{\mathrm{Ts}}$ does not have a heavy-tail, meaning that it is not Fréchet-type. $\qquad\square$

### G.2 Proof of Proposition 5.3

Here, we assume $\lambda_1 \leq \lambda_2$, so that $\underline{\lambda}_1 = 0$ and $\underline{\lambda}_2 = c$ for some constants $c > 0$ without loss of generality. For $i = 1$, by the unimodality of symmetric Pareto distribution, we have

$$
\begin{aligned}
\frac{-\phi_1'(\lambda)}{(\phi_1(\lambda))^{3/2}} &= \frac{\int_{-\infty}^{\infty} -f'(z)F(z+c)\mathrm{d}z}{\left(\int_{-\infty}^{\infty} f(z)F(z+c)\mathrm{d}z\right)^{3/2}} \\
&\leq \frac{\int_{0}^{\infty} -f'(z)F(z+c)\mathrm{d}z}{\left(\int_{-\infty}^{\infty} f(z)F(z+c)\mathrm{d}z\right)^{3/2}} && \text{(unimodality of } f) \\
&\leq 2\sqrt{2} \int_{0}^{\infty} -f'(z)F(z+c)\mathrm{d}z && (\phi_1(\underline{\lambda}) \geq 1/2) \\
&= 2\sqrt{2} \int_{0}^{\infty} \frac{3}{(z+1)^4}\left(1 - \frac{1}{2(z+c+1)^2}\right)\mathrm{d}z \\
&\leq 2\sqrt{2}. && (35)
\end{aligned}
$$

For $i = 2$, we have

$$
\frac{-\phi_2'(\lambda)}{(\phi_2(\lambda))^{3/2}} = \frac{\int_{-\infty}^{\infty} -f'(z+c)F(z)\mathrm{d}z}{\left(\int_{-\infty}^{\infty} f(z+c)F(z)\mathrm{d}z\right)^{3/2}}.
$$

For the denominator term, we have

$$
\begin{aligned}
\int_{-\infty}^{\infty} f(z+c)F(z)\mathrm{d}z &\geq \int_{0}^{\infty} f(z+c)F(z)\mathrm{d}z \\
&\geq \frac{1}{2}\int_{0}^{\infty} \frac{1}{(z+c+1)^3}\mathrm{d}z \\
&= \frac{1}{4(c+1)^2}. && (36)
\end{aligned}
$$

For the numerator term, we consider two cases separately, where (i) $c \in (0, 1.5)$ and (ii) $c \geq 1.5$. When $c \in (0, 1.5)$, we have

$$
\int_{-\infty}^{\infty} f(z+c)F(z)\mathrm{d}z \geq \frac{1}{2^2(2.5)^2}
$$

and

$$
\begin{aligned}
\int_{-\infty}^{\infty} -f'(z+c)F(z)\mathrm{d}z &= \int_{-\infty}^{\infty} -f'(z)F(z-c)\mathrm{d}z \\
&\leq \int_{0}^{\infty} -f'(z)F(z-c)\mathrm{d}z \\
&\leq \int_{0}^{\infty} -f'(z)F(z+c)\mathrm{d}z \leq 1. && \text{(by (35))}
\end{aligned}
$$

Therefore, we have

$$\frac{-\phi_2'(\lambda)}{(\phi_2(\lambda))^{3/2}} \leq 125.$$

When $c \geq 1.5$, we decompose the numerator into three terms.

$$\int_{-\infty}^{\infty} -f'(z+c)F(z)\mathrm{d}z = \underbrace{\int_{-\infty}^{-c} \frac{-3}{2(1-z-c)^4(1-z)^2}\mathrm{d}z}_{\dagger_1}$$

$$+ \underbrace{\int_{-c}^{0} \frac{3}{2(z+c+1)^4(1-z)^2}\mathrm{d}z}_{\dagger_2} + \underbrace{\int_{0}^{\infty} \frac{3}{(z+c+1)^4}\left(1 - \frac{1}{2(z+1)^2}\right)\mathrm{d}z}_{\dagger_3}$$

For the last term, we have

$$\dagger_3 \leq \int_{0}^{\infty} \frac{3}{(c+z+1)^4}\mathrm{d}z = \frac{1}{(c+1)^3}. \tag{37}$$

For the first term, by partial fraction decomposition, we obtain

$$\frac{1}{(1-z-c)^4(1-z)^2} = \frac{1/c^2}{(1-z-c)^4} + \frac{-2/c^3}{(1-z-c)^3} + \frac{3/c^4}{(1-z-c)^2} + \frac{-4/c^5}{1-z-c} + \frac{4/c^5}{1-z} + \frac{1/c^4}{(1-z)^2}.$$

Therefore, we have

$$-\frac{2}{3}\dagger_1 = \int_{-\infty}^{-c} \frac{1/c^2}{(1-z-c)^4} + \frac{-2/c^3}{(1-z-c)^3} + \frac{3/c^4}{(1-z-c)^2} + \frac{-4/c^5}{1-z-c} + \frac{4/c^5}{1-z} + \frac{1/c^4}{(1-z)^2}\mathrm{d}z$$

$$= \frac{1/c^2}{3(1-z-c)^3} + \frac{-1/c^3}{(1-z-c)^2} + \frac{3/c^4}{(1-z-c)} + \frac{4}{c^5}\log\left(\frac{1-z-c}{1-z}\right) + \frac{1/c^4}{1-z}\Big|_{z=-\infty}^{z=-c}$$

$$= \frac{1}{3c^2} - \frac{1}{c^3} + \frac{3}{c^4} - \frac{4}{c^5}\log(c+1) + \frac{1}{c^4(c+1)},$$

which implies

$$\dagger_1 = -\frac{1}{2c^2} + \frac{3}{2c^3} - \frac{9}{2c^4} + \frac{6}{c^5}\log(c+1) - \frac{3}{2c^4(c+1)}$$

$$\leq -\frac{1}{2c^2} + \frac{3}{2}\left(\frac{1}{c^3} + \frac{1}{c^4}\right). \qquad \text{(by } \log(c+1) \leq c\text{)}$$

Similarly, for the second term, partial fraction decomposition gives

$$\frac{1}{(z+c+1)^4(1-z)^2} = \frac{1/(c+2)^2}{(z+c+1)^4} + \frac{2/(c+2)^3}{(z+c+1)^3} + \frac{3/(c+2)^4}{(z+c+1)^2} + \frac{4/(c+2)^5}{z+c+1} + \frac{4/(c+2)^5}{1-z} + \frac{1/(c+2)^4}{(1-z)^2}.$$

Therefore, we have

$$\frac{2}{3}\dagger_2 = \frac{-1/(c+2)^2}{3(z+c+1)^3} + \frac{-1/(c+2)^3}{(z+c+1)^2} + \frac{-3/(c+2)^4}{z+c+1} + \frac{4}{(c+2)^5}\log\left(\frac{z+c+1}{1-z}\right) + \frac{1/(c+2)^4}{1-z}\Big|_{z=-c}^{z=0}$$

$$= \frac{1}{3(c+2)^2} + \frac{1}{(c+2)^3} + \frac{4}{(c+2)^4} + \frac{8\log(c+1)}{(c+2)^5}$$

$$- \left(\frac{1}{3(c+1)^3(c+2)^2} + \frac{1}{(c+1)^2(c+2)^3} + \frac{4}{(c+1)(c+2)^4}\right),$$

which implies for $c > 0$

$$\dagger_2 = \frac{1}{2(c+2)^2} + \frac{3}{2(c+2)^3} + \frac{6}{(c+2)^4} + \frac{12\log(c+1)}{(c+2)^5}$$

$$- \frac{3}{2}\left(\frac{1}{3(c+1)^3(c+2)^2} + \frac{1}{(c+1)^2(c+2)^3} + \frac{4}{(c+1)(c+2)^4}\right)$$

$$\leq \frac{1}{2c^2} + \frac{3}{2(c+2)^3} + \frac{18}{(c+2)^4} - \frac{3}{2}\left(\frac{1}{3(c+1)^3(c+2)^2} + \frac{1}{(c+1)^2(c+2)^3} + \frac{4}{(c+1)(c+2)^4}\right).$$

Therefore,

$$
\dagger_1 + \dagger_2 \le \frac{3}{2c^3} + \frac{3}{2(c+2)^3} + \frac{3}{2c^4} + \frac{18}{(c+2)^4}
$$

$$
- \frac{3}{2}\left( \frac{1}{3(c+1)^3(c+2)^2} + \frac{1}{(c+1)^2(c+2)^3} + \frac{4}{(c+1)(c+2)^4} \right) \quad (38)
$$

By combining (36) and (37) with (38), we have for any $c > 0$ that

$$
\frac{-\phi_2'(\lambda)}{(\phi_2(\lambda))^{3/2}} \le 12\left( \left(\frac{c+1}{c}\right)^3 + \left(\frac{c+1}{c+2}\right)^3 + \frac{(c+1)^3}{c^4} \right)
$$

$$
+ \frac{144(c+1)^3}{(c+2)^4} - \frac{4}{(c+2)^2} + \frac{12(c+1)}{(c+2)^2} - \frac{48(c+1)^2}{(c+2)^4} + 8.
$$

Then, for $c \ge 1.5$, we have

$$
\frac{-\phi_2'(\lambda)}{(\phi_2(\lambda))^{3/2}} \le 121,
$$

which concludes the proof.

### G.3   Proof of Proposition 5.5

Consider symmetric perturbation distributions $\mathcal{U}_{2,2}$, whose tails of both sides are Fréchet-type with index 2. Let $\mathcal{D}_{2,l}$ and $\mathcal{D}_{2,r}$ denote the distribution of left and right sides, respectively. Here, we assume that for all $x > 0$

$$
\frac{xf(x;\mathcal{D}_{2,l})}{1 - F(x;\mathcal{D}_{2,l})} \le 2 \quad \text{and} \quad \frac{xf(x;\mathcal{D}_{2,r})}{1 - F(x;\mathcal{D}_{2,r})}.
$$

This condition is known as a sufficient condition to satisfy Assumption 2.3. Also, it is not that restrictive conditions as several well-known Fréchet-type distributions, such as Fréchet, Pareto, and Student-$t$ distributions, satisfy this condition [38, see Appendix A]. Under this assumption, we have for $x > 0$

$$
F(-x;\mathcal{U}_{2,2}) = \frac{S_{F_l}(x)}{2(x+1)^2} \quad \text{and} \quad 1 - F(x;\mathcal{U}_{2,2}) = \frac{S_{F_r}(x)}{2(x+1)^2},
$$

where the corresponding slowly varying function $S_F(x)$ is an increasing function for $x > 0$. Here, a slowly varying function $g(x)$ satisfies $\lim_{x\to\infty} \frac{g(x)}{x^a} = 0$ for any $a > 0$, so that it is asymptotically negligible compared to any polynomial function.

By definition of $\phi$, we have for $i \ne 1$

$$
\phi_i(\lambda) = \int_{-\infty}^{\infty} f(z + \underline{\lambda}_i)F(z) \prod_{j \ne i,1} F(z + \underline{\lambda}_j)\mathrm{d}z.
$$

Then, we obtain for $K \ge 3$ and $\underline{\lambda} = (0, c\ldots, c)$ with $c > 0$ that

$$
-\phi_i'(\lambda) = \int_{-\infty}^{\infty} -f'(z + c)F(z)F^{K-2}(z + c)\mathrm{d}z
$$

$$
= \int_{-\infty}^{\infty} -f'(z)F(z - c)F^{K-2}(z)\mathrm{d}z
$$

$$
= \int_{-\infty}^{\infty} (K - 2)F(z - c)f^2(z)F^{K-3}(z)\mathrm{d}z + \int_{-\infty}^{\infty} f(z - c)f(z)F^{K-2}(z)\mathrm{d}z
$$

$$
\ge \int_{-\infty}^{\infty} (K - 2)F(z - c)f^2(z)F^{K-3}(z)\mathrm{d}z
$$

$$
\ge \frac{K - 2}{K - 1}\int_{-\infty}^{\infty} F(z - c)\frac{f(z)}{F(z)} \cdot (K - 1)f(z)F^{K-2}(z)\mathrm{d}z
$$

$$
\ge \frac{K - 2}{K - 1}\int_{\sqrt{K}-1}^{2\sqrt{K}-1} F(z - c)\frac{f(z)}{F(z)} \cdot (K - 1)f(z)F^{K-2}(z)\mathrm{d}z.
$$

Then, since $S_F$ is increasing, we have for $c \geq 2\sqrt{K}$

$$F(z - c) \geq F(\sqrt{K} - 1 - c) = \frac{S_{F_l}(c + 1 - \sqrt{K})}{2(c - \sqrt{K})^2} \geq \frac{S_{F_l}(c + 1 - \sqrt{K})}{2c^2} \geq \frac{S_{F_l}(\sqrt{K} + 1)}{2c^2}$$

uniformly over $z \in [\sqrt{K} - 1, 2\sqrt{K} - 1]$.

By Assumption 2.3, $f/F$ is decreasing for $x > 0$. Therefore, we have for $x \in [\sqrt{K} - 1, 2\sqrt{K} - 1]$ that

$$\frac{f(x)}{F(x)} \geq \frac{f(2\sqrt{K} - 1)}{F(2\sqrt{K} - 1)} \geq \frac{S_{f_r}(2\sqrt{K} - 1)}{4K\sqrt{K}}$$

since $F(2\sqrt{K} - 1) \geq 1/2$ and $f(x) = S_{f_r}(x)/(x + 1)^3$. Therefore, we obtain

$$\frac{K - 2}{K - 1} \int_{\sqrt{K} - 1}^{2\sqrt{K} - 1} F(z - c)\frac{f(z)}{F(z)} \cdot (K - 1)f(z)F^{K-2}(z)\mathrm{d}z$$

$$\geq \frac{(K - 2)S_{F_l}(\sqrt{K} + 1)}{2(K - 1)c^2} \int_{\sqrt{K} - 1}^{2\sqrt{K} - 1} \frac{f(z)}{F(z)} \cdot (K - 1)f(z)F^{K-2}(z)\mathrm{d}z$$

$$\geq \frac{(K - 2)S_{F_l}(\sqrt{K} + 1)S_{f_r}(2\sqrt{K} - 1)}{8K\sqrt{K}(K - 1)c^2} \int_{\sqrt{K} - 1}^{2\sqrt{K} - 1} (K - 1)f(z)F^{K-2}(z)\mathrm{d}z$$

$$\geq \frac{(K - 2)S_{F_l}(\sqrt{K} + 1)S_{f_r}(2\sqrt{K} - 1)}{8K\sqrt{K}(K - 1)c^2} F^{K-1}(2\sqrt{K} - 1)$$

$$\geq \frac{S_{F_l}(\sqrt{K} + 1)S_{f_r}(2\sqrt{K} - 1)}{16K\sqrt{K}c^2} F^{K-1}(2\sqrt{K} - 1). \qquad (\because K \geq 3)$$

Here, for $y = 2\sqrt{K} - 1$, we obtain

$$F^{K-1}(y) = (1 - (1 - F(y)))^{K-1} = \exp((K - 1)\log(1 - (1 - F(y))))$$

$$\geq \exp\left(-(K - 1)\frac{1 - F(y)}{F(y)}\right) \qquad (39)$$

$$\geq \exp(-2(K - 1)(1 - F(y))) \quad (\because F(x) \geq 1/2 \text{ for } x \geq 0)$$

$$= \exp\left(-2(K - 1)\frac{S_F(y)}{(y + 1)^2}\right)$$

$$\geq \exp\left(-S_F(2\sqrt{K} - 1)/2\right), \qquad (\because y = 2\sqrt{K} - 1)$$

where (39) follows from $\log(1 - x) > \frac{-x}{1-x}$ for $x \in (0, 1]$. This implies

$$-\phi_i'(\underline{\lambda}) \geq \frac{S_{F_l}(\sqrt{K} + 1)S_{f_r}(2\sqrt{K} - 1)}{16K\sqrt{K}c^2} \exp\left(-S_{F_r}(2\sqrt{K} - 1)/2\right) =: \frac{C(\mathcal{D}, K)}{16K\sqrt{K}c^2}. \qquad (40)$$

Here, $C(\mathcal{D}, K)$ is a constant that only depends on the distribution and $K$. Even though $C(\mathcal{D}, K)$ depends on $K$, it is important to note that its dependency is at most logarithmic order since both $S_F$ and $S_f$ are slowly varying functions. For example, $C(\mathcal{D}, K) = 2\exp(-1/2)$ holds when $\mathcal{D}$ is a symmetric Pareto with shape 2.

Next, we have

$$\phi_i(\underline{\lambda}) = \int_{-\infty}^{\infty} f(z + c)F(z)F^{K-2}(z + c)\mathrm{d}z$$

$$= \underbrace{\int_{-\infty}^{-c} f(z + c)F(z)F^{K-2}(z + c)\mathrm{d}z}_{\ddagger_1} + \underbrace{\int_{-c}^{0} f(z + c)F(z)F^{K-2}(z + c)\mathrm{d}z}_{\ddagger_2}$$

$$+ \underbrace{\int_{0}^{\infty} f(z + c)F(z)F^{K-2}(z + c)\mathrm{d}z}_{\ddagger_3}.$$

Then, we have

$$\ddagger_1 \le F(-c) \int_{-\infty}^{-c} f(z+c) F^{K-2}(z+c) \mathrm{d}z = \frac{F(-c)}{K-1} \frac{1}{2^{K-1}}.$$

For the second term, we obtain

$$\ddagger_2 = \int_0^c f(z) F(z-c) F^{K-2}(z) \mathrm{d}z$$

$$= \int_0^{c/2} f(z) F(z-c) F^{K-2}(z) \mathrm{d}z + \int_{c/2}^c f(z) F(z-c) F^{K-2}(z) \mathrm{d}z$$

$$\le F(-c/2) \int_0^{c/2} f(z) F^{K-2}(z) \mathrm{d}z + \int_{c/2}^\infty f(z) \mathrm{d}z \le 2F(-c/2),$$

where the last inequality follows from symmetry so that $1 - F(z) = F(-z)$ for $z > 0$. For the last term, we have

$$\ddagger_3 \le \int_0^\infty f(z+c) F(z) F^{K-2}(z+c) \mathrm{d}z \le \int_0^\infty f(z+c) \le 1 - F(c) = F(-c).$$

Therefore, we have for $K \ge 3$ that

$$\phi_i(\underline{\lambda}) \le \frac{F(-c)}{8} + 2F(-c/2) + F(-c) \le \frac{25}{8} F(-c/2)$$

$$= \frac{25}{8} \frac{S_{F_l}(c/2)}{(c/2+1)^2} = \frac{25}{2} \frac{S_{F_l}(c/2)}{(c+2)^2}. \tag{41}$$

By combining (40) and (41), we have for $c \ge 2\sqrt{K}$ that

$$\frac{-\phi_i'(\underline{\lambda})}{\phi_i(\underline{\lambda})} \ge \frac{2(c+2)^2}{25 S_{F_l}(c/2)} \frac{C(\mathcal{D}, K)}{16 K \sqrt{K} c^2} = \tilde{\Omega}\left(\frac{1}{K\sqrt{K}}\right),$$

where $\tilde{\Omega}$ hides any logarithmic dependency. Note that Assumption 2.3 implies $\limsup_{x \to \infty} S_F(x) < \infty$. Moreover,

$$\frac{-\phi_i'(\underline{\lambda})}{\phi_i^{3/2}(\underline{\lambda})} \ge \tilde{\Omega}\left(\frac{c+2}{K\sqrt{K}}\right),$$

which concludes the proof.

### G.4   Proof of Proposition 5.6

Since $\varphi(\nu)$ is a bijective function, there exists a unique $c(x)$ satisfying

$$\varphi((c(x), 0, 0)) = \left(x, \frac{1-x}{2}, \frac{1-x}{2}\right)$$

for any $x \in [\frac{1}{3}, 1)$. When $x < \frac{1}{3}$, there exists a unique $c(x)$ satisfying

$$\varphi((0, c(x), c(x))) = \left(x, \frac{1-x}{2}, \frac{1-x}{2}\right)$$

Note that $c(1/3) = 0$ holds since the perturbations are independently identically distributed. Let us define another function $\bar{V} : (0,1) \to \mathbb{R}_{\le 0}$ satisfying $\bar{V}(x) = V((x, (1-x)/2, (1-x)/2)$. From (3), it holds that

$$\bar{V}(x) = \begin{cases} xc(x) - \Phi((c(x), 0, 0)), & \text{if } x \ge 1/3 \\ (1-x)c(x) - \Phi((0, c(x), c(x))) & \text{if } x \le 1/3 \end{cases}.$$

Let us consider $x \ge 1/3$ case, where $\arg\max_i \nu_i = 1$ as Proposition 5.3. By explicitly considering the potential function in (9), we have

$$\Phi((c(x), 0, 0)) = \int_{-\infty}^\infty zf(z - c(x)) F^2(z) \mathrm{d}z + 2 \int_{-\infty}^\infty zf(z) F(z) F(z - c(x)) \mathrm{d}z.$$

Therefore, by relationship between regularization function and potential function given in (10), for $x \geq 1/3$, we have

$$\bar{V}'(x) = c(x) + xc'(x) - c'(x)\left(\int_{-\infty}^{\infty} zf'(z - c(x))F^2(z)dz + \int_{-\infty}^{\infty} zf(z)f(z - c(x))F(z)dz\right)$$

$$= c(x) + xc'(x) - c'(x)\varphi_1((c(x), 0, 0)) \qquad \text{(by (11))}$$

$$= c(x).$$

Thus, the derivative of $\bar{V}(x)$ depends on $c(x)$, which is related to the inverse function of $\varphi$.

Fix $c \geq 0$. Then, it holds that

$$\varphi_1((c, 0, 0)) = 1 - 2\varphi_2((c, 0, 0)).$$

From (42), it holds that

$$\varphi_2((c, 0, 0)) = \phi_2((0, c, c))$$

$$\leq \frac{3 + 1/16}{(c + 1)^2} \leq \frac{4}{(c + 1)^2}.$$

On the other hand,

$$\phi_2((0, c, c)) = \int_{-\infty}^{\infty} f(z + c)F(z)F(z + c)dz$$

$$\geq \int_{-c}^{0} f(z + c)F(z)F(z + c)dz + \int_{0}^{\infty} f(z + c)F(z)F(z + c)dz$$

$$\geq F(-c)\int_{-c}^{0} f(z + c)F(z + c)dz + F(0)\int_{0}^{\infty} f(z + c)F(z + c)dz$$

$$= \frac{1}{2}\left(F^2(c)F(-c) - \frac{F(-c)}{4} + \frac{1}{2}(1 - F^2(c))\right)$$

$$\geq \frac{1}{2}\left(F^2(c)F(-c) - \frac{F(-c)}{4} + \frac{1}{2}(1 - F(c))\right) \qquad (F(c) \leq 1)$$

$$= \frac{1}{2}\left(F^2(c)F(-c) - \frac{F(-c)}{4} + \frac{F(-c)}{2}\right)$$

$$\geq \frac{F(-c)}{4}. \qquad (F(c) \in (1/2, 1])$$

Therefore, we obtain

$$\frac{1}{8(c + 1)^2} \leq \varphi_2((c, 0, 0)) \leq \frac{4}{(c + 1)^2},$$

which implies that for $x \in [1/3, 1)$

$$1 - \frac{8}{(c(x) + 1)^2} \leq x \leq 1 - \frac{1}{4(c(x) + 1)^2}.$$

Hence, we obtain

$$c(x) = \Theta\left(\frac{1}{\sqrt{1 - x}}\right) - 1.$$

Note that the derivative of $\frac{1}{2}$-Tsallis entropy at $p = \left(x, \frac{1-x}{2}, \frac{1-x}{2}\right)$ is

$$V'_{1/2}(p) = -\frac{1}{\sqrt{x}} + \frac{\sqrt{2}}{\sqrt{1 - x}},$$

which implies that the derivative of $V(p)$ roughly coincides with that of $\frac{1}{2}$-Tsallis entropy when $x \to 1$.

### G.5 Counterexample for general index in three-armed bandits

**Proposition G.1.** *Let the unimodal symmetric Fréchet-type perturbation $\mathcal{D}_\alpha$ with index $\alpha > 1$ defined on $\mathbb{R}$, which is equivalent to a distribution in $\mathfrak{D}_\alpha$ on $\mathbb{R}_+$, and*

$$f'(z; \mathcal{D}_\alpha) \leq 0, \ \forall z \geq 0, \qquad \text{(Unimodality)}$$

$$F(-z; \mathcal{D}_\alpha) = 1 - F(z; \mathcal{D}_\alpha) = \Theta\left(\frac{1}{(z+1)^\alpha}\right), \ \forall z \geq 0 \ \text{(Symmetric Fréchet-type)}$$

$$\int_{-c}^0 f(z + c; \mathcal{D}_\alpha) F(z; \mathcal{D}_\alpha) F(z + c; \mathcal{D}_\alpha) \mathrm{d}z \leq C(\mathcal{D}_\alpha) F(-c; \mathcal{D}_\alpha), \ \forall c > 0. \qquad \text{(Condition)}$$

*for a constant $C(\mathcal{D}_\alpha)$ that depends only on the distribution and is independent of c. When $K = 3$ and $\lambda \in \mathbb{R}_+^3$ satisfies $\underline{\lambda} = (0, c, c)$ for some $c > 0$, then for $i \neq 1$, it holds that*

$$\frac{-\phi_i'(\lambda)}{\phi_i(\lambda)} \geq \Omega\left(\frac{1}{(\alpha + 1)C(\mathcal{D}_\alpha)}\right), \quad \text{and} \quad \frac{-\phi_i'(\lambda)}{\phi_i^{3/2}(\lambda)} \geq \Omega\left(\frac{(c+1)^{\alpha/2}}{(\alpha + 1)C(\mathcal{D}_\alpha)^{3/2}}\right).$$

*Proof.* By definition of $\phi$, for any $\lambda \in [0, \infty)^K$, we have

$$\frac{-\phi_i'(\lambda)}{\phi_i(\lambda)} = \frac{\int_{-\infty}^\infty -f'(z + \underline{\lambda}_i) \prod_{j \neq i} F(z + \underline{\lambda}_j) \mathrm{d}z}{\int_{-\infty}^\infty f(z + \underline{\lambda}_i) \prod_{j \neq i} F(z + \underline{\lambda}_j) \mathrm{d}z}.$$

Let us consider $K = 3$ and $\underline{\lambda} = (0, c, c)$ for some $c > 0$. Then, for $i \neq 1$, the numerator is written as

$$\int_{-\infty}^\infty -f'(z + \underline{\lambda}_i) \prod_{j \neq i} F(z + \underline{\lambda}_j) \mathrm{d}z$$

$$= \int_{-\infty}^\infty -f'(z + c) F(z) F(z + c) \mathrm{d}z$$

$$= \underbrace{\int_{-\infty}^{-c} -f'(z + c) F(z) F(z + c) \mathrm{d}z}_{\dagger_1} + \underbrace{\int_{-c}^0 -f'(z + c) F(z) F(z + c) \mathrm{d}z}_{\dagger_2}$$

$$+ \underbrace{\int_0^\infty -f'(z + c) F(z) F(z + c) \mathrm{d}z}_{\dagger_3}.$$

For the first term, since $f'(z + c) \geq 0$ for $z \leq -c$, we have

$$\dagger_1 = \int_{-\infty}^{-c} -f'(z + c) F(z) F(z + c) \mathrm{d}z$$

$$\geq F(-c) \int_{-\infty}^{-c} -f'(z + c) F(z + c) \mathrm{d}z = F(-c) \int_{-\infty}^0 -f'(z) F(z) \mathrm{d}z.$$

For the second term, since $f'(z + c) \leq 0$ for $z \geq -c$, we have

$$\dagger_2 = \int_{-c}^0 -f'(z + c) F(z) F(z + c) \mathrm{d}z$$

$$\geq F(-c) \int_{-c}^0 -f'(z + c) F(z + c) \mathrm{d}z = F(-c) \int_0^c -f'(z) F(z) \mathrm{d}z.$$

For the third term, we have

$$\dagger_3 = \int_0^\infty -f'(z + c) F(z) F(z + c) \mathrm{d}z$$

$$\geq F(-c) \int_0^\infty -f'(z + c) F(z + c) \mathrm{d}z = F(-c) \int_c^\infty -f'(z) F(z) \mathrm{d}z.$$

Therefore, we obtain

$$\int_{-\infty}^{\infty} -f'(z+\underline{\lambda}_i) \prod_{j\neq i} F(z+\underline{\lambda}_j) \mathrm{d}z \geq F(-c) \int_{-\infty}^{\infty} -f'(z)F(z)\mathrm{d}z$$

$$= F(-c) \int_{-\infty}^{\infty} f^2(z)\mathrm{d}z$$

$$= 2F(-c) \int_{0}^{\infty} f^2(z)\mathrm{d}z \qquad \text{(symmetry)}$$

$$= 2F(-c) \int_{0}^{\infty} \left( \frac{S_f(z)}{(z+1)^{\alpha+1}} \right)^2 \mathrm{d}z,$$

where $S_f(z)$ denotes the corresponding slowly varying function. Note that the slowly varying function is a function $g$ satisfying

$$\lim_{x\to\infty} \frac{g(tx)}{g(x)} = 1, \ \forall t > 0,$$

which implies $S_f(z) = o(z^a)$ for any $a > 0$. Moreover, by Assumption A.5, $\liminf_{z\to\infty} S_f(z) > 0$ holds (refer to Lee et al. [38, Appendix A] for more details). Here, $S_f(z) > 0$ holds for all $z \geq 0$ since we consider the unimodal symmetric distribution. Therefore, there exists a constant $c_f > 0$ such that $S_f(z) \geq c_f$ for all $z \geq 0$. Hence, we have

$$2F(-c) \int_{0}^{\infty} \left( \frac{S_f(z)}{(z+1)^{\alpha+1}} \right)^2 \mathrm{d}z \geq 2F(-c) \int_{0}^{\infty} \left( \frac{c_f}{(z+1)^{\alpha+1}} \right)^2 \mathrm{d}z$$

$$= \frac{2c_f^2 F(-c)}{2\alpha+1} = \Theta\left( \frac{1}{(\alpha+1)(c+1)^\alpha} \right),$$

since $F$ is the distribution function of symmetric Fréchet-type with $F(-z) = 1 - F(z) = \Theta\left( \frac{1}{(z+1)^\alpha} \right)$ for $z > 0$.

For the denominator, for $i \neq 1$, we have

$$\int_{-\infty}^{\infty} f(z+\underline{\lambda}_i) \prod_{j\neq i} F(z+\underline{\lambda}_j) \mathrm{d}z$$

$$= \int_{-\infty}^{\infty} f(z+c)F(z)F(z+c)\mathrm{d}z$$

$$= \underbrace{\int_{-\infty}^{-c} f(z+c)F(z)F(z+c)\mathrm{d}z}_{\ddagger_1} + \underbrace{\int_{-c}^{0} f(z+c)F(z)F(z+c)\mathrm{d}z}_{\ddagger_2}$$

$$+ \underbrace{\int_{0}^{\infty} f(z+c)F(z)F(z+c)\mathrm{d}z}_{\ddagger_3}.$$

For the first term, we obtain

$$\ddagger_1 = \int_{-\infty}^{-c} f(z+c)F(z)F(z+c)\mathrm{d}z$$

$$\leq F(-c) \int_{-\infty}^{-c} f(z+c)F(z+c)\mathrm{d}z$$

$$\leq \frac{F(-c)}{2} \int_{-\infty}^{-c} f(z+c)\mathrm{d}z = \frac{F(-c)}{4}.$$

The second term is directly bounded by condition, i.e., $\ddagger_2 \le C(\mathcal{D}_\alpha)F(-c)$. For the third term, we obtain

$$\ddagger_3 = \int_0^\infty f(z+c)F(z)F(z+c)\mathrm{d}z$$
$$\le \int_0^\infty f(z+c)\mathrm{d}z = 1 - F(c) = F(-c).$$

Therefore,

$$\int_{-\infty}^\infty f(z+\underline{\lambda}_i)\prod_{j\neq i}F(z+\underline{\lambda}_j)\mathrm{d}z \le F(-c)(C(\mathcal{D}_\alpha)+5/4).$$

Therefore, we obtain

$$\frac{-\phi_i'(\lambda)}{\phi_i(\lambda)} = \frac{\dagger_1 + \dagger_2 + \dagger_3}{\ddagger_1 + \ddagger_2 + \ddagger_3} \ge \frac{2c_f^2}{(2\alpha+1)(C(\mathcal{D}_\alpha)+5/4)} = \Omega(1)$$

and

$$\frac{-\phi_i'(\lambda)}{\phi_i^{3/2}(\lambda)} = \frac{\dagger_1 + \dagger_2 + \dagger_3}{\ddagger_1 + \ddagger_2 + \ddagger_3} \ge \frac{2c_f^2}{(2\alpha+1)(C(\mathcal{D}_\alpha)+5/4)^{3/2}\sqrt{F(-c)}} = \Omega\Big((c+1)^{\alpha/2}\Big).$$

$\square$

### G.6 Specific results for symmetric Pareto distribution

Here, we explicitly substitute the density function and distribution function of symmetric Pareto distribution.

**Proposition G.2.** *Let the perturbations be i.i.d. from the symmetric Pareto distribution with shape* $2$ *considered in Proposition 5.3. When $K = 3$ and $\lambda \in \mathbb{R}_+^3$ satisfies $\underline{\lambda} = (0,c,c)$ for some $c > 0$, then for $i \neq 1$, it holds that*

$$\frac{-\phi_i'(\lambda)}{\phi_i(\lambda)} \ge \frac{1}{31} \quad and \quad \frac{-\phi_i'(\lambda)}{\phi_i^{3/2}(\lambda)} \ge \frac{c+1}{11}.$$

*Proof.* Let us use the same notation used in the proof of Proposition G.1 For the first term of the numerator, we obtain

$$\dagger_1 = \int_{-\infty}^{-c} \frac{-3}{(1-z-c)^4} \frac{1}{2(1-z)^2} \frac{1}{2(1-z-c)^2}\mathrm{d}z$$
$$= \frac{3}{4}\int_{-\infty}^{-c} \frac{-1}{(1-z-c)^6(1-z)^2}\mathrm{d}z$$
$$\ge \frac{3}{4(c+1)^2}\int_{-\infty}^{-c} \frac{-1}{(1-z-c)^6}\mathrm{d}z = \frac{-3}{20(c+1)^2}.$$

For the second term, we obtain

$$\dagger_2 = \int_{-c}^0 \frac{3}{(z+c+1)^4} \frac{1}{2(1-z)^2}\left(1 - \frac{1}{2(z+c+1)^2}\right)\mathrm{d}z$$
$$= \frac{3}{4}\int_{-c}^0 \frac{2}{(z+c+1)^4(1-z)^2} - \frac{1}{(z+c+1)^6(1-z)^2}\mathrm{d}z$$
$$\ge \frac{3}{4}\int_{-c}^0 \frac{1}{(z+c+1)^4(1-z)^2}\mathrm{d}z \qquad (\because \tfrac{1}{(z+c+1)^2} \le 1, \forall z \in [-c,0])$$
$$\ge \frac{3}{4(c+1)^2}\int_{-c}^0 \frac{1}{(z+c+1)^4}\mathrm{d}z = \frac{1}{4(c+1)^2} - \frac{1}{4(c+1)^5}.$$

For the third term, we obtain

$$\dagger_3 = \int_0^\infty \frac{3}{(z+c+1)^4}\left(1 - \frac{1}{2(z+1)^2}\right)\left(1 - \frac{1}{2(z+c+1)^2}\right)\mathrm{d}z$$

$$\geq \int_0^\infty \frac{3/2}{(z+c+1)^4}\left(1 - \frac{1}{2(z+c+1)^2}\right)\mathrm{d}z$$

$$= \frac{1}{2(c+1)^3} - \frac{3}{20(c+1)^5}.$$

Therefore,

$$\int_{-\infty}^\infty -f'(z+\underline{\lambda}_i)\prod_{j\neq i} F(z+\underline{\lambda}_j)\mathrm{d}z \geq \frac{1}{10(c+1)^2} + \frac{1}{2(c+1)^3} - \frac{8}{20(c+1)^5}$$

$$\geq \frac{1}{10(c+1)^2}.$$

For the denominator term, we obtain

$$\ddagger_1 = \int_{-\infty}^{-c} \frac{1}{(1-z-c)^3}\frac{1}{2(1-z)^2}\frac{1}{2(1-z-c)^2}\mathrm{d}z$$

$$\leq \frac{1}{4(c+1)^2}\int_{-\infty}^{-c} \frac{1}{(1-z-c)^5}\mathrm{d}z = \frac{1}{16(c+1)^2}.$$

For the second term, we obtain

$$\ddagger_2 = \int_{-c}^0 \frac{1}{(z+c+1)^3}\frac{1}{2(1-z)^2}\left(1 - \frac{1}{2(z+c+1)^2}\right)\mathrm{d}z$$

$$\leq \int_{-c}^0 \frac{1}{(z+c+1)^3}\frac{1}{2(1-z)^2}\mathrm{d}z.$$

By partial fractional decomposition, one can obtain

$$\frac{1}{(z+c+1)^3}\frac{1}{(1-z)^2} = \frac{1/(c+2)^2}{(z+c+1)^3} + \frac{2/(c+2)^3}{(z+c+1)^2} + \frac{3/(c+2)^4}{z+c+1} + \frac{1/(c+2)^3}{(1-z)^2} + \frac{3/(c+2)^4}{1-z}.$$

Therefore,

$$\int_{-c}^0 \frac{1}{(z+c+1)^3}\frac{1}{2(1-z)^2}\mathrm{d}z$$

$$= \frac{1}{4(c+2)^2}\left(1 - \frac{1}{(c+1)^2}\right) + \frac{1}{(c+2)^3}\left(1 - \frac{1}{c+1}\right) + \frac{3\log(c+1)}{2(c+2)^4}$$

$$+ \frac{1}{2(c+2)^3}\left(1 - \frac{1}{c+1}\right) + \frac{3\log(c+1)}{2(c+2)^4}$$

$$\leq \frac{1}{4(c+2)^2} + \frac{3}{2(c+2)^3} + \frac{3c}{(c+2)^4}$$

$$\leq \frac{1}{4(c+2)^2} + \frac{9}{2(c+2)^3}$$

$$\leq \frac{10}{4(c+2)^2} \leq \frac{5}{2(c+1)^2}.$$

For the third term, we obtain

$$\ddagger_3 = \int_0^\infty \frac{1}{(z+c+1)^3}\left(1 - \frac{1}{2(z+1)^2}\right)\left(1 - \frac{1}{2(z+c+1)^2}\right)\mathrm{d}z$$

$$\leq \int_0^\infty \frac{1}{(z+c+1)^3}\mathrm{d}z$$

$$\leq \frac{1}{2(c+1)^2}.$$

Therefore, for $K = 3$, when $\lambda = (0, c, c)$

$$\int_{-\infty}^{\infty} f(z + \underline{\lambda}_i) \prod_{j \neq i} F(z + \underline{\lambda}_j) \mathrm{d}z \leq \frac{3 + 1/16}{(c+1)^2}. \tag{42}$$

In sum, for $K = 3$, when $\lambda = (0, c, c)$, we obtain

$$\frac{-\phi_i'(\lambda)}{\phi_i(\lambda)} \geq \frac{1}{10} \frac{1}{3 + 1/16} \geq \frac{1}{31}.$$

Moreover, we obtain

$$\frac{-\phi_i'(\lambda)}{\phi_i^{3/2}(\lambda)} \geq \frac{1/10}{7^3/(4^3 \cdot 3\sqrt{3})}(c+1) = \frac{96\sqrt{3}}{1715}(c+1) \geq \frac{c+1}{11}. \qquad \Box$$

