# OpenReview forum: "Revisiting Follow-the-Perturbed-Leader with Unbounded Perturbations in Bandit Problems"
_NeurIPS.cc/2025/Conference — NeurIPS 2025 poster_

### Official Review · Reviewer_FXse · 2025-06-04

**Clarity:** 3
**Significance:** 2
**Originality:** 3
**Rating:** 4
**Confidence:** 3

**Summary:**

This paper studies FTPL for MAB. While all existing results either use unbounded symmetric perturbation (e.g., Laplace noise) or non-negative Frechet-type noise. This work is the first to show that BOBW bounds can be achieved even using unbounded but asymmetric perturbation, that is, Frechet distribution on both sides but with different tails.

**Questions:**

n/a

**Ethical Concerns:**

["NO or VERY MINOR ethics concerns only"]

**Final Justification:**

my final recommendation would be 4: Borderline accept. I highly appreciate the technical novelty of this paper but I'm not sure about how useful the results could be, e.g., establishing connections to FTRL

**Limitations:**

yes

**Quality:**

3

**Strengths And Weaknesses:**

Strengths

1. BOBW guarantee of FTPL with asymmetric unbounded perturbation is a new result. It also needs new analysis.
2. Sharp remarks/discussions are presented.

Weaknesses

1. My main concern is the motivation of the significance of the result. "These results not only extend the BOBW results of FTPL but also offer new insights into designing alternative FTPL policies competitive with hybrid regularization approaches." How is asymmetric perturbation exactly connected to hybrid regularier in FTRL? How would this new result let us mimick the effect of hybrid regularizer from FTRL into FTPL?

---

> ### Author Rebuttal · Authors · 2025-07-30
>
> Thank you for taking the time to review our paper. We have addressed your questions and comments below.
>
> ---
>
> ### Q1. How is asymmetric perturbation exactly connected to hybrid regularizer in FTRL
>
> In FTRL, additional regularization terms are often introduced to stabilize the arm-selection probabilities, e.g., to prevent abrupt changes as in Jin et al. [30].
> In the FTPL framework, such stabilizing effect can be implicitly controlled by the choice of the perturbation distribution.
> Asymmetric perturbations can therefore be interpreted as a way to control the behavior of FTPL, where the lighter left tail introduces a stabilizing effect similar to that of an auxiliary regularizer in FTRL, such as Shannon entropy [54].
> While similar effects might be reproduced by summing two independent semi-infinite perturbations with different tail indices, we expect that introducing a small probability of negative perturbations can subtly increase the conservativeness of FTPL, without altering its overall structure.
>
> However, it remains open which class of perturbations actually corresponds to hybrid regularizers.
> This correspondence may be related to either a summation of independent perturbations or a single perturbation with asymmetric tail behavior, as considered in our work.
> Nevertheless, we believe that asymmetric perturbations, particularly those supported on $\mathbb{R}$ provide a reasonable starting point, motivated by the regularization–perturbation duality from discrete choice theory, as discussed in Section 3.2 and in our first response to Reviewer vYoW.

---

> > ### Comment · Reviewer_FXse · 2025-08-05
> >
> > I thank authors for their response. I have no doubt on the technical side, but I'm just not sure about the impact/implication of those results.
> >
> > I will engage in further discussion with reviewers and AC regarding this.

---

> ### Author Response · Authors · 2025-08-06
>
> We thank the reviewer for the constructive feedback. Regarding the impact and implications of our results, we have briefly summarized our contributions in our response to Reviewer zV4p. We are happy to provide any additional clarification if needed.

---

### Official Review · Reviewer_vYoW · 2025-06-30

**Clarity:** 3
**Significance:** 4
**Originality:** 3
**Rating:** 5
**Confidence:** 4

**Summary:**

This work generalizes the BOBW analysis in previous works [1,2] by showing that FTPL can achieve BOBW results using symmetric unbounded Fréchet-type perturbations. Further, the authors also show that FTPL with the symmetric Pareto distribution with shape 2 is also feasible to achieve the BOBW results, in the 2-arm setting. On the negative side, the authors also demonstrate that in the general MABs with more than 2 arms, FTPL with symmetric Fréchet-type perturbations does not satisfy the usual condition for establishing BOBW results.

[1] Honda et al. Follow-the-Perturbed-Leader achieves best-of-both-worlds for bandit problems. ALT, 2023.

[2] Lee et al. Follow-the-perturbed-leader with Fréchet-type tail distributions: Optimality in adversarial bandits and best-of-both-worlds. COLT, 2024.

**Questions:**

1. There seems to be a concurrent work [3] which simply uses the i.i.d. Fréchet distribution with shape 2 on each base arm and achieves the BOBW results for $m$-set CMAB. Could the authors briefly discuss whether the results of FTPL with hybrid perturbations can be extended to the CMAB setting? If so, are there any advantages of the method in this work compared to that in [3]?

[3] Zhan et al. Follow-the-Perturbed-Leader Approaches Best-of-Both-Worlds for the m-Set Semi-Bandit Problems. arXiv, 2025.

**Ethical Concerns:**

["NO or VERY MINOR ethics concerns only"]

**Final Justification:**

The authors have resolved all my previous questions.

**Limitations:**

yes

**Quality:**

3

**Strengths And Weaknesses:**

**Strengths**

1. **Motivation and Novelty**: Given the progress of FTRL with hybrid regularizers in various bandit problems, it is of great interest to study FTPL with hybrid perturbations. I think both the techniques and the results in this work are valuable and novel to the online learning literature.
2. **Writing**: This paper is generally well written.

**Weaknesses**

Though most parts of the writing are clear, I personally feel that the writing may be further improved if some parts of the paper could be further explained or discussed. For instance,
1. For the term “unbounded perturbations” throughout the paper, I am not fully sure how the authors define it. For me, [1] consider the Fréchet distribution with shape 2 on the support $\mathbb{R}_{\ge 0}$, which seems to be “unbounded” to me. However, after a complete reading of this paper,  I feel that the authors seem to denote “unbounded distribution” by meaning that a distribution has support on both the positive and negative parts.
2. From Line 2 to Line 8, the authors seem to express an opinion that FTRL with a hybrid regularizer (roughly) corresponds to FTPL with perturbation, which has support on both the positive and negative parts. I am wondering whether FTRL with a hybrid regularizer can (roughly) correspond to FTPL with a summation of two independent perturbations that only have supports on the positive parts. If so, shouldn’t it be an easier starting point to investigate the FTPL counterparts of FTRL with hybrid regularizers?

Besides, as the main results in the paper body are a bit heavy, I would like to suggest that the authors include a table summarizing the results of this work. I understand that it might be difficult to fit into the main body. Perhaps including such a table in the appendix might also benefit the readers.

---

> ### Author Rebuttal · Authors · 2025-07-30
>
> Thank you for taking the time to review our paper and for your valuable feedback.
> We will include a summary table or figure (of Venn diagram) in the revised version to provide a clear overview of the position and scope of our results.
> As pointed out, the term "unbounded distribution" was intended to refer to distributions with support on both the positive and negative real line.
> We acknowledge that this terminology was unclear and will revise the manuscript to clarify our notation accordingly.
>
> We address your specific questions and comments below.
> Throughout, by "semi-infinite support" we mean a distribution satisfying $\mathrm{supp} F \subseteq [\nu,\infty)$ for some finite $\nu\in \mathbb{R}$.
>
> ---
>
> ### Q1. Hybrid regularization and hybrid perturbation (weakness 2)
>
> We agree that summing two independent perturbations with semi-infinite supports and different tail indices $\alpha_1, \alpha_2$ can be a promising and perhaps more accessible starting point.
> However, it remains unclear what forms of such perturbations can effectively mimic hybrid regularizers that achieve a BOBW guarantee.
> Hence, it would be more beneficial to find a more systemic approach rather than adhoc approach.
> In previous hybrid regularization approaches, the regularizer is constructed from two distinct types.
> To name a few examples, as discussed in Line 82-89:
> * Zimmert et al. [54] used Tsallis entropy with Shannon entropy applied to the complement in combinatorial semi-bandits, i.e., $-\sqrt{x} + \gamma (1-x)\log(1-x)$
> * Jin et al. [30] used a Tsallis entropy with log-barrier to extend BOBW guarantees of FTRL in the MAB setting $-\sqrt{x} + \gamma \log x$.
>
> In some earlier works (not necessarily in the BOBW context), hybrid regularizers are designed to penalize small probabilities, i.e., $x\to 0$, typically using log-barrier terms like $\log x$.
> In such cases, summing two positive-support perturbations may capture the desired behavior as you mentioned.
> However, in the BOBW literature, especially for combinatorial semi-bandits, hybrid regularizers commonly include an additional penalty near $x\to 1$, such as $(1-x)\log(1-x)$ (Shannon entropy to the complement).
> To the best of our knowledge, all known BOBW hybrid regularizers in combinatorial settings fall into this category [28, 54]. Capturing such effects via FTPL, penalizing both over-exploitation ($x\to 1$) and insufficient exploration ($x \to 0$), likely requires perturbations with support on both positive and negative sides.
>
> From this perspective, a natural candidate for hybrid perturbation in more complex settings would be a mixture or composition of Fréchet-type and another perturbation corresponding to a non-Tsallis entropy defined on $\mathbb{R}$.
> As discussed in Section 3.2, the perturbations on $\mathbb{R}$ correspond to generalized entropies, via the inverse gradient of the potential function, which defines the associated FTRL policy.
> In this sense, perturbations on $\mathbb{R}$ may offer a more unified and systematic way to study the behavior of FTPL with the regularization–perturbation duality, particularly when leveraging insights from discrete choice theory.
> For this reason, we view the study of perturbations on $\mathbb{R}$ as a more principled and broadly applicable approach to designing FTPL policies.
>
> ---
>
> ### Q2. Advantages of hybrid perturbations compared to Zhan et al. (2025) (question 1)
>
> Thank you for pointing out this interesting reference.
> It is indeed interesting that a simple choice of Fréchet perturbations can achieve the BOBW guarantee.
>
> As far as we understand, one limitation of the result in Zhan et al. (2025) lies in the looseness of the minimax regret bound, where they obtain a near-optimal guarantee of $O(\sqrt{mT}(\sqrt{d\log d}+m^{5/6}))$.
> While this improves upon the previous FTPL bound with exponential distribution of $O(m\sqrt{dT\log(d/m)})$ by Neu and Bartok [41], especially when $m$ does not scale with $d$, it remains suboptimal compared to the optimal $O(\sqrt{mdT})$ bounds by FTRL methods in Zimmert et al. [54] and Ito [28].
> As Zhan et al. (2025) noted, achieving tighter bounds (and BOBW guarantee) are likely to require fundamentally different or more refined techniques.
> In that context, designing hybrid perturbations could be one of promising directions.

---

> > ### Comment · Reviewer_vYoW · 2025-08-03
> >
> > I appreciate the authors' responses, which have addressed all my concerns. I will maintain my rating for this work.

---

> > > ### Author Response · Authors · 2025-08-06
> > >
> > > We thank the reviewer for the insightful and constructive feedback. We will incorporate these suggestions (especially the table/figure) and clarify our notation in the revised manuscript.

---

### Official Review · Reviewer_QAp1 · 2025-07-01

**Clarity:** 3
**Significance:** 3
**Originality:** 3
**Rating:** 5
**Confidence:** 3

**Summary:**

The authors study Follow-the-Perturbed-Leader (FTPL) policies for a large class of asymmetric distributions and derive Best-of-Both-Worlds (BOBW) regret guarantees, i.e., that are optimal in both the stochastic and adversarial settings. The authors also study FTPL with symmetric distributions, where they show that BOBW guarantees are achievable when the number of arms is $K=2$, but that their analysis do not generalize to $K \ge 3$.

**Questions:**

1. Can you give insights regarding why the stochastic regret guarantee of Corollary 4.2 only holds for alpha=2?
2. Regarding the implications of the negative result in Section 5.3, as you note in Remark 5.7, your results do not constitute a full impossibility proof. Could you elaborate on what this would imply for the prospect of achieving BOBW guarantees with symmetric perturbations? Does it suggest that an entirely new analytical technique is needed, or that perhaps only a small subset of "well-behaved" symmetric distributions might work?

**Ethical Concerns:**

["NO or VERY MINOR ethics concerns only"]

**Final Justification:**

The authors have addressed my questions well. I remain confident that this paper contributes significantly to the FTPL literature, so I am keeping my score.

**Limitations:**

yes

**Paper Formatting Concerns:**

no major concerns

**Quality:**

4

**Strengths And Weaknesses:**

This paper is theoretically very solid. All statements in the main text are proven in great detail in the appendices.

I found the paper to be well-written and generally clear. The extensive literature review in Sections 2,3, and Appendix A.1 help greatly with situating this work and understanding its contributions. Section 4 is well organized and it is commendable that the leading constants are tracked in the statements of Theorems 4.3 and 4.4. However, I found the exposition in Section 5 somewhat harder to follow.

In my opinion, the main contribution of the paper is the derivation of BOBW regret guarantees for FTPL under asymmetric perturbations  (Section 4). The BOBW guarantee for symmetric perturbations in two-armed bandits (Section 5.2) is also a meaningful contribution, and the negative results of Section 5.3 are particularly valuable for future work,  but as the authors point out in Remark 5.7, they only show the failure of a specific analysis technique, which does not entail that FTPL with symmetric frechet-type distributions cannot achieve BOBW guarantees.

Additionally, I have noticed a few minor typos and/or potentially confusing wording (which do not impact my overall positive assessment):
- Line 105: `might be enable'
- Line 135: I believe there is a typo in the definition of the importance-weighted estimator. Shouldn't its $i$-th coordinate be  $\frac{\ell_{t,I_t}}{\mathbb{P}(I_t=i)}\mathbb{1}[I_t=i]$ ?
- Line 250: 'let $\lambda_i$ is the...' -> `let $\lambda_i$ be the...'. I also personally found the wording "let $\lambda_i$ be the $\sigma_i$-th smallest among $\lambda_1,\dots,\lambda_K$" slightly confusing, and would recommend something like "let $\sigma_i$ be the position of $\lambda_i$ in the nondecreasing ordering of $\{\lambda_1,...\lambda_K\}$"
- Line 298: 'if exists' -> 'if it exists'
- Line 349: 'do not occur' -> `that does not occur'

---

> ### Author Rebuttal · Authors · 2025-07-30
>
> Thank you for taking the time to review our paper and pointing several typos in the current manuscript. We will revise the manuscript accordingly.
>
> ---
>
> ### Q1. Insights on optimal result only for $\alpha= 2$ in stochastic setting
>
> While $\alpha$ also influences the leading constants in the adversarial setting [37], its role is particularly critical in the stochastic setting, where the logarithmic regret is the goal.
> The tail index $\alpha$ controls the trade-off between exploration and exploitation in FTPL.
> A small $\alpha$ induces heavier tails and more frequent large deviations, encouraging exploration (e.g., by selecting arms with large $\hat{L}_t$) but potentially preventing exploitation.
> Conversely, larger $\alpha$ leads to lighter tails and more exploitative behavior, but with insufficient exploration to correct early misestimates.
>
> This phenomenon is analogous to the role of $\beta$ in $\beta$-Tsallis entropy, which similarly influences the exploration–exploitation balance of FTRL.
> In particular, Zimmert and Seldin [53] identified $\beta = 1/2$ (corresponding to $\alpha = 2$ in our setting) as a special case that achieves optimality in both cases, and provided an intuitive explanation for its effectiveness in their Section 4.3.
> They also discussed why $\beta \neq 1/2$ may lead to suboptimality in Section 4.2, which further supports the delicate balance needed in tuning this parameter.
>
> ---
>
> ### Q2. Further implication on BOBW guarantee with symmetric perturbations
>
> As you mentioned, our current analysis does not formally rule out the possibility of achieving a BOBW guarantee with symmetric perturbations.
> While we do not provide a concrete counterexample, we conjecture the existence of an adaptive adversary that induces $\Omega(\sqrt{KT\log K})$ regret in this setting.
>
> This conjecture is analogous to that in Abernethy et al. [2], who showed that FTPL achieves $O(\sqrt{KT\log K})$ regret when the hazard function $-\phi_i'/\phi_i$ is uniformly bounded.
> Their analysis is based on the following decomposition, believed to be tight:
>
> $$
> \begin{aligned}
> \sum\_{t=1}^T \mathbb{E}\left[\langle w\_t - e\_{i^* }, \hat{\ell}\_t \rangle\right]
>     &= \sum\_{t=1}^T \mathbb{E}\left[\langle \phi(\eta_t \hat{L}\_{t}) - \phi(\eta_t \hat{L}\_{t+1}), \hat{\ell}\_t \rangle\right] + \sum\_{t=1}^T \mathbb{E}\left[\langle \phi(\eta\_t \hat{L}\_{t+1}) - e_{i^*}, \hat{\ell}\_t \rangle\right] \\\\
>     &\geq \sum\_{t=1}^T \mathbb{E}\left[\langle \phi(\eta\_t \hat{L}\_{t}) - \phi(\eta\_t \hat{L}\_{t+1}), \hat{\ell}\_t \rangle\right] \\\\
>     &\approx \sum\_{t=1}^T \sum\_{i=1}^K \mathbb{E}\left[\langle -1[I\_t = i] \eta\_t \ell\_{t,i} \frac{\phi\_i'(\eta\_t \hat{L}\_t)}{\phi\_i(\eta\_t \hat{L}\_t)}, \hat{\ell}\_{t,i} \rangle\right] \\\\
>     &\approx \sum\_{t=1}^T \sum\_{i=1}^K \mathbb{E}\left[-\eta\_t \ell\_{t,i}^2 \frac{\phi\_i'(\eta\_t \hat{L}\_t)}{\phi\_i(\eta\_t \hat{L}\_t)}\right].
> \end{aligned}
> $$
>
> To achieve $O(\sqrt{KT\log K})$ regret, it suffices to uniformly bound $\phi_i'/\phi_i$ under appropriate $\eta_t$.
> Since this is not possible for Gaussian perturbations (which have unbounded hazard), they conjectured that FTPL with Gaussian perturbations suffers linear regret under an adaptive adversary.
> In our case, the goal is to achieve the sharper $O(\sqrt{KT})$ regret, which requires $\phi_i'$ to be smaller than $\phi_i^{3/2}$ rather than $\phi_i$.
> Proposition 5.5 shows that this is not valid under FTPL with symmetric perturbations, which strongly suggests the suboptimality of symmetric perturbations.
> Although our analysis does not yield a formal lower bound, the level of informality is similar to that of Abernethy et al. [2], as both rely on decompositions regarded as tight and use them to argue that certain perturbations likely fail in adversarial settings.

---

> > ### Comment · Reviewer_QAp1 · 2025-08-03
> >
> > Thank you for addressing my questions. I have no further concerns and will keep my score.

---

> > > ### Author Response · Authors · 2025-08-06
> > >
> > > We appreciate your insightful and constructive feedback on our manuscript. We will incorporate your suggestions and correct the typos in the revised version.

---

### Official Review · Reviewer_zV4p · 2025-07-03

**Clarity:** 4
**Significance:** 3
**Originality:** 2
**Rating:** 4
**Confidence:** 3

**Summary:**

Follow-the-Regularized-Leader (FTRL) and Follow-the-Perturbed-Leader (FTPL) are well-known meta-algorithms for online learning problems that achieve optimal rates in both adversarial and stochastic settings. In some cases, they even attain a best-of-both-worlds (BOBW) guarantee.

For FTRL, prior work has shown that hybrid regularizers, typically combining negative entropy and Tsallis entropy, can be used to obtain a BOBW guarantee. For FTPL, recent research has demonstrated how to use hybrid perturbations to achieve a similar BOBW result.

This paper extends that line of work by refining the conditions on the types of perturbations used in FTPL and by analyzing both asymmetric and symmetric unbounded perturbations. Specifically, the authors introduce:

A family of asymmetric perturbations that can achieve BOBW guarantees in the general case.

A family of symmetric perturbations that achieve BOBW guarantees in the special case of two arms.

**Questions:**

See the above.

Line 156, Possible typo?

**Ethical Concerns:**

["NO or VERY MINOR ethics concerns only"]

**Final Justification:**

POST REBUTTAL:
My initial score may have been a bit harsh, so I’m raising it from 3 to 4 (borderline accept).
The technical contribution is solid, and the problem is both interesting and important.

**Limitations:**

Limitations are addressed.

**Quality:**

3

**Strengths And Weaknesses:**

Strengths:

The paper deepens our theoretical understanding of FTPL, particularly how to design hybrid perturbations that can mimic the behavior of FTRL with hybrid regularizers.

The study advances the long-term goal of understanding how FTPL can be made comparable to FTRL in a principled and efficient way.

There are several technically non-trivial contributions related to the analysis of FTPL, which I consider to be the main strength of the paper.

Weaknesses:

Although the paper provides additional insights into how to design FTPL algorithms with BOBW guarantees, prior work has already established that FTPL can achieve BOBW under certain settings.

As a result, the contribution of this paper feels incremental.

I am unsure whether the contribution is significant enough. For me, it is borderline.

---

> ### Author Rebuttal · Authors · 2025-07-30
>
> Thank you for taking the time to review our paper.
>
> > Comment: The contribution of this paper feels incremental.
>
> We believe our contribution goes beyond distributional extensions in Section 4 as explained below.
> By considering perturbations fully supported on $\mathbb{R}$, it becomes possible to clearly connect FTPL to discrete choice theory (Section 3, Appendix A.3), which offers a more principled approach for understanding and designing FTPL.
> In addition, Section 5.2 proves the first BOBW guarantee under symmetric heavy-tailed perturbations for $K=2$, building on the FTPL–FTRL duality discussed in Section 5.1.
> Section 5.3 then establishes a new negative result for $K\geq 3$, revealing the fundamental limitations of current analysis techniques.
> Taken together, these contributions go beyond simply showing that “another distribution works,” as they advance theoretical understanding and suggest new directions for FTPL design grounded in duality and tail behavior.
>
> ---
>
> A more detailed explanation is summarized below:
>
> ### A1. FTPL–FTRL correspondence and regret analysis beyond known cases
>
> One motivation of our work is to explore when the recent successes of FTRL, particularly with Tsallis entropy and hybrid regularizers, can be replicated by FTPL policies, which are simpler and optimization-free.
> In this regard, we revisit the relationship between regularization and perturbation and investigate under what conditions such design is possible in Section 3, which may inspire the future design of FTPL.
>
> In the two-arm setting, we show that FTRL with Tsallis entropy corresponds to FTPL with symmetric Fréchet-type tails, which numerically resemble symmetric Pareto distributions (Section 5.2).
> We then establish the first BOBW guarantee for this symmetric unbounded case not covered in previous FTPL work.
>
> ---
>
> ### A2. Revealing limitations for symmetric perturbations in the general case
>
> In contrast to the positive two-arm result, Section 5.3 shows that FTPL with symmetric perturbations fails to achieve BOBW guarantees for $K\geq 3$ under very widely used framework.
> Proposition 5.5 shows that key stability terms can be unbounded in this case, revealing a concrete limitation of current analysis techniques.
> This suggests that perturbation designs effective in two-arm settings may not generalize to larger action spaces, revealing the limitation on the approach that relies on the FTRL–FTPL correspondence established in the two-arm case [3, 32].
>
> As briefly discussed in Section 1 and Remark 5.7, Abernethy et al. [2] analyzes FTPL using $-\phi_i'/\phi_i$ term, showing near-optimal $O(\sqrt{KT\log K})$ regret for perturbations with bounded hazard functions.
> Since their regret decomposition is quite tight, they conjectured that FTPL with Gaussian perturbations, whose hazard functions are unbounded, could suffer linear regret under an adaptive adversary.
> In our case, where the goal is $O(\sqrt{KT})$ regret, the analysis relies on the term $-\phi_i'/\phi_i^{3/2}$ as done in recent BOBW studies for FTRL and FTPL [27, 30, 37, 53].
> Proposition 5.5 thus strongly suggests the existence of such adversary that forces FTPL with symmetric perturbations to suffer $\Omega(\sqrt{KT\log K})$ regret.
> Since the current decomposition is already almost tight (up to minor refinements), our results at least show that the current technique fails to obtain optimal regret even though the current decomposition is already almost tight.
> Please also see the response to Q2 of Reviewer QAp1.
>
> ---
>
> ### A3. Bridging to discrete choice theory
>
> We also see our results as contributing to a broader unification of FTPL design and discrete choice models extensively studied in econometrics and operations research.
> While prior FTPL analyses in the ML literature were often focused on perturbations with nonnegative support, our generalization to perturbations on $\mathbb{R}$ aligns more naturally with well-established (discrete choice) models like the multinomial logit or nested logit.
> This connection enables the application of insights and techniques from those mature fields, which may inspire future FTPL design in online learning contexts.

---

> > ### Comment · Reviewer_zV4p · 2025-08-05
> >
> > Thanks for your detailed response.
> > I have read the other reviews as well.
> > My initial score may have been a bit harsh, so I’m raising it from 3 to 4 (borderline accept).
> > The technical contribution is solid, and the problem is both interesting and important.

---

> > > ### Author Response · Authors · 2025-08-06
> > >
> > > We appreciate your constructive feedback on our manuscript.
> > > We will clarify the contributions of our paper in the revised version.

---

### Decision · Program_Chairs · 2025-09-17

**Decision:**

Accept (poster)

**Comment:**

The paper studies the Follow-the-Perturbed-Leader family of bandit meta-algorithms known for achieving optimal scaling for both stochastic and adversarial settings (the so called Best of Both Worlds guarantees). Here the authors study a family of perturbation distributions for FTPL policies and derive BoBW guarantees for asymmetric unbounded perturbations. They further derive similar guarantees for symmetric perturbations in problem instances with two arms ($K=2$), and demonstrate that there exist problem instances with $K>2$ where their analysis technique does not suffice (the conditions shown to be necessary for the derivation of BoBW guarantees do not hold).

Overall the paper is well written and the reviewers agree that the results further the understanding of an important family of methods, FTPL, and details further connections with the FTRL (Follow the regularized leader) family of meta-algorithms. Reviewers explicitly mention the importance of studying hybrid perturbations in the FTPL, given the success of hybrid regularizers in the FTRL approaches and the lighter computation required in FTPL family of algorithms.

In light of this contribution, I recommend acceptance for this submission and encourage the authors to address the reviewer comments in their final version.